# TFRBench: A Reasoning Benchmark for Evaluating Forecasting Systems

**Md Atik Ahamed** [* 1 2]   **Mihir Parmar** [* 1]   **Palash Goyal** [1]   **Yiwen Song** [1]   **Long T. Le** [1]   **Qiang Cheng** [2]
**Chun-Liang Li** [1]   **Hamid Palangi** [1]   **Jinsung Yoon** [1]   **Tomas Pfister** [1]

## Abstract

We introduce *TFRBench*, the first benchmark designed to evaluate the reasoning capabilities of forecasting systems. Traditionally, time-series forecasting has been evaluated solely on numerical accuracy, treating foundation models as "black boxes." Unlike existing benchmarks, *TFRBench* provides a protocol for evaluating the reasoning generated by forecasting systems–specifically their analysis of cross-channel dependencies, trends, and external events. To enable this, we propose a systematic multi-agent framework that utilizes an iterative verification loop to synthesize numerically grounded reasoning traces. Spanning ten datasets across five domains, our evaluation confirms that this reasoning is causally effective; useful for evaluation; and prompting LLMs with our generated traces significantly improves forecasting accuracy compared to direct numerical prediction (e.g., avg. $\sim 40.2\% \rightarrow 56.6\%$), validating the quality of our reasoning. Conversely, benchmarking experiments reveal that off-the-shelf LLMs consistently struggle with both reasoning (lower LLM-as-a-Judge scores) and numerical forecasting, frequently failing to capture domain-specific dynamics. *TFRBench* thus establishes a new standard for interpretable, reasoning-based evaluation in time-series forecasting. Our benchmark is available at: https://tfrbench.github.io

## 1. Introduction

Time-series forecasting is a foundation of decision-making across critical domains, from energy grid management and

This work was done while Md Atik Ahamed was a student researcher at Google Cloud AI Research. [*]Equal contribution [1]Google [2]University of Kentucky. Correspondence to: Md Atik Ahamed <atikahamed@uky.edu>, Mihir Parmar <mihirparmar@google.com>.

*Proceedings of the $43^{rd}$ International Conference on Machine Learning*, Seoul, South Korea. PMLR 306, 2026. Copyright 2026 by the author(s).

supply chain logistics to financial market analysis (Petelin et al., 2023; Kong et al., 2025). Traditionally, the evaluation of forecasting systems has been dominated by numerical metrics such as Mean Absolute Scaled Error (MASE) or Root Mean Squared Error (RMSE) (Li & Law, 2024). While recent statistical (Makridakis & Hibon, 1997) and deep learning foundation models (Das et al., 2024; Ansari et al., 2024) have achieved impressive performance on these metrics, they predominantly operate as "black boxes" by ingesting historical data and outputting numerical predictions without offering insights into the underlying reasoning.

This exclusive focus on numerical performance creates a significant gap in reliability and explainability (Rudin, 2019). A model may correctly predict a spike in electricity demand, but without understanding the cause, such as a specific weather event or a holiday. Evaluating this capability, however, presents a non-trivial challenge: unlike numerical targets, *ground truth* reasoning does not naturally exist in recent time-series benchmarks. Constructing it requires synthesizing and verifying many causal chains to ensure explanations are factually aligned with numerical outputs. As a result, to the best of the authors' knowledge, there are no standardized benchmarks or metrics for forecasting that capture the reasoning quality behind the numerical forecast.

To address this, we introduce *TFRBench*, a novel benchmark for reasoning-aware time-series forecasting. We define "reasoning" in this context as a model's ability to analyze input data across multiple channels, identify and explain interdependencies, and justify significant forecast events based on external context. Unlike traditional benchmarks (Li et al., 2025; Chen et al., 2025), *TFRBench* targets the evaluation of systems that generate natural reasoning with numerical forecasts. Figure 1 illustrates this paradigm shift, contrasting the traditional approaches of previous benchmarks with the reasoning-aware forecasting that our benchmark supports.

To overcome the lack of verifiable reference reasoning, we develop *TFRBench* using a systematic multi-agent framework designed to synthesize and refine predictive logic. This architecture orchestrates five agents–Search, Reasoning, Verifier, Forecasting, and Summary–into an iterative *generate-verify-refine* loop. Specifically, the framework generates parallel reasoning traces, identifies the best candidate

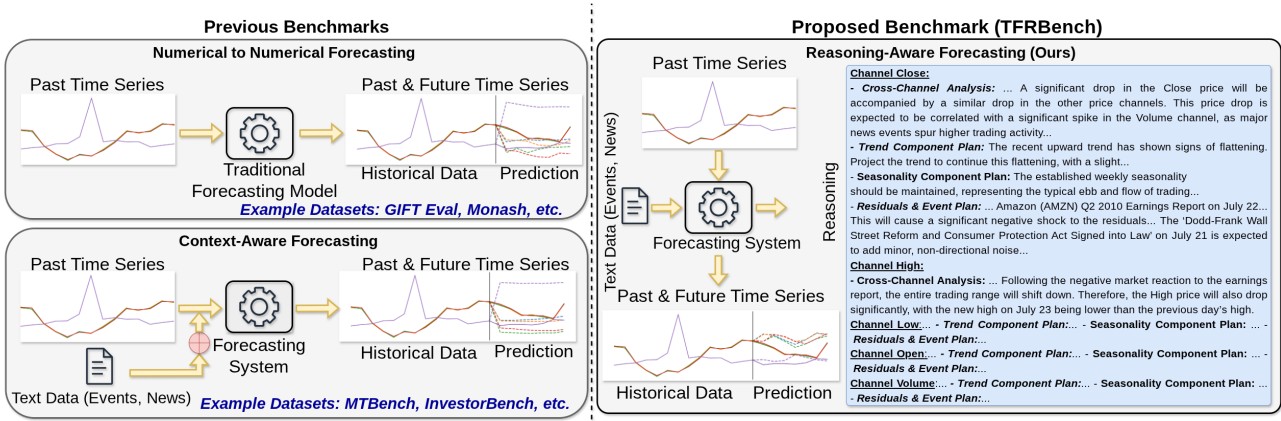

*Figure 1.* **An example task from the proposed *TFRBench*.** We augment the forecasting process by providing past time series and text data to generate a comprehensive reasoning output. This output explicitly captures cross-channel analysis, trend/seasonality component plan and residuals & event plan. The comprehensive reasoning may guide the forecasting system further for accurate forecasting.

among them using a composite quality score, and iteratively refines until the resulting forecast outperforms a naive baseline. This process yields a high-quality benchmark spanning ten datasets across five diverse domains (Energy, Sales, Web/CloudOps, Transportation, and Finance). We validate the efficacy of this generated data through extensive quantitative analysis, demonstrating high overall success rates (Figure 6), systematic feedback-driven improvement, strong verifier reliability, and many more (App. C), thereby ensuring the generated traces serve as robust reference standards.

Finally, we evaluate a suite of frontier LLMs, and time-series foundation models on *TFRBench*. Using our dual-evaluation protocol–combining LLM-as-a-Judge for reasoning quality and MASE for numerical forecasting–we provide several findings. For instance, reasoning quality is strongly correlated with forecasting accuracy (Figure 5); models that achieve higher judge scores consistently yield lower MASE, validating the predictive utility of our benchmark. Furthermore, despite this potential, off-the-shelf LLMs exhibit significant "narrative bias," frequently generating fluent but logically inconsistent reasoning that degrade performance in stochastic domains such as Finance. By quantifying this reasoning gap, *TFRBench* thus establishes a necessary standard for diagnosing reasoning ability of LLMs and driving the development of interpretable forecasting agents.

## 2. Related Works

**Purely Numerical Benchmarks.** For decades, the standard for evaluating forecasting systems has been defined by large-scale competitions and archives. The M-Competitions (M3, M4, M5) (Makridakis et al., 2020) and the Monash Time Series Forecasting Archive (Godahewa et al., 2021) established the foundational metrics for the field. These repositories aggregate thousands of diverse time series

across domains such as finance, demographics, and macroeconomics. However, they share a critical limitation: they treat time series as isolated numerical sequences, devoid of metadata or external context. Models evaluated on these benchmarks are mainly optimized for statistical patterns (e.g., autocorrelation) without access to the causal drivers, such as weather events, holidays, or economic policy changes, that often dictate real-world outcomes (Lim & Zohren, 2021). Thus, they are unsuitable for evaluating the *semantic* reasoning capabilities of recent LLMs.

**Multimodal and Semantic Benchmarks.** Recent works have attempted to integrate textual context into forecasting. MTBench (Chen et al., 2025) pairs time series with textual news reports to evaluate temporal reasoning, but primarily focuses on Question Answering (e.g., "Did the price go up?") rather than precise numerical forecasting. Similarly, InvestorBench (Li et al., 2025) assesses LLM-based agents on financial decision-making tasks using expansive document retrieval. While pioneering, these benchmarks are often domain-restricted (e.g., exclusively Finance) or test high-level decision logic rather than the fundamental ability to predict future values based on reasoning. GIFT-Eval (Aksu et al., 2024) represents a significant step forward in scale, offering 144,000 time series for zero-shot evaluation, yet it remains a numerical benchmark that does not assess interpretability or reasoning quality.

**Numerical Foundation Models.** The success of the Transformer architecture in NLP has spurred the development of Time-Series Foundation Models (TSFMs). Models such as **TimesFM** (Das et al., 2024), **Chronos** (Ansari et al., 2024), **MOIRAI** (Woo et al., 2024), and **Lag-Llama** (Rasul et al., 2023) have achieved state-of-the-art performance. These approaches typically employ specialized tokenization strategies: either patching numerical values (Nie et al., 2023)

*Table 1.* **Comparison of *TFRBench* against existing benchmarks.** The table compares benchmarks based on various parameters. Unlike prior works that rely on statistical pattern matching or static contexts, *TFRBench* provides verifiable reference reasoning. We define the "Reasoning" here as the derivation of a numerical forecast from causal logic, distinguishing it from tasks such as trading decisions.

| Benchmark | Domains | Context Source | Primary Task | Reasoning | Goal |
|---|---|---|---|---|---|
| M4 Competition (Makridakis et al., 2020) | Diverse | ✗ | Statistical Forecast | ✗ | Pattern Matching |
| Monash Archive (Godahewa et al., 2021) | Diverse | ✗ | Statistical Forecast | ✗ | Pattern Matching |
| GIFT-Eval (Aksu et al., 2024) | Diverse | ✗ | Zero-shot Forecast | ✗ | Scaling Laws |
| MTBench (Chen et al., 2025) | Fin/Weather | **Provided (Static)** | Trend QA & Forecast | ✗ | Multimodal Fusion |
| InvestorBench (Li et al., 2025) | Finance | **Vector DB (RAG)** | Trading Decision | ✗ | Profit Maximization |
| ***TFRBench* (Ours)** | **Diverse** | **Agentic Web Search** | **Step-by-Step Reasoning** | ✓ | **Causal Attribution** |

or discretizing them into vocabulary tokens (Gruver et al., 2023) to process time series as a language. While effective at capturing complex dependencies and scaling laws, these models operate largely as "black boxes." They do not leverage any external knowledge (e.g., knowing that retail sales peak in December) or LLMs' reasoning capabilities. Our work demonstrates that while these models set a high numerical bar, they can be matched or outperformed by general-purpose LLMs when the latter are augmented with structured reasoning and external knowledge.

**Reasoning-Enhanced Forecasting.** A growing body of work explores using general-purpose LLMs (e.g., GPT-4, Gemini, Claude) for forecasting by prompting them to analyze the data textually and make accurate predictions. **PromptCast** (Xue & Salim, 2023) was among the first to frame forecasting as a sentence-to-sentence generation task. Subsequent works have introduced Chain-of-Thought (CoT) prompting to elicit intermediate reasoning steps (Wei et al., 2022). However, a critical limitation of existing CoT forecasting research is the lack of evaluation benchmarks. Studies have shown that LLMs often "hallucinate" reasoning, inventing fictitious events or causal links to justify a prediction (Ji et al., 2023). Existing benchmarks do not penalize this; a model can achieve a good numerical score for the wrong reasons. To this end, *TFRBench* provides a comprehensive reasoning benchmark for systematic evaluation of LLMs' reasoning w.r.t. forecasting task.

**Explainable Forecast.** Traditional Explainable AI (XAI) methods for time series, such as SHAP or LIME, focus on feature attribution, identifying *which* past time steps influenced the prediction (Rojat et al., 2021). While useful for debugging, these methods do not provide *causal* explanations. They can tell a user that "t-12" was important, but not *why* (e.g., "t-12 corresponds to the start of the fiscal quarter"). By generating natural language reasoning traces that explicitly reference external events and cross-channel dependencies, our approach aligns with the goal of "inherent interpretability" (Rudin, 2019), offering users a transparent narrative that builds trust in the automated forecast.

As summarized the comparison with existing datasets in Table 1, *TFRBench* provides a multi-domain evaluation suite that demands both accurate forecasting and verifiable, step-by-step reasoning which all datasets fail to provide.

## 3. *TFRBench*: Data Creation

To ensure a comprehensive and robust evaluation, we design to evaluate reasoning-aware forecasting performance across different data distributions. Our datasets in *TFRBench* span five distinct domains: Energy, Sales, Web/CloudOps, Transportation, and Economics/Finance. In particular, we utilize specific datasets, including Solar (Lai et al., 2018), Electricity (Lai et al., 2018), Car-parts (Godahewa et al., 2021), Hierarchical Sales (Mancuso et al., 2021), Bitbrains - Fast Storage (Shen et al., 2015), Web Traffic (Godahewa et al., 2021), NYC Taxi (New York City Taxi and Limousine Commission, 2025), Amazon Pricing (Yahoo! Finance, 2025), and Apple Pricing (Yahoo! Finance, 2025). Further details on each dataset are provided in App. A.1.

### 3.1. Problem Setting

This work addresses the problem of reasoning-aware forecasting, where the goal is to produce numerical predictions derived from an explicit, verifiable reasoning. Let $\mathbf{X}_{hist} = [\mathbf{x}_1, \ldots, \mathbf{x}_T]$ represent a series of random variables corresponding to historical observations in discrete time, where $\mathbf{X}_{hist} \in \mathbb{R}^{T \times C}$ with $C$ channels. Let $\mathbf{Y} = [\mathbf{y}_{T+1}, \ldots, \mathbf{y}_{T+H}]$ represent the future observations over a forecast horizon $H$. In classical statistical forecasting, the goal is to estimate the distribution of future observations given only the history, $P(\mathbf{Y}|\mathbf{X}_{hist})$. We further assume access to an external information retrieval mechanism that yields a set of events, denoted $\mathcal{E}$, complementary to $\mathbf{X}_{hist}$. Unlike standard context-aided methods that implicitly model $P(\mathbf{Y}|\mathbf{X}_{hist}, \mathcal{E})$, our framework introduces an intermediate latent variable $R$, denoting a natural language *reasoning trace*. This trace explicitly decomposes the signal into trend, seasonality, and event impacts. The task then becomes a two-stage estimation: first generating $R \sim P(R|\mathbf{X}_{hist}, \mathcal{E})$, and subsequently deriving the forecast conditioned on this reasoning, $\sim P(\mathbf{Y}|R, \mathbf{X}_{hist})$.

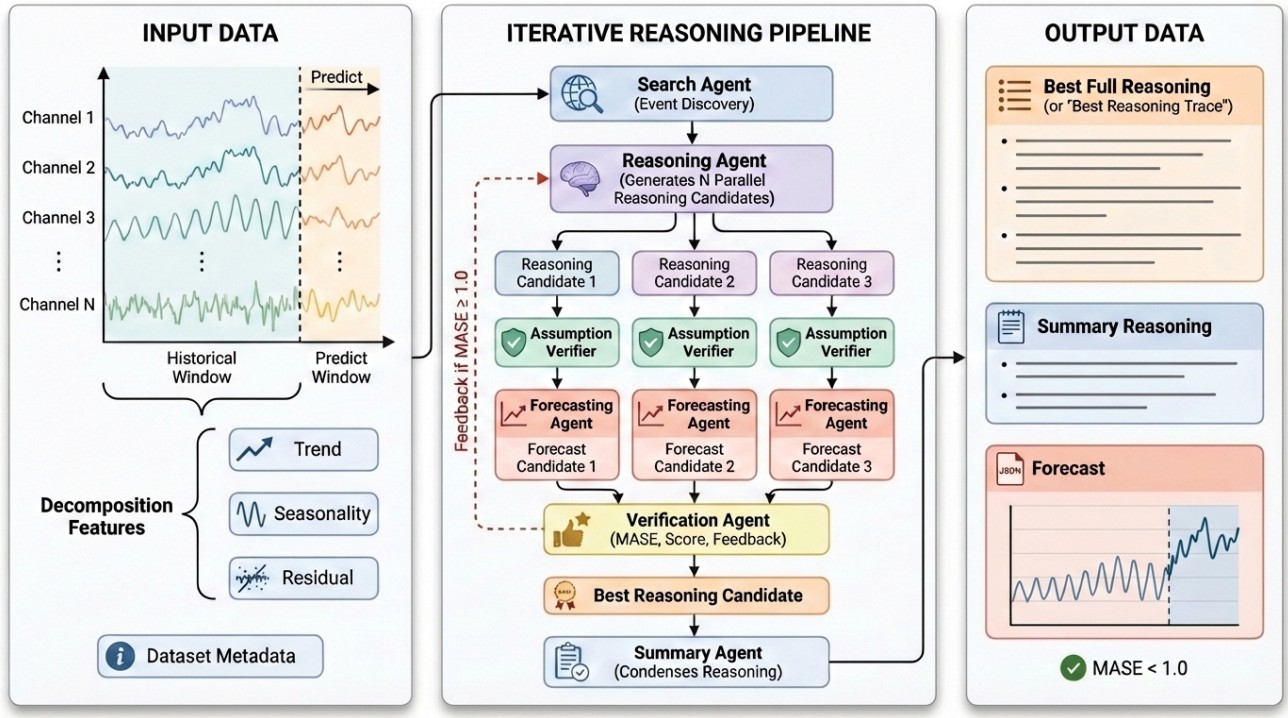

*Figure 2.* **Overview of Data Creation System.** The pipeline integrates context retrieval, assumption verification, and a feedback loop to align reasoning with forecasting. The system orchestrates agents to iteratively refine reasoning traces until the forecast outperforms the naive baseline (MASE < 1.0), and LLM verifier gives high $S$. Entire reasoning trace with each agent's outputs are provided in App. J.

## 3.2. Data Creation Pipeline

### 3.2.1. AGENT DEFINITIONS

**Search Agent** ($\mathcal{A}_{search}$). Acting as the grounding module, this agent retrieves and validates external context, and operates in two modes. First as **Event Discovery Module**, it identifies significant real-world events $\mathcal{E}$ (e.g., earnings reports, holidays, or weather events) within the forecasting timeline. To ensure that the retrieved context is well-aligned, the agent is constrained by a hyperparameter to prioritize only the most impactful, domain-relevant events. Second as **Assumption Verifier**, it cross-references the Reasoning Agent's specific hypotheses against grounded web searches, classifying them as verified or unverified to ensure the final reasoning trace is informed only factually correct information, and prevents unrealistic knowledge leakage.

**Reasoning Agent** ($\mathcal{A}_{reason}$). Acting as the core module, this agent generates the forecasting reasoning $R$. It operates in a two-stage process: first as an *Initial $\mathcal{A}_{reason}$* that formulates hypotheses based on historical data $X_{hist}$ and retrieved events, and subsequently as a *Final $\mathcal{A}_{reason}$* that integrates verification reports to produce refined, step-by-step reasoning. To ensure diversity, it generates multiple parallel reasoning candidates for a highly robust and rigorous evaluation of the models.

**Forecasting Agent** ($\mathcal{A}_{forecast}$). This agent maps the $R$ and $X_{hist}$ into a numerical forecast $\hat{Y} \in \mathbb{R}^{H \times C}$. To ensure the output is a reproducible derivative of the reasoning, the agent utilizes a deterministic greedy decoding strategy ($Temperature = 0.0$). It operationalizes linguistic directives (e.g., "decelerating growth" or "sharp 10-15 unit spike") by using $X_{hist}$ as a historic reference, ensuring that the generated numbers are consistent with historical units and provided reasoning. Additionally, we provide other historic time series features (e.g., trends, seasonality, residuals, etc.) and dataset description for further informed forecasting. Note that, $\mathcal{A}_{forecast}$ operates without access to the ground truth $Y$, maintaining a strict no data leakage setting, where the forecast is driven entirely by the reasoning trace.

**Verification Agent** ($\mathcal{A}_{verify}$). For checking the quality of generated reasoning, this agent evaluates the alignment between the generated $R$ and the resulting forecast $\hat{Y}$. It assigns a qualitative consistency score $S \in \{1, \ldots, 5\}$ and acts as a diagnostic module: if a forecast fails to beat the naive baseline, it analyzes the failure mode and generates actionable, textual feedback $F$ to guide $\mathcal{A}_{reason}$.

**Summary Agent** ($\mathcal{A}_{summary}$). To enhance user readability, this agent condenses the lengthy reasoning trace ($R$) into a concise contextual summary ($R_{sum}$), a concise prompt that retains all critical causal explanations.

*Table 2.* **Dataset Statistics.** *TFRBench* provides a multi-domain suite with diverse temporal frequencies (Hourly to Monthly) and dimensionalities, including univariate to multivariate series (2–5 channels) to test cross-channel reasoning. We utilize different forecasting horizons, mostly 96-96 window for most domains following prior works. (Ch. denotes number of channels). Reference Samples are the selected samples utilized for baseline evaluation.

| Domain | Dataset | Source | Total Samples | Reference Samples | Freq. | Window (in-out) | Ch. | Series |
|---|---|---|---|---|---|---|---|---|
| Energy | Solar | GIFT-Eval | 274 | 193 | Daily | 96-96 | 1 | 137 |
| | Electricity | GIFT-Eval | 1346 | 353 | Daily | 96-96 | 1 | 81 |
| Sales | Car-parts | GIFT-Eval | 1037 | 692 | Monthly | 25-25 | 1 | 1037 |
| | Hierarchical Sales | GIFT-Eval | 1370 | 1076 | Daily | 96-96 | 1 | 81 |
| Web/CloudOps | Bitbrains - Fast Storage | GIFT-Eval | 1153 | 678 | Hourly | 96-96 | 2 | 197 |
| | Web Traffic | Monash archive | 1214 | 710 | Daily | 96-96 | 1 | 162 |
| Transportation | Traffic | Autoformer et al. | 1104 | 449 | Hourly | 96-96 | 1 | 6 |
| | NYC Taxi | nyc.gov | 1000 | 391 | Hourly | 96-96 | 1 | 1 |
| Economics/Finance | Amazon pricing | Yahoo Finance/nasdaq | 512 | 328 | Daily | 14-7 | 5 | 1 |
| | Apple pricing | Yahoo Finance/nasdaq | 809 | 509 | Daily | 14-7 | 5 | 1 |

### 3.2.2. ITERATIVE GENERATION PROCESS

Our pipeline employs an iterative "generate, verify, and re-fine" cycle (Algorithm 1) involving five specialized agents. Initially, the $\mathcal{A}_{search}$ (Event Discovery) retrieves external events $\mathcal{E}$ (e.g., holidays) and verifies initial assumptions from the $\mathcal{A}_{reason}$ to prevent hallucinations, producing a verification report ($V$). Conditioned on this context, the system enters a refinement loop (up to $K$ times). In each iteration, $\mathcal{A}_{reason}$ generates parallel reasoning traces $\{R_i\}$, which the deterministic $\mathcal{A}_{forecast}$ utilize for numerical forecast predictions $\hat{Y}_i$. To determine the optimal candi-date, $\mathcal{A}_{verify}$ performs a joint evaluation: candidates are ranked based on a combined score that integrates quantita-tive accuracy (MASE) with a qualitative consistency score ($S \in \{1, ..., 5\}$) which measures the logical alignment be-tween the narrative plan and the numerical output. If the best candidate fails to beat the naive baseline ($MASE \geq 1.0$), $\mathcal{A}_{verify}$ provides actionable feedback ($F$) to guide the next generation. Finally, the $\mathcal{A}_{summary}$ distills the optimal trace into a concise $R_{sum}$. Further details are in App. A.2.

## 4. *TFRBench*: Benchmark Overview

To evaluate the generalization of forecasting systems across varying contexts, *TFRBench* incorporates a diverse collec-tion of datasets spanning five domains: Energy, Sales, Web/-CloudOps, Transportation, and Finance. As detailed in Ta-ble 2, the benchmark comprises ten distinct datasets sourced from established repositories. This structured diversity en-sures that *TFRBench* evaluates a model's ability to adapt its reasoning strategies to different domain-specific logic.

**Selection of Data Samples for *TFRBench*.** To ensure quality, we employ a strict outcome-conditioned filtering. A reasoning trace $R$ is included in the benchmark only if it satisfies: 1) achieving a high consistency score ($S \geq 4$)

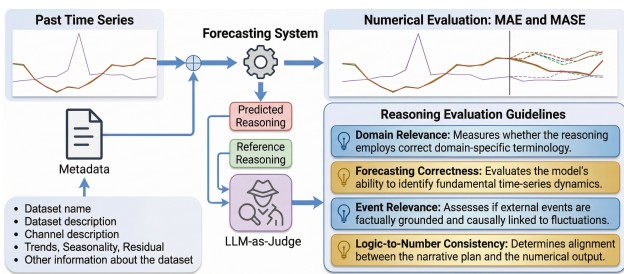

*Figure 3.* **Overview of the *TFRBench* Evaluation Pipeline.** While standard metrics track numerical error, the core focus of our framework is evaluating the *quality of the reasoning process*. We employ an LLM-as-a-Judge to audit a candidate's reasoning trace against our verified "Reference Reasoning".

from $\mathcal{A}_{verify}$, and 2) yielding a forecast that strictly outper-forms the naive seasonal baseline ($MASE < 1.0$). This ensures every reference sample represents logic validated by predictive utility rather than mere fluency. App. C provides quantitative validation of our framework–including success rates, usefulness of feedback, verifier reliability, and more.

**Seasonality and Baseline Selection.** To establish a qual-ity floor, we employ the **Seasonal Naive** method as the performance baseline for calculating the MASE score (Eq. 1 - App. A.2). This baseline assumes future values simply replicate the last observed cycle ($\hat{Y}_{t+h} = Y_{t+h-m}$), a heuris-tic that is difficult to outperform in high-entropy domains. Here, $m$ is a seasonality period. By requiring valid reason-ing traces to strictly beat this standard ($MASE < 1.0$), we ensure that the generated reasoning captures causal dy-namics rather than merely describing repetitive historical patterns. Following the implementation in GIFT-Eval (Aksu et al., 2024), we utilize the GluonTS framework (Alexan-drov et al., 2020) for detecting seasonality.

*Table 3.* **LLM-as-a-Judge Results (Mean$_{std}$).** Score Range: 1–5 (higher is better). Color intensity reflects performance (green: high, red: low). Dom.: Domain Relevance, Fcst.: Forecasting Correctness, Evt.: Event Relevance, Logic: Logic-to-Number Consistency.

| Dataset | Models | w/ Reasoning | | | | Event Forecast + Reasoning | | | |
|---|---|---|---|---|---|---|---|---|---|
| | | Dom. | Fcst. | Evt. | Logic | Dom. | Fcst. | Evt. | Logic |
| Solar Daily | Gemini-2.5-Pro | $3.026_{0.000}$ | $2.316_{0.000}$ | $2.036_{0.000}$ | $4.528_{0.000}$ | $3.803_{0.000}$ | $2.731_{0.000}$ | $2.207_{0.000}$ | $4.207_{0.000}$ |
| | Gemini-3-Pro | $4.699_{0.000}$ | $3.212_{0.000}$ | $2.508_{0.000}$ | $4.881_{0.000}$ | $4.119_{0.000}$ | $2.699_{0.000}$ | $2.368_{0.000}$ | $4.523_{0.000}$ |
| | Claude-Sonnet-4.5 | $4.922_{0.000}$ | $3.062_{0.000}$ | $2.891_{0.000}$ | $3.363_{0.000}$ | $4.554_{0.000}$ | $2.637_{0.000}$ | $2.472_{0.000}$ | $3.368_{0.000}$ |
| Electricity | Gemini-2.5-Pro | $2.318_{0.013}$ | $2.768_{0.020}$ | $1.720_{0.022}$ | $4.590_{0.015}$ | $4.688_{0.000}$ | $3.076_{0.000}$ | $3.442_{0.000}$ | $3.198_{0.000}$ |
| | Gemini-3-Pro | $4.167_{0.000}$ | $3.173_{0.000}$ | $2.178_{0.000}$ | $4.773_{0.000}$ | $4.892_{0.000}$ | $3.606_{0.000}$ | $4.147_{0.000}$ | $4.484_{0.000}$ |
| | Claude-Sonnet-4.5 | $3.810_{0.000}$ | $2.054_{0.000}$ | $2.093_{0.000}$ | $3.167_{0.000}$ | $4.609_{0.000}$ | $1.971_{0.001}$ | $2.885_{0.001}$ | $2.203_{0.001}$ |
| Car Parts | Gemini-2.5-Pro | $3.123_{0.000}$ | $2.658_{0.000}$ | $1.863_{0.000}$ | $4.181_{0.000}$ | $4.926_{0.000}$ | $3.464_{0.000}$ | $3.811_{0.000}$ | $4.923_{0.000}$ |
| | Gemini-3-Pro | $4.068_{0.000}$ | $2.900_{0.000}$ | $2.088_{0.000}$ | $4.945_{0.000}$ | $4.962_{0.000}$ | $3.012_{0.000}$ | $4.158_{0.000}$ | $4.883_{0.000}$ |
| | Claude-Sonnet-4.5 | $4.120_{0.000}$ | $2.675_{0.000}$ | $2.022_{0.000}$ | $3.384_{0.000}$ | $4.939_{0.001}$ | $3.402_{0.000}$ | $4.084_{0.000}$ | $4.402_{0.000}$ |
| Hierarchical Sales | Gemini-2.5-Pro | $2.637_{0.014}$ | $2.972_{0.007}$ | $1.635_{0.010}$ | $4.730_{0.015}$ | $2.519_{0.000}$ | $2.456_{0.000}$ | $2.230_{0.000}$ | $3.346_{0.000}$ |
| | Gemini-3-Pro | $3.701_{0.000}$ | $2.878_{0.000}$ | $1.967_{0.000}$ | $4.335_{0.000}$ | $2.476_{0.000}$ | $2.492_{0.000}$ | $2.205_{0.000}$ | $3.798_{0.000}$ |
| | Claude-Sonnet-4.5 | $3.757_{0.000}$ | $2.613_{0.000}$ | $2.046_{0.000}$ | $3.651_{0.000}$ | $2.198_{0.000}$ | $2.019_{0.000}$ | $1.947_{0.000}$ | $2.836_{0.000}$ |
| Bitbrains Fast Storage | Gemini-2.5-Pro | $3.807_{0.000}$ | $2.565_{0.000}$ | $2.944_{0.000}$ | $3.845_{0.000}$ | $3.802_{0.000}$ | $1.313_{0.003}$ | $2.652_{0.001}$ | $1.324_{0.000}$ |
| | Gemini-3-Pro | $4.531_{0.000}$ | $3.451_{0.000}$ | $3.249_{0.000}$ | $4.879_{0.000}$ | $4.746_{0.000}$ | $2.963_{0.000}$ | $2.704_{0.006}$ | $4.174_{0.002}$ |
| | Claude-Sonnet-4.5 | $4.714_{0.000}$ | $2.780_{0.000}$ | $3.427_{0.001}$ | $2.507_{0.000}$ | $3.360_{0.000}$ | $1.432_{0.000}$ | $2.083_{0.000}$ | $1.543_{0.000}$ |
| Web Traffic | Gemini-2.5-Pro | $3.585_{0.000}$ | $2.831_{0.000}$ | $2.039_{0.000}$ | $4.693_{0.000}$ | $4.576_{0.000}$ | $2.761_{0.000}$ | $3.249_{0.000}$ | $3.645_{0.000}$ |
| | Gemini-3-Pro | $4.034_{0.000}$ | $3.169_{0.000}$ | $2.179_{0.000}$ | $4.904_{0.000}$ | $4.634_{0.000}$ | $2.804_{0.000}$ | $3.435_{0.000}$ | $4.699_{0.000}$ |
| | Claude-Sonnet-4.5 | $3.928_{0.000}$ | $2.325_{0.000}$ | $2.120_{0.000}$ | $3.727_{0.000}$ | $4.276_{0.000}$ | $2.097_{0.000}$ | $2.669_{0.000}$ | $2.637_{0.001}$ |
| Traffic | Gemini-2.5-Pro | $2.762_{0.012}$ | $1.946_{0.004}$ | $1.896_{0.027}$ | $4.584_{0.034}$ | $3.643_{0.001}$ | $3.143_{0.000}$ | $2.203_{0.000}$ | $4.777_{0.000}$ |
| | Gemini-3-Pro | $3.793_{0.000}$ | $2.483_{0.000}$ | $2.249_{0.000}$ | $4.842_{0.000}$ | $2.604_{0.000}$ | $2.664_{0.000}$ | $2.022_{0.000}$ | $3.935_{0.000}$ |
| | Claude-Sonnet-4.5 | $4.294_{0.000}$ | $2.254_{0.000}$ | $2.488_{0.000}$ | $3.586_{0.000}$ | $3.381_{0.000}$ | $2.379_{0.000}$ | $1.962_{0.000}$ | $3.454_{0.000}$ |
| Nyc Taxi | Gemini-2.5-Pro | $2.814_{0.018}$ | $1.922_{0.010}$ | $1.703_{0.018}$ | $4.360_{0.040}$ | $4.972_{0.000}$ | $3.529_{0.001}$ | $3.425_{0.000}$ | $4.652_{0.000}$ |
| | Gemini-3-Pro | $3.969_{0.000}$ | $2.678_{0.000}$ | $2.113_{0.000}$ | $4.872_{0.000}$ | $4.941_{0.001}$ | $3.844_{0.000}$ | $3.941_{0.000}$ | $4.555_{0.000}$ |
| | Claude-Sonnet-4.5 | $4.532_{0.000}$ | $2.297_{0.000}$ | $2.363_{0.000}$ | $2.767_{0.000}$ | $4.890_{0.000}$ | $2.719_{0.000}$ | $3.156_{0.000}$ | $3.097_{0.000}$ |
| Amazon | Gemini-2.5-Pro | $2.262_{0.000}$ | $1.488_{0.000}$ | $1.637_{0.000}$ | $3.213_{0.000}$ | $4.966_{0.000}$ | $2.596_{0.001}$ | $3.617_{0.001}$ | $4.640_{0.002}$ |
| | Gemini-3-Pro | $2.936_{0.000}$ | $1.829_{0.000}$ | $1.784_{0.000}$ | $3.976_{0.000}$ | $4.997_{0.000}$ | $3.433_{0.000}$ | $4.125_{0.000}$ | $4.857_{0.000}$ |
| | Claude-Sonnet-4.5 | $3.588_{0.000}$ | $2.012_{0.000}$ | $1.893_{0.000}$ | $3.970_{0.000}$ | $4.988_{0.000}$ | $2.543_{0.000}$ | $3.179_{0.001}$ | $4.491_{0.000}$ |
| Apple | Gemini-2.5-Pro | $2.217_{0.001}$ | $1.557_{0.001}$ | $1.695_{0.000}$ | $3.380_{0.002}$ | $4.986_{0.000}$ | $2.556_{0.000}$ | $3.648_{0.000}$ | $4.642_{0.000}$ |
| | Gemini-3-Pro | $3.000_{0.000}$ | $1.842_{0.001}$ | $1.800_{0.000}$ | $3.821_{0.001}$ | $5.000_{0.000}$ | $3.327_{0.001}$ | $4.313_{0.001}$ | $4.826_{0.002}$ |
| | Claude-Sonnet-4.5 | $3.617_{0.000}$ | $2.075_{0.000}$ | $1.917_{0.000}$ | $3.786_{0.000}$ | $4.996_{0.000}$ | $2.515_{0.000}$ | $3.387_{0.000}$ | $4.642_{0.001}$ |

**Data Integrity and Leakage Prevention.** We explicitly distinguish the role of ground truth ($Y$) and future events/context across *Data Creation Phase* and *Evaluation Phase*. During data creation, we utilize oracle knowledge of future events to prioritize factuality over ex-ante simulation. We argue that excluding key future drivers (e.g., earnings, holidays) would force the reference reasoning to misattribute value shifts to noise; by incorporating them, *TFRBench* evaluates models against a grounded, event-aware reasoning. Crucially, $Y$ is withheld from generation, serving only as a post-hoc validation signal (via MASE). In contrast, during evaluation phase, models are restricted solely to $X_{hist}$, ensuring no ex-ante leakage. This design establishes the reference reasoning as a necessary "upper bound" for measuring alignment with the domain dynamics rather than just plausible guesses. In summary, oracle knowledge of future window-bounded events is to guarantee factual correctness.

**Evaluation Protocol.** To assess reasoning validity alongside numerical accuracy, *TFRBench* introduces a standardized evaluation protocol (Figure 3). We employ an LLM-as-a-Judge to compare candidate outputs against our generated reasoning trajectories (which we consider the best possible reasoning available for evaluation) across four dimensions: *Domain Relevance*, *Forecasting Correctness*, *Event Relevance*, and *Logic-to-Number Consistency*.

# 5. Experiments and Results

## 5.1. Experimental Setup

**Data Generation Framework.** We utilize Gemini-2.5-Pro as the backbone model for each agent (described in §3). The $\mathcal{A}_{search}$ is augmented with Gemini Grounded Search[1]. The generation process relied on a set of system prompts designed to enforce strict role adherence for each agent. These prompts are provided in App. B. Our decision to utilize Gemini-2.5-Pro is grounded in a preliminary ablation study (App. C.10) evaluating the trade-off between reasoning success rates and computational cost.

**Benchmarking Models.** We evaluate a diverse range of foundation models to establish baseline performance on *TFRBench*. This includes multiple Gemini family versions, such as Gemini-2.5-Flash, Gemini-2.5-Pro, and Gemini-3.0-Pro, along with Claude-Sonnet versions 4.0 and 4.5 to compare reasoning capabilities. We also include domain-specific forecasting models, specifically TimesFM-2.5 (Das et al., 2024) and Chronos-2.0 (Ansari et al., 2025), to provide a comparative numerical baseline against LLMs, supplemented by a statistical ARIMA baseline (Nelson, 1998;

---

[1] https://ai.google.dev/gemini-api/docs/google-search

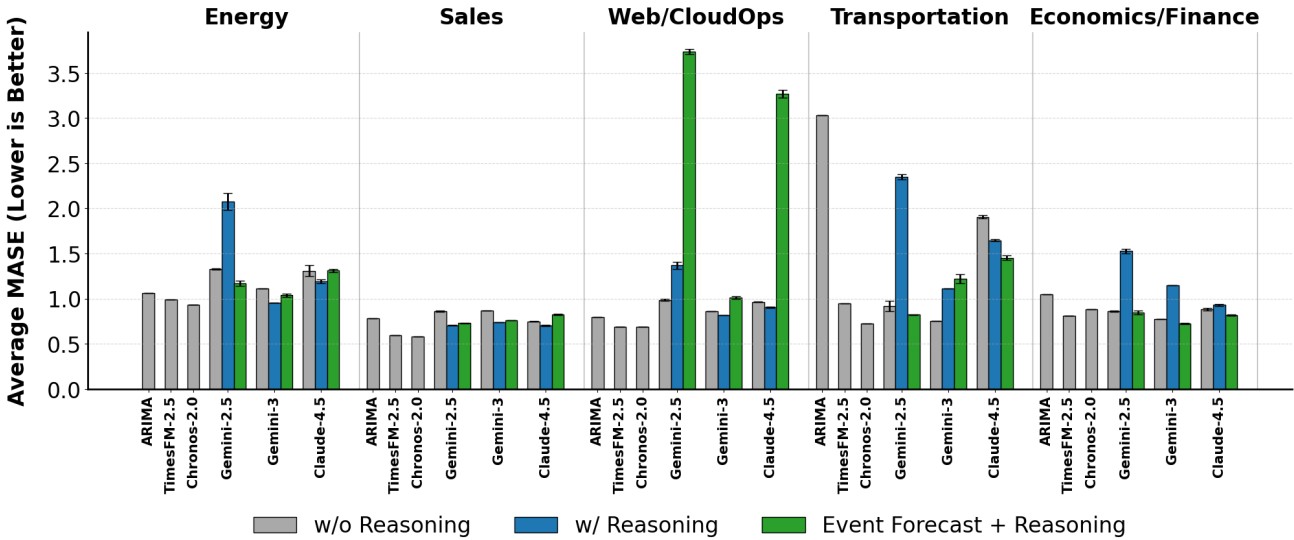

*Figure 4.* **Avg. MASE performance by domains.** We compare baseline models across five domains; lower MASE indicates better accuracy. Models: Gemini-2.5-Pro, Gemini-3-Pro, and Claude-Sonnet-4.5. Full numerical results are provided in App. I (Table 10).

Siami-Namini & Namin, 2018). Open-source models were also evaluated, but due to their lower competitiveness, results are provided as an ablation study in App. E. We exclude models requiring substantial training data or per-dataset training, such as Autoformer (Wu et al., 2021) or TimeXer (Wang et al., 2024), as our benchmark focuses on zero-shot forecasting without prior training.

**Experimental Configurations.** We evaluate models across three primary settings, with specific prompts and hyperparameters detailed in App. F and App. G.

- **Direct Numeric Forecast:** In this setting, models are prompted to output numerical predictions directly based on the historical context window, without generating any intermediate reasoning or explanation. This setting is represented as "w/o Reasoning" in Table 10.
- **CoT Forecasting:** In this setting, models are prompted to first generate a step-by-step reasoning. The model must analyze trends, seasonality, and cross-channel dependencies before producing the final numerical forecast. This setting evaluates the model's intrinsic ability to reason about time-series data without external tools, represented as "w/ Reasoning" in Tables 3 and App. I, Table 10.
- **Event Forecast + Reasoning:** We augment CoT (Wei et al., 2022) by retrieving historical context (e.g., holidays, news) and explicitly "predicting" events over the future horizon. The model generates reasoning incorporating these predictions to guide the final numerical forecast, a setting denoted as "Event Forecast + Reasoning".

**LLM-as-Judge Metric.** We assess reasoning quality using a Gemini-3-Pro as evaluator, scoring traces on a stan-

dardized Likert scale of 1-5 across four dimensions (see App. F and App. K). These four dimensions in Figure 3 were chosen to provide a comprehensive audit of the human-like cognitive loop required for reasoned decision-making in forecasting. By evaluating domain relevance, forecasting correctness, event relevance, and logic consistency, these metrics systematically capture reasoning failures, such as narrative bias or poor causal attribution.

### 5.2. Main Results

We present a comprehensive evaluation of reasoning and forecasting performance in Table 3, and Figure 4. Table 3 reports reasoning performance using LLM-as-Judge metrics. Figure 4 reports MASE across three distinct experimental settings (w/o Reasoning, w/ Reasoning, and Event Forecast + Reasoning) for the corresponding models. We report the mean and standard deviation (mean$_{std}$) across three runs for statistical significance. Default parameters are used for reasoning models, while the "w/ Reasoning" setting enforces an external CoT trace. Detailed results are in App. I.

**Model capabilities and reasoning quality.** Our analysis of Table 3 reveals that reasoning capability is not uniform. We observe that larger models like Gemini-3-Pro outperform smaller Gemini-2.5-Pro, particularly in complex tasks like *Solar* (avg. score 3.825 vs. 2.977). However, this advantage is domain-dependent. Models struggle significantly in stochastic domains like Finance compared to physical domains like Energy. This variance shows the risk of relying on "black box" LLM capabilities without the reasoning verification, *TFRBench* makes an effort to enable this.

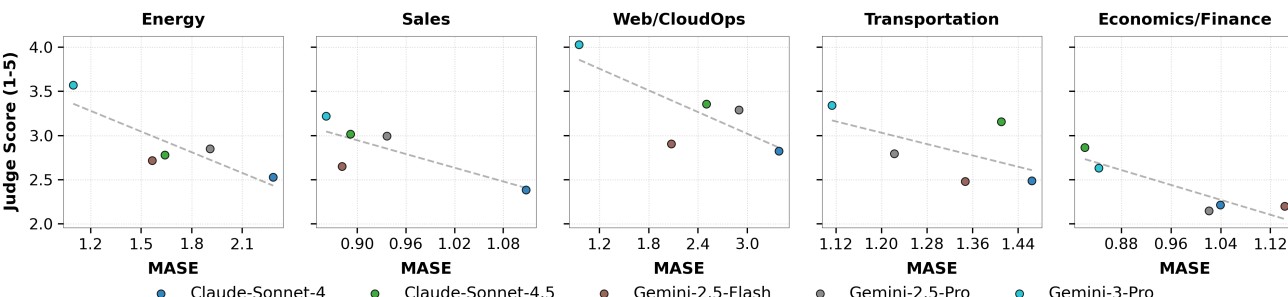

*Figure 5.* Reasoning Quality vs. MASE. Models with higher reasoning quality (Judge Score) achieve better forecasting accuracy (lower MASE). Conversely, lower reasoning scores are associated with higher forecasting errors.

**Forecasting performance and reasoning correlation.**
We quantify the relationship between reasoning quality (Table 3) and forecasting performance (Figure 4). As illustrated in Figure 5, our analysis reveals a strong correlation: models that achieve higher reasoning scores consistently yield lower MASE (i.e., higher forecasting performance). This finding is pivotal as it establishes that "reasoning quality" is a reliable driver of numerical performance. While the inclusion of previous-generation models highlights that general capability scaling improves both metrics simultaneously, the variance observed among state-of-the-art models reinforces our core thesis: as foundation models become increasingly competitive in pure numerical accuracy, evaluating how they arrive at their predictions becomes critical to differentiating true causal understanding from spurious correlation. Furthermore, the consistently low reasoning scores observed across off-the-shelf models indicates reasoning performance gap, underscoring the necessity of *TFRBench* to evaluate system for interpretable forecasting.

**Impact of reasoning on forecasting performance.**   We observe a dichotomy in the efficacy of CoT reasoning. In pattern-rich domains such as *Solar* and *Car Parts*, Gemini-3-Pro reduces MASE (w/o Reasoning → w/ Reasoning) by $\sim 20\%$ ($1.037 \rightarrow 0.834$) and $\sim 18\%$ ($0.821 \rightarrow 0.672$), respectively. Conversely, in stochastic domains like *Finance*, reasoning introduces "narrative bias,"–a phenomenon where models hallucinate causal narratives which degrade forecasting performance (e.g., Amazon Pricing performance reduced from 0.715 to 1.114; see App. L). This failure mode underscores the need for *TFRBench*'s verification loops.

**Impact of event forecasting on performance.**   Our experiments show that augmenting reasoning with external search for $X_{hist}$ and event forecasting for future horizon improves forecasting performance, particularly in domains sensitive to exogenous factors. As shown in Figure 4 and App. I (Table 10), accessing context for past data (e.g., weather, holidays) allows models to reason better, which is highly effective in *Energy* and *Transportation*. For instance, Gemini-2.5-Pro reduces MASE on Electricity from 2.853 (w/ Reasoning) to 1.327 (w/ Event Forecast) , while Claude-Sonnet-4.5 achieves similar gains in Traffic, improving from 1.458 to 1.285. However, this strategy is not always beneficial. In Web/CloudOps, event-aware reasoning acts as a distractor (Figure 4). Claude-Sonnet-4.5, for example, shows error increase from $0.938 \rightarrow 5.469$ on Bitbrains Fast Storage. Because high-entropy technical workloads like Bitbrains are driven by internal processes rather than external news, models succumb to narrative bias (App. L).

**TSFMs *vs.* Reasoning Models.**   TSFMs (e.g., TimesFM-2.5) set a high bar for accuracy, achieving a MASE of 0.732 on *Solar* (Figure 4). In contrast, zero-shot LLMs struggle; Gemini-3-Pro yields a MASE of 1.037. However, explicit CoT reasoning reduces this error to 0.834, narrowing the gap. Adding external context further improves performance: Gemini-2.5-Flash improves from 1.168 to 0.896 with event forecasting. While a narrow numerical delta remains, reasoning-enhanced models offer a superior value proposition: they deliver competitive accuracy with interpretability, effectively solving the "black box" problem inherent in traditional foundation models.

## 6. Ablation Study

**Quantitative Analysis.**   We evaluate reasoning and forecasting performance across 10 datasets along five dimensions: success rate, verifier calibration, stability, feedback necessity, and error correction (see App. C). We focus here on the *Overall Success Rate*, defined as outperforming the naive seasonality: $P(\text{Success}) = \frac{1}{N} \sum_{i=1}^{N} \mathbb{I}(\text{MASE}_i < 1.0)$. We compare *Direct Forecasting* (predicting future values directly from history) against *Reasoning-Guided Forecasting* (prompting with our verified reasoning traces). As shown in Figure 6, reasoning yields significant gains: success rates rise on *Solar* ($30.7\% \rightarrow 70.4\%$) and *Hierarchical Sales* ($54.7\% \rightarrow 78.5\%$). This confirms our reasoning traces provide an informative signal rather than noise. While TSFMs act as a numerical upper bound, our reasoning significantly narrows the performance gap while offering interpretability. This establishes our reasoning as the reference

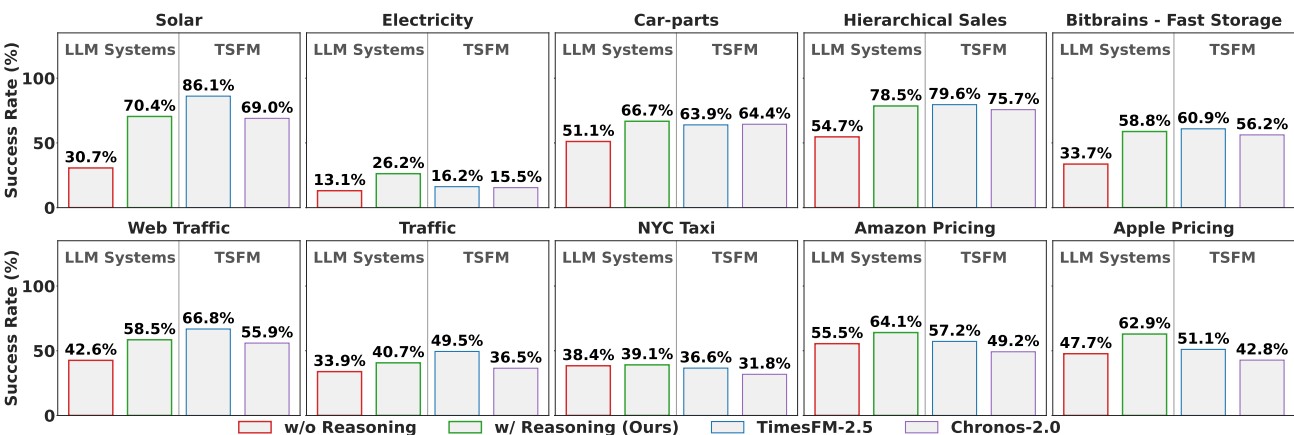

*Figure 6.* **Overall Success Rate.** Percentage of samples achieving MASE < 1.0 across datasets. The strategic reasoning (w/ Reasoning (Ours)) module consistently outperforms the direct prompting baseline. TSFM part represents Time Series Foundational models.

standard for evaluation, confirming that they capture the necessary causal logic to minimize MASE.

**Event Extraction Accuracy.** We evaluate retrieval via "Evt." scores (Table 3). Results reveal a dichotomy: search is effective for discrete entities in *Finance* (Gemini-3-Pro: *Apple* 4.313, *Amazon* 4.125) but degrades for continuous physical variables. In *Solar* and *Traffic*, scores drop to 2.368 and 2.022, respectively. This suggests search-augmented agents excel at structured textual events (e.g., earnings calls) but struggle with unstructured, high-frequency metadata (e.g., cloud opacity) (see App. K.1 and L).

**Robustness.** Additionally, we conduct a robustness analysis of the LLM-as-a-Judge (App. D), which demonstrates a high correlation between independent judge scores (Spearman correlation coefficient of 0.842) and strong predictive validity when compared against a panel of human experts (70.4% Pearson correlation). Furthermore, independent co-verification by Claude-4.5 confirmed our pipeline successfully mitigates confirmation bias by heavily penalizing flawed reasoning traces (App. D.3). We further justify our model choices through an ablation study (App. C.10) that compares performance across model generations. These combined findings support the objectivity of our evaluation framework and ensure the judging rubric remains independent of any specific model family or scaling strategy.

## 7. Conclusions

We introduced *TFRBench*, a novel benchmark that evaluates the reasoning capabilities of forecasting systems alongside their numerical accuracy. Our multi-agent framework synthesizes high-quality reasoning traces across ten datasets and five domains, demonstrating that structured reasoning allows Large Language Models to achieve performance

competitive with domain-specific foundation models. Practically, *TFRBench* serves as a diagnostic tool for identifying narrative biases and ensuring that automated forecasts are derived from factually grounded, causal logic rather than statistical artifacts. This standard enables the development of reliable and interpretable forecasting agents suitable for high-stakes decision-making. While we conducted an initial human expert evaluation to validate our benchmark, future work will focus on incorporating large-scale human studies across a broader range of domains to further validate the alignment between reasoning and domain expert intuition. We will also expand this study to more under-explored forecasting domains such as healthcare and nature.

## Impact Statement

This work advances machine learning by shifting time-series forecasting from opaque "black box" models to interpretable, reasoning-based systems. By requiring models to explain their predictions in high-stakes domains such as Energy and Finance, *TFRBench* aims to enhance human trust and accountability in automated decision-making. We confirm that all datasets included in *TFRBench* are sourced from established, publicly available repositories and standard forecasting benchmarks (e.g., Monash, GIFT-Eval). We have ensured proper attribution to the original data curators for all ten datasets spanning Energy, Sales, WebOps, Transportation, and Finance. No new private user data was collected or generated during the construction of this benchmark. In addition, this research addresses the ethical risk of "narrative bias," where models may generate persuasive but factually incorrect justifications. By enforcing rigorous verification protocols, our benchmark provides a necessary safeguard to detect hallucinations and ensure reliability. We hope this framework will inspire future innovations that prioritize transparent reasoning alongside numerical accuracy.

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

# Appendix

## A. Further Details on *TFRBench*

### A.1. Source Data Details

The specific datasets selected for *TFRBench* are detailed below:

**Solar:** Contains solar power production records sampled from PV plants in Alabama (Lai et al., 2018). This dataset captures the volatility associated with renewable energy generation, which is critical for grid management and stability.

**Electricity:** Contains the electricity consumption of individual clients, monitoring usage patterns over time (Lai et al., 2018). It is essential for understanding consumer behavior and load profiling in energy distribution networks.

**Car-parts:** Contains intermittent time series representing car parts sales (Godahewa et al., 2021). This dataset highlights the challenges of supply chain management, particularly for inventory items with sporadic or irregular demand patterns.

**Hierarchical Sales:** Gathered from a grocery store in Italy, which consists of time series representing the demand for pasta products (Mancuso et al., 2021). It serves as a standard benchmark for retail inventory planning and demand forecasting.

**Bitbrains - Fast Storage:** Consists of traces from VMs connected to fast Storage Area Network (SAN) devices (Shen et al., 2015). It includes metrics related to storage performance and usage, representing distinctive workloads found in modern cloud infrastructure and resource provisioning.

**Web Traffic:** Used in the Kaggle Wikipedia Web Traffic forecasting competition, this dataset tracks the number of hits for a set of Wikipedia pages (Godahewa et al., 2021). It is widely used to evaluate systems capable of handling erratic internet traffic and capacity planning.

**Traffic:** A collection of data from the California Department of Transportation describing road occupancy rates measured by sensors on San Francisco Bay Area freeways (Lai et al., 2018; Wu et al., 2021). This dataset is vital for analyzing urban traffic congestion and infrastructure usage.

**NYC Taxi:** Provides a time series of Yellow Taxi trip volumes in New York City (New York City Taxi and Limousine Commission, 2025). This dataset captures complex city-wide transportation demand dynamics and urban mobility trends.

**Amazon Pricing:** Provides a historical time series of Amazon.com, Inc. (AMZN) equity prices, spanning from the company's IPO. It captures market dynamics through Open, High, Low, Close, and Volume channels, serving as a proxy for market liquidity and investor interest (Yahoo! Finance, 2025).

**Apple Pricing:** Provides a historical time series of Apple Inc. (AAPL) equity prices, spanning from the company's IPO. Similar to the Amazon dataset, it captures long-term market valuation trends and daily price volatility through standard financial indicators (Yahoo! Finance, 2025).

### A.2. Iterative Generation Process

Unlike standard CoT (Wei et al., 2022) approaches that generate reasoning in a single pass, our pipeline employs a "generate, verify, and refine" cycle to minimize the MASE against ground truth. The process proceeds through six distinct steps:

**Step 1: External Contextualization.** Given the historical time-series $X_{hist}$, the $\mathcal{A}_{search}$ performs a targeted information retrieval to identify external events $\mathcal{E}$ relevant to the specific domain (e.g., "energy grid load" or "stock volatility"). This agent searches for significant $\mathcal{E}$, such as holidays, weather anomalies, or earnings reports, occurring within both the historical and future horizons. This step ensures that subsequent reasoning is conditioned on verified external metadata.

**Step 2: Hypothesis Formulation and Verification.** To prevent the propagation of false premises, we employ a strict verification method. First, the $\mathcal{A}_{reason}$ acts as an "Initial Reasoning Generator," formulating a set of explicit assumptions about trends and cross-channel dependencies based on $X_{hist}$ and $\mathcal{E}$. These assumptions are immediately passed back to the $\mathcal{A}_{search}$, which switches modes to act as an "Assumption Verifier." It validates each hypothesis against web-grounded sources, producing a **Verification Report** ($V$) that classifies each assumption as either "Verified" (supported by evidence) or "Unverified" (contradicted by evidence).

---

**Algorithm 1** Iterative Algorithm for *TFRBench*

---

**Input**    : History $X_{hist}$, Max Loops $K$
**Output**   : Best reasoning $R^*$, Summary $R_{sum}$, Best Forecast $\hat{Y}^*$
**Variables** : Feedback $F \leftarrow \emptyset$

**for** $k \leftarrow 1$ *to* $K$ **do**
    $\mathcal{E} \leftarrow \mathcal{A}_{search}(X_{hist})$ *// Step 1: External Contextualization*
    $H_{init} \leftarrow \mathcal{A}_{reason}(X_{hist}, \mathcal{E}, F)$
    $V \leftarrow \mathcal{A}_{search}.\texttt{Verify}(H_{init})$ *// Step 2: Hypothesis Formulation & Verification*
    $\{R_1, \dots, R_n\} \leftarrow \mathcal{A}_{reason}(X_{hist}, \mathcal{E}, V)$ *// Step 3: Parallel Candidate Generation*
    *// Step 4: Joint Evaluation & Scoring*
    **for** $i \leftarrow 1$ *to* $n$ **do**
        $\hat{Y}_i \leftarrow \mathcal{A}_{forecast}(R_i, X_{hist})$
        $S_i \leftarrow \mathcal{A}_{verify}.\texttt{Verify}(R_i, \hat{Y}_i)$
    **end**
    $R^*, \hat{Y}^* \leftarrow \texttt{SelectBest}(\{R_i, \hat{Y}_i, S_i\})$ *// Select the best candidate*
    *// Step 5: Feedback-Driven Refinement*
    **if** *Success Criteria Met* **then**
        **break** *// Success: Beat Naive Seasonality Baseline and highest Verifier Score*
    **end**
    $F \leftarrow \mathcal{A}_{verify}.\texttt{Diagnose}(R^*, \hat{Y}^*)$
**end**

$R_{sum} \leftarrow \mathcal{A}_{summary}.\texttt{Summarize}(R^*)$ *// Step 6: Distillation & Validation*

**return** $R^*, R_{sum}, \hat{Y}^*$

---

**Step 3: Parallel Candidate Generation.**    Conditioned on the $V$, the $\mathcal{A}_{reason}$ generates a refined reasoning. To capture diverse reasoning, the agent generates a set of $n$ parallel reasoning candidates $\{R_1, \dots, R_n\}$. Each candidate $R_i$ provides a comprehensive, step-by-step narrative that details specific strategies for decomposing the time series into trend, seasonality, and residual components, while explicitly citing the verified external events.

**Step 4: Deterministic Execution and Scoring.**    To evaluate the efficacy of the reasoning, the $\mathcal{A}_{forecast}$ maps each reasoning trace $R_i$ into a numerical forecast $\hat{Y}_i$. Crucially, this agent operates deterministically and without access to the ground truth, ensuring that the prediction is strictly derived from the reasoning logic. We then evaluate each candidate using a dual-scoring mechanism:

- **Quantitative Score:** We calculate the MASE for each candidate against the ground truth $Y$ to measure accuracy, where $H$ is the forecasting horizon, $T$ is the historical length, and $m$ is the seasonal period.

$$MASE_i = \frac{\frac{1}{H}\sum_{t=1}^{H}|Y_t - \hat{Y}_{i,t}|}{\frac{1}{T-m}\sum_{j=m+1}^{T}|X_j - X_{j-m}|} \tag{1}$$

- **Qualitative Score:** Simultaneously, the $\mathcal{A}_{verify}$ assigns a consistency score $S_i \in \{1, \dots, 5\}$, checking if the numerical output aligns logically with the narrative plan.

**Step 5: Feedback-Driven Refinement.**    If the best candidate $R^*$ fails to outperform the naive seasonality baseline (i.e., $MASE \geq 1.0$), a feedback loop is triggered. The $\mathcal{A}_{verify}$ acts as a diagnostic module, analyzing the discrepancy between the reasoning logic and the numerical error to generate actionable textual feedback $F$. This feedback is injected back into the $\mathcal{A}_{reason}$ for the next iteration ($k$), explicitly instructing it to adjust specific assumptions or trend estimates. This cycle repeats for a maximum of $K = 3$ loops to enforce convergence.

**Step 6: Distillation and Validation.**    Upon successful convergence (or reaching the loop limit), the $\mathcal{A}_{summary}$ compresses the final, complex reasoning trace $R^*$ into a concise, high-quality summary $R_{sum}$. To ensure no critical information was lost during compression, the system performs a final validation check: it generates a new forecast solely from $R_{sum}$ and retains the summary only if it maintains high forecasting accuracy.

# B. Prompts for Data Generation

---

**Prompt: Search Agent**

You are a specialized Search Agent. Your sole mission is to find relevant, time-specific external events that could impact a time-series forecast. Your task is to find events in BOTH the recent past and the future window.
**Core Search Parameters:**
Dataset Name: {dataset_name}
Recent Past Window: {past_start} to {past_end}
Future Window: {future_start} to {future_end}
Location Hint: {location_hint}
**Search Strategy:**
1. Time-Bound: Focus only on events within the specified windows.
2. Impactful Topics: Search for objective, external factors like public holidays, major weather events (heatwaves, storms), sporting events, conferences, or economic announcements.
3. No Impact Analysis: Do NOT analyze the 'potential impact'. Just state the event and its date.
4. Prioritize: Return only the top {search_events} most significant, time-specific events. Do not return a long, unprioritized list.
**Output Format Requirements:**
IMPORTANT: Your entire response must be a single, concise, numbered list of events.
- Do NOT add any preamble (e.g., 'Here are the events:').
- Format: 1. [Event Name] ([Date or Date Range])
- If no events are found, return the single word: None
Provide your findings directly.

---

**Prompt: Initial Reasoning Generator**

You are a master forecaster. Your task is to generate a step-by-step reasoning plan for a downstream forecasting AI. You must analyze the data and events, make predictions, and clearly state the assumptions behind your reasoning.
**Guidelines:**
1. State Assumptions First: Your first step for each channel MUST be to state your key assumptions about its trend, seasonality, event impacts, and relationship to other channels. The assumptions MUST be related to the events and their impacts.
2. Cross-Channel Analysis: You MUST include a specific analysis of cross-channel dependencies.
3. Qualitative & Flexible Reasoning: Your plan must be qualitative. Do NOT provide exact equations or simple linear projections. Describe the behavior (e.g., 'The trend should continue its recent upward path, but at a decelerating rate').
4. Qualitative Event Impacts: Describe the directional impact of events qualitatively (e.g., 'The heatwave will exert strong upward pressure on the residuals').
5. No Example Calculations: Do NOT include 'Example Forecast' calculations or equations.
6. Integrate Events: You MUST integrate the 'External Events' where relevant. Cite them using '(Source: Google Search)'.
7. You MUST generate reasoning for each channel.
**Full Data Context:**
{data_context_string}
**External Events Found by Search:**
{external_events}
**Feedback From Previous Loop:**
{feedback_prompt}
**Output Format Requirements:**
IMPORTANT: Your entire response must start immediately with 'Forecasting Reasoning:' and follow this exact step-by-step format for each channel:
Forecasting Reasoning:

[Channel 1 Name]: - Assumptions: [Your explicit assumptions for this channel.] - Cross-Channel Analysis: [Your analysis of how this channel relates to others.] - Trend Component Plan: [Your qualitative plan for the trend.] - Seasonality Component Plan: [Your qualitative plan for seasonality.] - Residuals & Event Plan: [Your qualitative plan for residuals and how events will impact them.]

[Channel 2 Name]: - Assumptions: [...] - Cross-Channel Analysis: [...] - Trend Component Plan: [...] - Seasonality Component Plan: [...] - Residuals & Event Plan: [...]

...etc. for all channels

---

### Prompt: Search Verifier

You are an expert Verifier. You will be given a 'Forecasting Reasoning' plan that contains several assumptions. Your sole task is to use Google Search to verify if these assumptions are factually right or wrong.

**Your Task:**

1. Read the 'Initial Reasoning' and identify every statement explicitly listed under an '- Assumptions:' bullet point for each channel.

2. For each assumption, use your search tool to find objective, factual evidence (news articles, official reports, statistics) that either confirms (RIGHT) or denies (WRONG) the assumption.

3. Report your findings as a simple, concise list. Do NOT add any preamble.

**Initial Reasoning to Verify:**

{initial_reasoning}

**Output Format Requirements:**

IMPORTANT: Your entire response must be a numbered list.

- Do NOT add any preamble (e.g., 'Here is the verification:').

- Format: 1. [Quote or summary of the assumption] - VERDICT: RIGHT/WRONG (Reason: [Briefly state the evidence you found, e.g., 'News reports confirm...']

- If you cannot find any information, state VERDICT: UNVERIFIABLE.

Begin your verification report now.

---

### Prompt: Final Reasoning Generator

You are a Master Forecaster. You must create the FINAL, executable reasoning plan. You will synthesize an 'Initial Reasoning' with a 'Verification Report' that has fact-checked its assumptions.

**Your Task:**

1. Review all inputs: Read the 'Initial Reasoning', the 'Verification Report', and the original 'External Events' and 'Data Context'.

2. Incorporate Verification: This is your most important job. - If the 'Verification Report' said an assumption was RIGHT, keep that part of the reasoning. - If the 'Verification Report' said an assumption was WRONG, you MUST CORRECT the reasoning plan. - CRITICAL: Do NOT mention the original wrong assumption or the verification process (e.g., 'this was wrong' is forbidden). Just provide the single, corrected, final plan as if it were correct all along.

3. Provide a Qualitative, Directive Plan: The final plan must be a concrete, step-by-step, channel-specific plan. - It must be qualitative. Do NOT use exact equations or math. - You MUST include numeric values (e.g., ranges/directions) in a flexible way to make directions concrete (e.g., "temporary upward shift of roughly 10-15 units").

4. Include Cross-Channel Analysis: Integrate the cross-channel analysis within each channel's reasoning block.

5. Omit Assumptions: Do NOT include an 'Assumptions' section in your final output.

6. No Example Calculations: Do NOT include 'Example Forecast' calculations or equations.

7. Make sure your output reasoning is not contradictory at all.

**Input Data:**

Full Data Context: {data_context_string}

External Events Found by Search (Search 1): {external_events}

Initial Reasoning (Reasoning 1): {initial_reasoning}

Assumption Verification Report (Search 2): {verification_report}

**Output Format Requirements:**

IMPORTANT: Your entire response must start immediately with 'Final Reasoning:' and follow this exact format:
Final Reasoning:
[Channel 1 Name]: - Cross-Channel Analysis: [Your analysis of how this channel relates to others, corrected by verifications.] - Trend Component Plan: [Your qualitative plan for the trend, corrected by verifications.] - Seasonality Component Plan: [Your qualitative plan for seasonality, corrected by verifications.] - Residuals & Event Plan: [Your qualitative plan for residuals and how verified events will impact them.]
[Channel 2 Name]: - Cross-Channel Analysis: [...] - Trend Component Plan: [...] - Seasonality Component Plan: [...] - Residuals & Event Plan: [...]
...etc. for all channels

---

## Prompt: Forecast Verifier

You are an expert Time-Series Forecasting Analyst. Your task is to score a 'Final Reasoning' based on the 'Forecast' it produced. You do not have the ground truth. You are judging the quality and coherence of the reasoning, its resulting forecast, and its numerical error (MASE) relative to the naive seasonal benchmark.
**Success Metric:** A MASE score $< 1.0$ is GOOD (it beats the naive benchmark). A MASE score $>= 1.0$ is BAD.
**Input Data:**
LLM Result (T=0.0) MASE: {mase_t0_str}
Final Reasoning: {final_reasoning}
Resulting Forecast (T=0.0): {forecast_str_t0}
Historical Context (for comparison): {data_context_string}
**Scoring Criteria (1-5 Scale):**
5 (Excellent): Reasoning is specific and well-verified. Forecasts clearly match reasoning. The LLM's MASE is excellent (e.g., $< 0.5$).
4 (Good): Reasoning is good. Forecasts generally follow reasoning. The LLM's MASE is good (e.g., 0.5 - 1.0).
3 (Average): Forecasts or reasoning are generic/don't perfectly match. The LLM's MASE is poor (e.g., 1.0 - 1.5), but the forecast isn't nonsensical.
2 (Poor): Reasoning is vague, forecasts don't match. The LLM's MASE is very poor (e.g., 1.5 - 2.0).
1 (Very Poor): Reasoning is illogical. Forecasts are nonsensical. The LLM's MASE is extremely high (e.g., $> 2.0$).
**Output Format Requirements:**
IMPORTANT: Your entire response must be a single, valid JSON object.
- Do NOT add any preamble.
- Strictly adhere to this exact format:
{"score": <your 1-5 score>, "feedback": " < Your concise feedback > "}

---

## Prompt: Feedback Generation Agent

You are an expert diagnostic agent. A forecasting attempt has failed (MASE was $>= 1.0$). Your task is to **generate actionable feedback** for the *next* loop. **You must not ask for the ground-truth data.** Analyze all the provided inputs to hypothesize *why* the forecast failed and what to do differently.
**Input Data:**
Failed Attempt's MASE (T=0.0): {mase_t0_str}
Verifier's Quality Score (1-5): {verifier_score}
Verifier's Feedback: {verifier_feedback}
Failed Attempt's Reasoning: {final_reasoning}
Failed Attempt's Forecast (T=0.0): {forecast_str_t0}
Full Data Context: {data_context_string}
**Your Diagnostic Process:**
1. **Analyze Verifier:** The 1-5 score and feedback are primary clues. If the score is low (1-2), the *reasoning itself* was likely flawed (vague, illogical, missed events, bad assumptions). Focus feedback on fixing the reasoning logic.
2. **Analyze Reasoning vs. Forecasts:** Did the forecast actually follow the reasoning? If the reasoning looks sound but the forecast doesn't match, the *forecasting AI* might have misinterpreted. Feedback: 'The reasoning seemed reasonable, but the resulting forecasts didn't numerically match its directives. Try again, following the reasoning

more literally.'

3. **Analyze Reasoning vs. History:** Does the reasoning make sense given the historical context? Did it suggest something contradictory without justification? Critique this logic.

4. **Formulate Recommendation:** Based on your analysis, provide a single, clear, forward-looking instruction for the *reasoning generation* in the next attempt (e.g., 'The assumption verification step may have missed a key point. Re-evaluate the impact of the heatwave on the OT channel. The initial assumption was wrong, but the correction was also not effective. Try a more moderate adjustment.'). Be specific.

**Output Format Requirements:**

IMPORTANT: Your entire response must be a single block of text.

- Do NOT add any preamble.

- Provide only the actionable recommendation for the next loop.

Actionable Recommendation: [Your concise feedback here]

---

**Prompt: Summary Generation Agent**

You are an expert time series forecasting reasoning summarization agent. Your sole task is to create a concise, effective summary of the provided 'Reasoning Plan'. This summary will be the *only* instruction given to a downstream forecasting AI.

**CRITICAL GOAL:** The summary must contain all *essential* information for the AI to replicate the forecast:

1. **Keep Key Numbers:** Retain specific numerical guidance (e.g., "upward shift of 10-15 units", "decelerating trend").

2. **Keep Key Events:** Retain all event impacts (e.g., "heatwave causes upward pressure").

3. **Keep Key Behaviors:** Capture the core trend and seasonality directives.

Do NOT be *too* concise. It is better to be slightly longer and retain all forecasting instructions than to be short and vague.

**Feedback on Previous Summary (if applicable):**

Your last summary produced a poor forecast. Here is the feedback:

{feedback}

Refine your summary based on this feedback.

**Reasoning Plan to Summarize:**

{reasoning_or_summary}

**Output Instructions:**

Provide your new, concise, *actionable* summary now. Do not add any preamble.

---

**Prompt: Summary Feedback Agent**

You are an expert diagnostic agent. A forecast generated *only* from the summary below has failed.

**The Goal is MASE < 1.0. This summary produced MASE: {mase}**

Your task is to provide feedback to the *summary agent* on how to improve the summary. You must **not** ask for the ground truth. Hypothesize why the summary was insufficient.

**Diagnostic Questions:**

- Was it too vague? Did it lose key numerical guidance from the original reasoning?

- Did it omit an important event impact?

- Did it fail to describe the trend or seasonality with enough detail?

**Failed Summary:**

{summary}

**Output Instructions:**

Provide a single, actionable recommendation for the summary agent.

Example: "The summary lost the specific numerical range for the trend. Re-add the guidance about the '10-15 unit' increase."

Actionable Recommendation: [Your concise feedback here]

# C. Additional Quantitative Analysis

## C.1. Candidate Stability

To assess the robustness of the generation process, we analyze the distribution of MASE scores across all generated candidates. We aim to determine if the model consistently produces high-quality hypotheses or relies on high-variance "lucky" outliers. Figure 7 presents violin plots of the candidate distributions. The probability density mass is concentrated around low MASE values, indicating that the reasoning engine generates stable, plausible forecasts rather than random hallucinations.

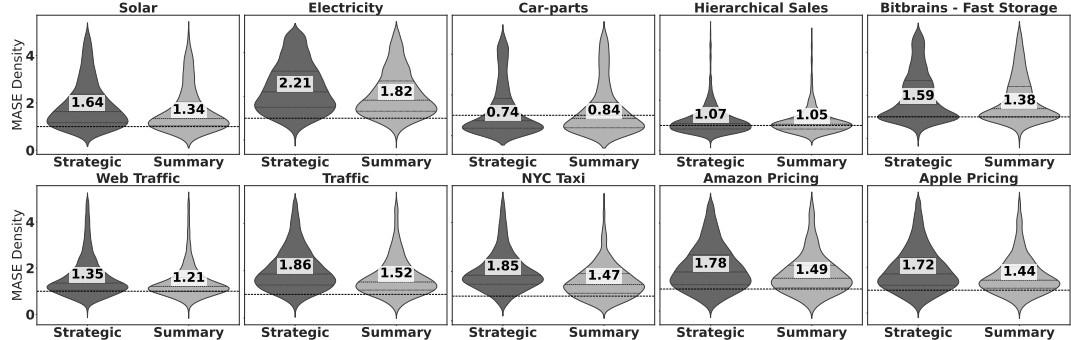

*Figure 7.* **Candidate Stability.** Distribution of MASE scores across generated reasoning candidates. The concentration of density around the naive baseline (MASE=1) indicates generation stability.

## C.2. Feedback Improvement Trajectory

To further isolate the effect of iterative refinement, we analyze the "Success Cohort", defined as the subset of samples that demonstrated improvement from Loop 1 to Loop 3. Figure 8 plots the mean MASE with standard deviation markers for this cohort at each iteration step. We observe a monotonic decrease in error:

$$\mu_{\text{MASE}}(L_1) > \mu_{\text{MASE}}(L_2) > \mu_{\text{MASE}}(L_3) \tag{2}$$

This trajectory confirms that the improvements are not random fluctuations but the result of systematic error correction, where the model incorporates verifier feedback to progressively converge on a more accurate forecast.

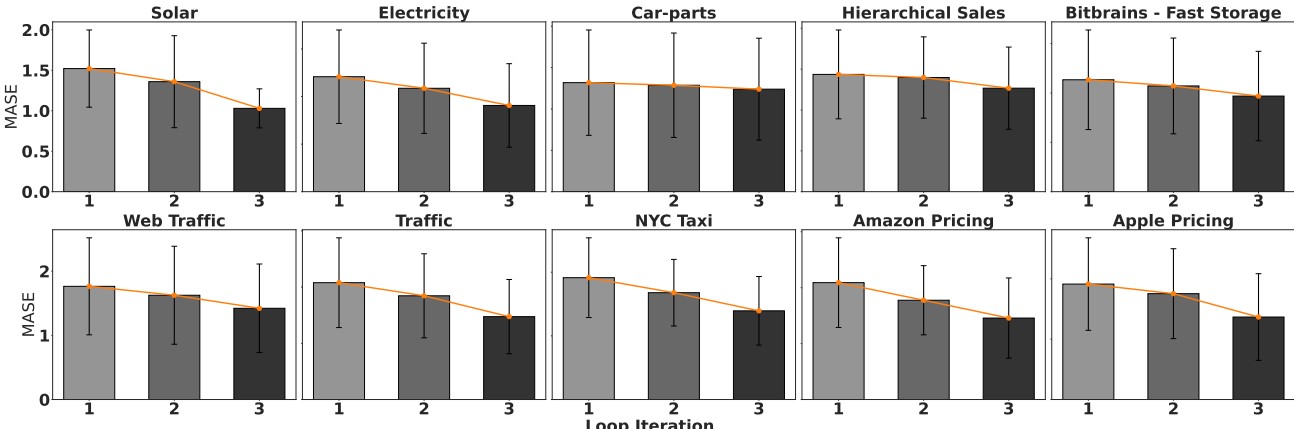

*Figure 8.* **Feedback Improvement Trajectory.** Mean MASE (with standard deviation) for the cohort of samples that improved over three iterations. The trend line confirms systematic convergence toward lower error.

## C.3. Feedback Rescue Rate

Finally, we evaluate the system's "Rescue Rate", its ability to correct initially failed forecasts. We identify the set of samples where the initial forecast failed to beat the baseline ($\text{MASE}_{init} \geq 1.0$) and calculate the proportion that were successfully refined to $\text{MASE}_{final} < 1.0$ through feedback.

$$\text{Rescue Rate} = \frac{|\{x \mid \text{MASE}_{init} \geq 1.0 \wedge \text{MASE}_{final} < 1.0\}|}{|\{x \mid \text{MASE}_{init} \geq 1.0\}|} \tag{3}$$

As shown in Figure 9, the rescue rate ranges from $\sim 6.5\%$ to $\sim 52.4\%$ across datasets. We provided short forms of the datasets for better visibility. This highlights the critical role of the feedback loop in salvaging difficult forecasting scenarios that would otherwise result in failure.

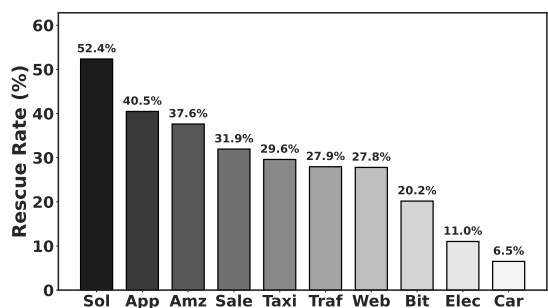

*Figure 9.* **Feedback Rescue Rate.** Percentage of initially poor forecasts (MASE $\geq 1.0$) successfully corrected to MASE $<$ 1.0 via feedback loop. For representation, dataset names are displayed in short form (e.g., Solar (Sol)).

## C.4. Feedback Loop Distribution

We analyze the computational cost and necessity of iterative reasoning by examining the distribution of feedback loops required per sample. As shown in Figure 10, while some samples are resolved in a single pass, a significant portion requires multiple iterations ($k > 1$). This non-uniform distribution justifies the adaptive nature of our framework; complex forecasting scenarios trigger deeper reasoning chains, whereas simpler patterns are resolved efficiently.

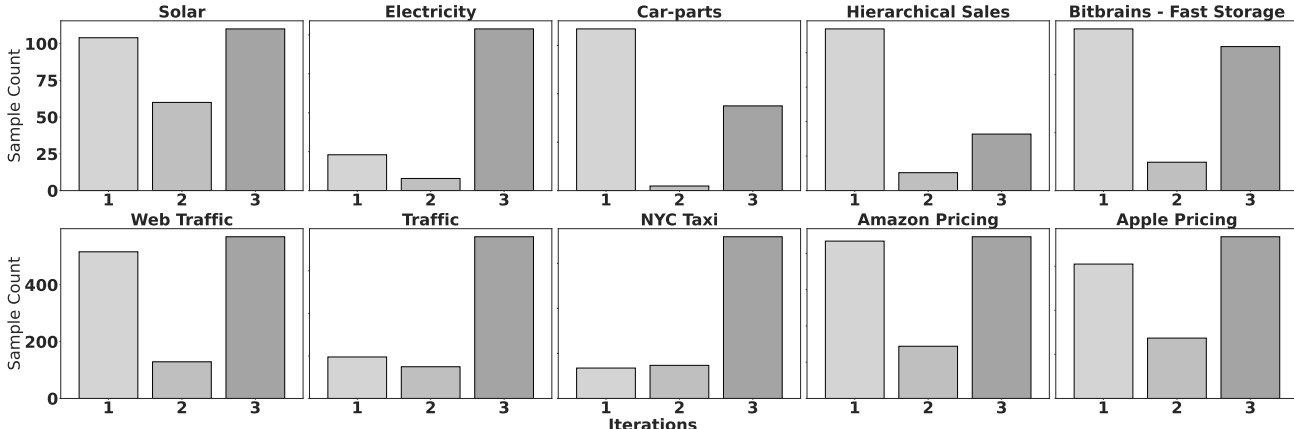

*Figure 10.* **Feedback Loop Distribution.** Histogram of iterations required. The variance in loop counts highlights the adaptive complexity of the reasoning process.

## C.5. Impact of Feedback Loop

We measure the marginal utility of the feedback mechanism by comparing the forecast error at the initial loop ($L_1$) versus the final loop ($L_{final}$). Figure 11 visualizes this transition. Points below the diagonal identity line ($y = x$) represent samples where performance improved through feedback:

$$\Delta\text{MASE} = \text{MASE}_{final} - \text{MASE}_{initial} < 0 \tag{4}$$

The density of samples in the lower-right triangle demonstrates that the feedback loop acts as an effective refinement operator, reducing error in the majority of multi-step cases.

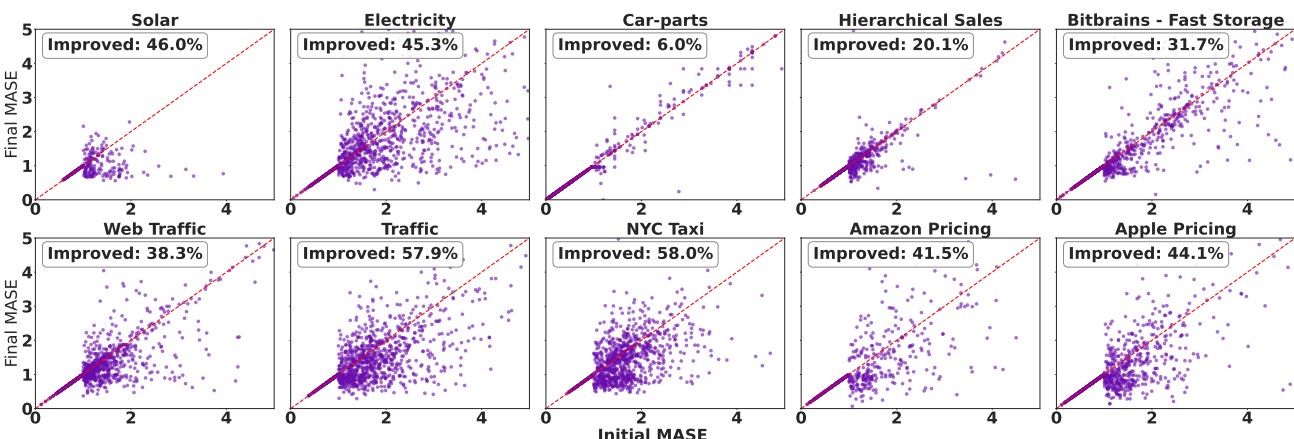

*Figure 11.* **Impact of Feedback.** Scatter plot of Initial vs. Final MASE. Points below the red dashed line indicate improved accuracy after feedback iterations.

## C.6. Verifier Reliability and Calibration

A critical component of our architecture is the internal Verifier, which assigns a confidence score $S \in \{1, \ldots, 5\}$ to generated forecasts. To validate calibration, we analyze the correlation between the Verifier Score and the MASE. As shown in Figure 12, we observe a strong negative correlation. Higher verifier scores consistently correspond to lower MASE values (higher accuracy). This confirms that the self-evaluation mechanism is well-calibrated and serves as a reliable proxy for forecast quality during inference.

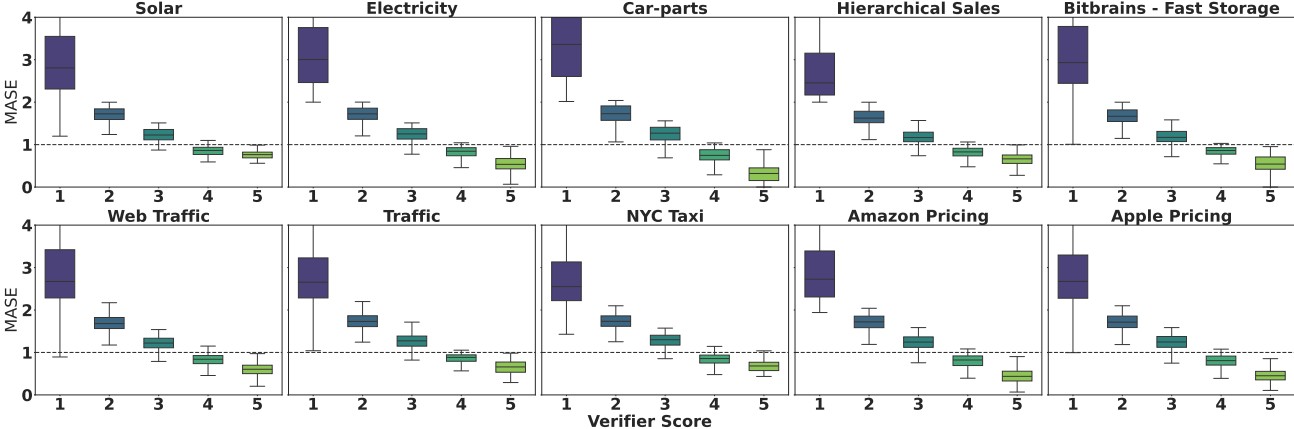

*Figure 12.* **Verifier Reliability.** Box plots showing the distribution of MASE for each assigned Verifier Score (1-5). The inverse relationship confirms that the model accurately self-assesses high-quality forecasts.

Furthermore, to ensure verification reliability, we stress-tested 50 reference samples (original score: 4.22/5.0) with targeted perturbations like "Domain Context Swaps" and "Fabricated Events." Our Verifier successfully caught and penalized these flawed traces, dropping the average score to 1.34/5.0 and rejecting 94.0% of the manipulated samples. This validates that our framework acts as a strict safeguard against incorrect reasoning.

## C.7. Sensitivity Analysis on Temperature for Forecasting

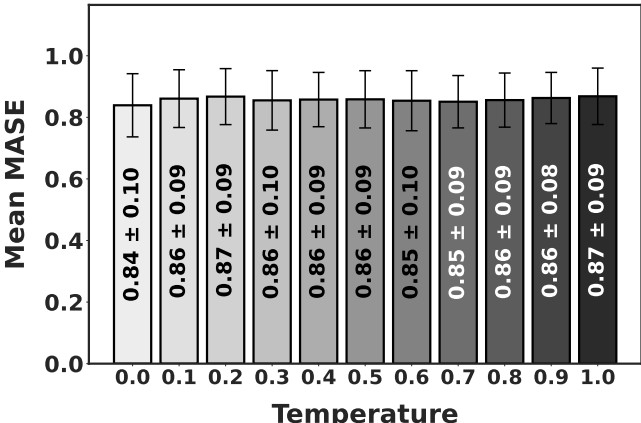

*Figure 13.* **Performance vs. Temperature.** Stable MASE across temperatures; optimal accuracy (lowest MASE) is achieved at Temperature = 0.

We investigate the impact of the sampling temperature parameter on model performance, as this hyperparameter directly controls the stochasticity of the generation process of forecasting. We evaluated the model across a range of temperatures on the Solar dataset, filtering for successful cases (MASE $< 1.0$).

As shown in Figure 13, the model exhibits remarkable stability across the swept temperature range, indicating that the reasoning capabilities are robust to variations in sampling randomness. Notably, the best performance is achieved with a deterministic greedy decoding strategy (Temperature = 0), which yields the lowest mean MASE. This suggests that for this forecasting task, minimizing stochasticity allows the model to most effectively leverage its reasoning chains without introducing unnecessary noise.

## C.8. Number of Feedback Loops

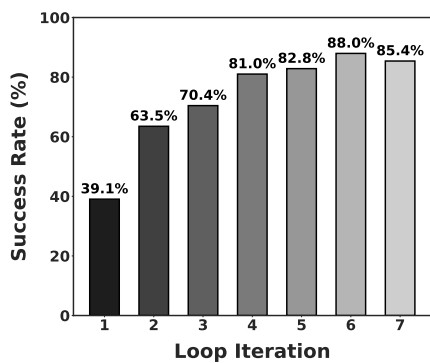

*Figure 14.* **Feedback Iterations.** Success rate ($MASE < 1.0$) on the Solar dataset. Global performance increases from 39.1% (Loop 1) to a peak of 88.0% (Loop 6), validating that the feedback mechanism systematically corrects reasoning errors rather than relying on random stochasticity. Each loop represents independent run starting from scratch.

To validate that our iterative refinement process is performing systematic error correction rather than random stochastic sampling, we analyze the cumulative success rate across 7 feedback iterations on the Solar dataset. As illustrated in Figure 14, the system exhibits a strong improvement in performance, rising from a baseline success rate of 39.1% in Loop 1 to a peak of 88.0% in Loop 6. This strictly upward trajectory refutes the hypothesis that the model is merely "guessing" randomly; if the improvements were due to stochastic variance, we would expect the success rate to oscillate or plateau early. Instead, the consistent gains confirm that the $\mathcal{A}_{verify}$ is successfully diagnosing failure modes and guiding the $\mathcal{A}_{reason}$ toward convergence.

We select Loop=3 as the optimal stopping criteria for our benchmark generation. At Loop 3, the system achieves a robust 70.4% success rate, nearly doubling the zero-shot performance of Loop 1 (39.1%). While extending the process to Loop

6 yields further gains (reaching 88.0%), the marginal utility of these additional iterations diminishes (e.g., the gain from Loop 4 to 5 is only 1.8%). Therefore, Loop=3 represents the most efficient trade-off point, balancing high data quality with computational resource constraints.

## C.9. Reasoning Length

To understand the cognitive load required for effective time-series reasoning, we analyze the distribution of reasoning trace lengths for "reference" samples where the model achieves a Mean Absolute Scaled Error (MASE) below 1.0. The histograms reveal distinct "complexity clusters" across the benchmark. Standard forecasting tasks, such as Solar, Electricity, and Traffic, exhibit a unimodal, bell-shaped distribution centered between 400 and 600 tokens, suggesting a standardized reasoning template is sufficient for these domains.

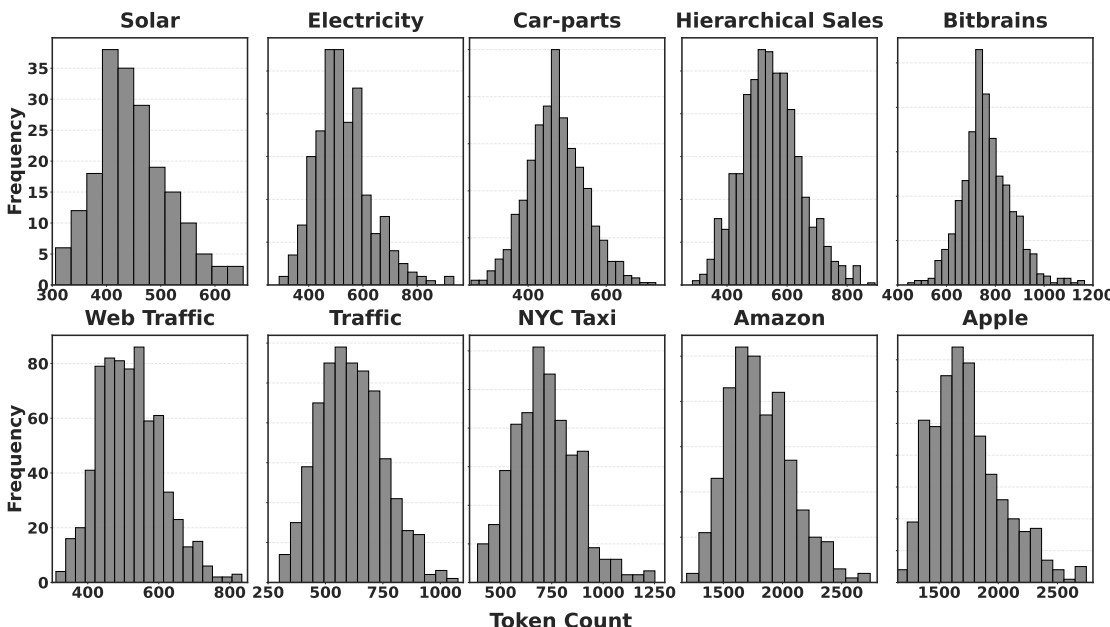

*Figure 15.* Histograms displaying the token count frequency for successful reasoning traces ($MASE < 1.0$) across datasets.

In contrast, the Amazon and Apple Pricing datasets display a significant rightward shift, with distributions centering around 1,600–1,800 tokens. This indicates that pricing tasks impose a substantially higher cognitive load, likely necessitating deeper causal analysis, arithmetic verification, or competitive comparison that simpler datasets do not. Notably, the distributions are consistently Gaussian across all domains, implying that the model converges on a stable, domain-specific depth of reasoning rather than varying its effort randomly.

## C.10. Backbone Model Selection and Scaling

We conducted an ablation study to compare the performance of different Gemini family models during the reasoning generation process. Specifically, we evaluated Gemini-2.5-Pro against Gemini-3-Pro on a subset of 100 samples from the Electricity dataset to measure the success rate in achieving $MASE < 1.0$. The results indicated only a marginal performance gain, with Gemini-3-Pro achieving a success rate of 26% compared to 21% for Gemini-2.5-Pro. Given this minor 5% improvement and the significantly higher computational overhead associated with frontier models, we selected Gemini-2.5-Pro as the primary backbone for scaling our benchmark. This decision was driven by its superior cost efficiency and competitive state-of-the-art performance, ensuring the systematic generation of high-quality reasoning traces across all ten datasets.

Furthermore, to validate potential generation bias, where Gemini-family models might score higher simply due to stylistic alignment with the reference traces, we emphasize that the choice of backbone impacts the generation yield rate rather than the final benchmark quality. Because all reference traces are strictly gated by objective predictive utility ($MASE < 1.0$),

*Table 4.* Candidate Discard Rates (Across All Iterative Runs)

| Dataset | Total Candidates Generated | Discarded by ($MASE < 1.0$ & $S \geq 4$) | Discard Rate |
|---|---|---|---|
| Solar | 2,770 | 2,404 | 86.8% |
| Electricity | 17,500 | 16,621 | 95.0% |
| Car-parts | 8,780 | 5,582 | 63.6% |
| Hierarchical Sales | 10,650 | 6,529 | 61.3% |
| Bitbrains Fast Storage | 11,225 | 9,090 | 81.0% |
| Web Traffic | 12,405 | 9,938 | 80.1% |
| Traffic | 13,865 | 12,931 | 93.3% |
| NYC Taxi | 12,915 | 12,263 | 95.0% |
| Amazon Pricing | 5,150 | 4,354 | 84.5% |
| Apple Pricing | 8,400 | 7,239 | 86.2% |

*Table 5.* MASE Distribution

| Dataset | 10th Pct. | 25th Pct. | Median (50th) | 75th Pct. | 90th Pct. |
|---|---|---|---|---|---|
| Solar | 0.68 | 0.76 | 0.89 | 1.03 | 1.17 |
| Electricity | 0.71 | 0.94 | 1.32 | 1.95 | 3.05 |
| Car-parts | 0.05 | 0.24 | 0.48 | 0.96 | 3.26 |
| Hierarchical Sales | 0.63 | 0.72 | 0.84 | 0.97 | 1.14 |
| Bitbrains Fast Storage | 0.65 | 0.80 | 0.93 | 1.56 | 2.58 |
| Web Traffic | 0.61 | 0.76 | 0.94 | 1.16 | 1.50 |
| Traffic | 0.71 | 0.88 | 1.09 | 1.47 | 1.91 |
| NYC Taxi | 0.71 | 0.88 | 1.12 | 1.45 | 1.78 |
| Amazon Pricing | 0.42 | 0.64 | 0.86 | 1.19 | 1.70 |
| Apple Pricing | 0.46 | 0.65 | 0.88 | 1.16 | 1.61 |

this mathematical grounding mitigates stylistic bias and ensures the benchmark evaluates true forecasting success. To empirically validate this, we recreated reference traces for 100 samples across representative datasets from each domain using Claude-Sonnet-4.5. The underlying trends remained highly consistent, demonstrating a strong agreement on which specific samples succeeded ($MASE < 1.0$). Specifically, we observed agreement rates of 84% for Electricity (Energy), 96% for Car-parts (Sales), 87% for Web Traffic (Web/CloudOps), 83% for NYC Taxi (Transportation), and 86% for Amazon pricing (Economics/Finance). This robustness is further supported by Claude's highly competitive performance in our main evaluations. We plan to utilize newer models for future benchmark iterations (e.g., TFRBench V2, V3).

### C.11. Filter Discard Rates and Task Difficulty

To further highlight the rigor of our data creation pipeline and the difficulty of the forecasting task, we analyze the discard rates and MASE distributions of the generated candidates.

**High Discard Rates:** Our strict filtering criteria ($MASE < 1.0$, $S \geq 4$) discarded 83.9% of all generated candidates (86,951 out of 103,660). For highly stochastic datasets like Electricity and NYC Taxi, the discard rate reached up to 95.0%. This high rejection volume proves that our verifier actively and aggressively filters out numerical hallucinations and narrative biases.

**Task Difficulty:** Even when isolating the single best candidate per sample, the median and upper-percentile MASE scores frequently exceed 1.0 (e.g., the 90th percentile MASE is 3.26 for the Car-parts dataset). This confirms that achieving a mathematically sound and causally grounded forecast is inherently difficult for off-the-shelf LLMs, thereby necessitating our strict, multi-agent filtering pipeline to curate a high-quality benchmark.

## C.12. Impact of Oracle Future Event Knowledge

During the data creation phase, we utilize oracle knowledge of future events to ensure factual correctness and prioritize causality. To quantify how much of the performance gap between evaluated baseline models and our reference traces is attributable to this information asymmetry versus a genuine reasoning deficiency, we conducted a targeted ablation study. We generated and evaluated 100 instances where we granted the baseline models explicit oracle knowledge of future events, thereby eliminating the asymmetry.

Bridging this gap yielded negligible improvements in baseline performance. The average MASE shifted minimally from 1.78 to 1.76, and the LLM-as-a-Judge reasoning quality scores remained virtually stagnant (shifting from 3.08 to 3.11). These results empirically prove that the performance gap is not primarily driven by an "idealized oracle" advantage. Instead, TFRBench accurately measures an inherent deficiency in the baseline models' ability to synthesize causal reasoning and translate it into accurate numerical forecasts, even when provided with the relevant external events.

# D. Reliability of the Evaluation Framework

## D.1. Inter-Model Judge Reliability

To ensure the validity and objectivity of our automated evaluation framework, we conducted a rigorous inter-rater reliability analysis. A fundamental challenge in utilizing Large Language Models (LLMs) as evaluators is the potential for specific model bias, where a judge might preferentially score outputs that align with its own training distribution or generation style. To demonstrate that our scoring criteria are robust to the choice of the evaluator, we cross-validated the judgments of our primary model, Gemini-3-Pro, against a secondary independent judge, Claude-4.5.

We performed this analysis across a representative subset of the benchmark datasets. For each evaluated sample, we calculated a total quality score by aggregating the results from our four distinct criteria: Domain Relevance, Forecasting Correctness, Event Relevance, and Logic Consistency. We then measured the statistical agreement between the two judges using the Spearman Rank Correlation coefficient. This metric was specifically chosen because it assesses the monotonic relationship between the two sets of scores. In this context, this is critical as it validates that both judges agree on the relative ranking of the candidate models, although their absolute numerical scales differ due to calibration differences.

The analysis yielded a Spearman correlation coefficient of 0.842. This result indicates a strong and statistically significant consensus between the two state-of-the-art models regarding the quality of the forecasts. Such a high agreement rate provides strong evidence that our judging rubric is objective and that the resulting performance hierarchy is not an artifact of the evaluator's architecture. Based on this validation, we standardized on Gemini-3-Pro as the scalable evaluator for the full experimental suite, as it represented the state-of-the-art capability at the time of our study.

## D.2. Human Expert Validation

To establish external trust in the benchmark, we conducted a human expert evaluation on a subset of 20 samples utilizing a panel of three independent experts. The evaluators' expertise ranges from near-graduating PhD students to a Staff Research Scientist. To ensure an unbiased assessment, the human judges were kept strictly blind to our model's generated verifier scores and MASE scores.

The human ensemble (acting as a "Judge-like" score) demonstrated strong predictive validity, achieving a high Pearson correlation of 70.4% with TFRBench's verifier score. Furthermore, as detailed in Table 6, the human judges consistently discriminated between reference samples and discarded samples across all fine-grained sub-metrics.

*Table 6.* Human Expert Evaluation Scores

| Evaluation Metric | Reference Samples | Discarded Samples |
|---|---|---|
| **Domain** | 4.26 | 3.67 |
| **Forecast** | 3.88 | 2.72 |
| **Event** | 4.17 | 3.50 |
| **Logic** | 4.07 | 3.78 |

**D.3. Independent Model Co-Verification**

To further mitigate confirmation bias and prove our pipeline curates grounded reasoning rather than self-reinforcing hallucinations, an independent model (Claude-Sonnet-4.5) blindly co-verified 100 summary traces (50 with $MASE < 1.0$ and 50 with $MASE \geq 1.0$). Claude scored successful summaries highly (averaging 4.36/5.0, with 98% scoring $\geq 4$) and accurately penalized failed ones (averaging 1.62/5.0, with 100% scoring $\leq 2$). This strong alignment between objective mathematical filtering and independent semantic auditing ensures the reliability of our reference set.

## E. Analysis on Open Source Model

To evaluate the democratization of time-series reasoning, we perform an ablation study using the state-of-the-art open-source model, **Gemma-3-27B**. We focus our evaluation on two representative datasets: *Solar Daily* (Energy domain) and *Car Parts* (Sales domain).

As shown in Table 7, Gemma-3-27B struggles to maintain forecasting precision when forced to reason. In the *Solar Daily* dataset, the transition from *w/o Reasoning* to *w/ Reasoning* results in a significant performance degradation, with MAE increasing from 7.337 to 13.5. This suggests that for open-source models of this scale, the cognitive overhead of generating a reasoning chain can interfere with the numerical accuracy of the forecast output.

However, the introduction of our **Event Forecast + Reasoning** method shows a partial recovery. For *Solar Daily*, the MAE drops significantly to 3.060, and for *Car Parts*, the MASE improves from a state (13.0) back to a competitive 0.568. This indicates that providing structured event context is crucial for guiding smaller models toward logical grounding.

*Table 7.* Numerical Forecasting Results (Open Source Model). MASE represents the forecasting performance (Lower value represents better performance).

| Dataset | Models | w/o Reasoning | | w/ Reasoning | | Event Forecast + Reasoning | |
|---|---|---|---|---|---|---|---|
| | | **MAE** | **MASE** | **MAE** | **MASE** | **MAE** | **MASE** |
| Solar Daily | gemma-3-27b | $7.337_{0.000}$ | $3.604_{0.000}$ | $1.35_{0.00} \times 10^1$ | $6.240_{0.000}$ | $3.060_{0.196}$ | $1.423_{0.062}$ |
| Car Parts | gemma-3-27b | $0.305_{0.000}$ | $0.441_{0.000}$ | $1.01_{0.22} \times 10^1$ | $1.30_{0.32} \times 10^1$ | $0.386_{0.057}$ | $0.568_{0.076}$ |

*Table 8.* LLM-as-Judge Results (Open Source Model). Higher value represents better quality reasoning in the respective criteria.

| Dataset | Models | w/ Reasoning | | | | Event Forecast + Reasoning | | | |
|---|---|---|---|---|---|---|---|---|---|
| | | **Dom.** | **Fcst.** | **Evt.** | **Logic** | **Dom.** | **Fcst.** | **Evt.** | **Logic** |
| Solar Daily | gemma-3-27b-local | $2.098_{0.000}$ | $1.283_{0.002}$ | $1.922_{0.004}$ | $1.314_{0.005}$ | $2.254_{0.000}$ | $1.352_{0.000}$ | $1.886_{0.000}$ | $1.725_{0.000}$ |
| Car Parts | gemma-3-27b-local | $2.119_{0.007}$ | $1.802_{0.016}$ | $1.868_{0.007}$ | $1.929_{0.012}$ | $4.250_{0.000}$ | $2.512_{0.000}$ | $2.480_{0.000}$ | $1.921_{0.000}$ |

Table 8 details the LLM-as-a-Judge scores for reasoning quality. The results highlight a stark contrast between Gemma-3-27B and the proprietary models detailed in Table 9:

- **Logical Consistency:** While models like Gemini-3-Pro consistently score near 4.8 in logic, Gemma-3-27B peaks at 1.929 on the *Car Parts* dataset. The model often fails to connect the identified historical events to the final numerical prediction.

- **Domain and Event Relevance:** The model demonstrates a baseline understanding of the domain (scoring $\approx 4.25$ on Car Parts with Event context), but it lacks the depth required to utilize this information for precise forecasting corrections.

- **Scaling Limitations:** The performance disparity observed here justifies our decision to prioritize proprietary models for the full-scale benchmark. Gemma-3-27B requires significantly more prompt engineering or fine-tuning to reach a level of "forecasting correctness" (currently scoring between 1.2 and 2.5) that is useful for high-stakes decision-making.

# F. Prompts for Experiments

You are an expert Time Series Forecasting Evaluator participating in the TRFBench Human Eval process. Your goal is to audit a Candidate Model's prediction by comparing its reasoning and numerical outputs against a Ground Truth analysis.

**1. Input Data**
**Context:** {context_str}
**Ground Truth (The Ideal Analysis):**
Reasoning: {ground_truth_reasoning}
Actual Future Values: {gt_vals_str}
**Candidate Prediction (To Evaluate):**
Generated Reasoning: {candidate_reasoning}
Predicted Values: {cand_vals_str}
**2. Task Annotation Instructions**
You must rate the Candidate Prediction on the following four metrics. Use the specific rubrics below to assign a score (1-5) for each.
**Metric 1: Domain Relevance (1-5)** Does the reasoning incorporate domain-specific terminology and logic appropriate for the dataset context?

- 1 (Irrelevant/Wrong): Wrong domain terminology. Logic makes no sense.

- 2 (Generic): Vague language without domain terms.

- 3 (Acceptable): Basic terms used correctly but lacks depth.

- 4 (Good): Specific terminology and standard forecasting logic used correctly.

- 5 (Expert): Deep domain expertise, precise jargon, matches Ground Truth logic.

**Metric 2: Forecasting Correctness (1-5)** Does the reasoning correctly identify fundamental time-series dynamics (global trend, seasonality)?

- 1 (Incorrect): Completely misses trend or seasonality.

- 2 (Weak): Identifies trend but ignores obvious seasonality.

- 3 (Average): Captures main trend/seasonality but misses magnitude/shape.

- 4 (Strong): Correctly identifies trend, seasonality, and general shape.

- 5 (Exact): Perfectly captures trend, seasonality, and inflection points.

**Metric 3: Event Relevance (1-5)** Are external events factually grounded and causally relevant?

- 1 (Hallucination): Mentions non-existent or false events.

- 2 (Irrelevant): Real events but no logical connection.

- 3 (Correlated): Identifies event but explanation is vague.

- 4 (Causal): Clearly links event to specific data movement.

- 5 (Aligned): Detailed, accurate causal explanation of impact.

**Metric 4: Logic-to-Number Consistency (1-5)** Does the narrative explanation logically support the generated numerical forecast?

- 1 (Contradiction): Text and numbers are opposites.

- 2 (Disconnected): Text describes shapes not seen in numbers.

- 3 (Weak Consistency): Direction matches but magnitude is off.

- 4 (Consistent): Numbers generally follow the narrative.

- 5 (Alignment): Numerical forecast is a precise translation of the reasoning.

**3. Output Format**

Provide your assessment as a single valid JSON object. Do not include any text before or after the JSON.

{

"metric_1_domain_relevance": { "score": <int 1-5>, "reasoning": "..." },

"metric_2_forecasting_correctness": { "score": <int 1-5>, "reasoning": "..." },

"metric_3_event_relevance": { "score": <int 1-5>, "reasoning": "..." },

"metric_4_logic_consistency": { "score": <int 1-5>, "reasoning": "..." },

"final_critique": "<Summary of strengths/weaknesses>"

}

---

**Prompt: w/o Reasoning**

Here is a raw numerical sequence (Input Stream):

{context_str}

**Task:** Generatively forecast the next {pred_len} steps for the {num_channels} channels.

**Instructions:**

1. Treat this as a stochastic pattern completion task.

2. Forecast a plausible continuation based on the signal structure.

3. Do not think/reason. Just provide the forecast quickly.

**Output Format Requirements:**

Your output MUST be a JSON object with ONLY a 'forecast' key holding a numerical array of shape ({pred_len}, {num_channels}).

Do not include any text, explanations, or analysis. Just the JSON.

---

**Prompt: w/ Reasoning**

Your task is to do time series forecasting WITH reasoning.

Given the input Data with context:

{context_str}

Forecast the next {pred_len} steps for the {num_channels} channels.

**Strict Operational Protocols:**

1. Forecast a plausible continuation based on the signal structure.

2. **Scale Check**: Read the 'Data Scale Reference' above. Your forecast must match this order of magnitude.

3. You are permitted to use the provided features, and you must reason over them.

4. Your output must be strictly limited to the final predicted values.

5. You must output step-by-step thinking, which is your reasoning. Your reasoning must be about pre-analysis. That is it should capture why certain forecast should be made rather than post explanation of the forecast.

6. Your reasoning is like a directive to an LLM, which will be used to improve the forecasting performance of a downstream LLM. Hence, it must be detailed and specific.

7. Utilize your reasoning first, then derive the forecast.

8. Provide the result ONLY as a JSON object containing the reasoning and numerical forecast array. Do not include any additional text.

9. YOUR REASONING IS NOT ABOUT THE POST ANALYSIS RATHER IT IS A FUTURE DIRECTION FOR THE DOWNSTREAM LLM TO FOLLOW.

Your entire response should consist of nothing but the JSON object.

**Required Output Specification:**

Your response must be a valid JSON object with exactly two keys:
- "reasoning": A detailed text string documenting your reasoning.
- "forecast": The 2D numerical array of shape ({pred_len}, {num_channels}) representing the final prediction.

---

**Prompt: Event Forecast + Reasoning (Part 1: Search)**

Your task is to find historical context for a time series dataset.
**Dataset:** {dataset_name}
**Timeframe:** {start_date} to {end_date}
**Operational Protocols:**
1. Use Google Search to find significant real-world events that occurred strictly between {start_date} and {end_date}.
2. Focus on events relevant to this dataset domain (e.g. if sales data, look for holidays or economic shifts; if weather data, look for storms).
3. Do not search for anything after {end_date}.
4. Summarize your findings in a concise list.

---

**Prompt: Event Forecast + Reasoning (Part 2: Forecast)**

Your task is time series forecasting using an EVENT-DRIVEN CHAIN OF THOUGHT approach.
**Input Data Context:**
{context_str}
**External Historical Events Found (from search):**
{historical_events_context}
**Task:** Forecast the next {pred_len} steps for the {num_channels} channels.
**Strict Operational Protocols:**
1. Step 1 Historical Analysis: Review the provided numerical history and the External Historical Events listed above. Correlate them.
2. Step 2 Future Event Forecasting: Based on the history, PREDICT the likely future events that will occur during the prediction window.
3. Step 3 Numerical Forecasting: Using the forecasted events as a guide, generate the numerical forecast values.
**Output Format Requirements:**
Your output must be a valid JSON object following this strict schema:
{ "historical_events_analysis": "string... analysis of how the searched events match the data", "future_events_forecast": "string... prediction of likely future events", "final_numerical_forecast": [[...], [...]] }

## G. Additional Experimental Details

**Experiments Hyperparameter.**    To ensure reproducibility, we adhere to strict hyperparameter settings across the generation pipeline. For reasoning generation, temperature is set to 0.7 to encourage diverse hypothesis generation and strategic reasoning. For forecasting & verification, temperature is set to 0.0 to ensure deterministic execution of the reasoning and consistent verification scoring. For windowing strategy, we utilize a sequence length of 96 and a prediction length of 96 time steps for most datasets, consistent with standard long-term forecasting benchmarks (Liu et al., 2024; Nie et al., 2023; Ahamed & Cheng, 2024; Wang et al., 2024). For the sliding window generation, we utilize a stride (slide step) of 96 to create non-overlapping samples, maximizing the diversity of temporal patterns. While candidate generation exhibits inherent stochastic variance across independent runs, the reported standard deviations for our judge scores (Table 9) approach zero strictly due to statistical aggregation. Because we average these scores across hundreds of samples per dataset, the run-to-run variance of the mean naturally converges to near-zero by the law of large numbers.

**Computational Cost and Accessibility.**    To help researchers assess the practical adoption of TFRBench, we detail the computational overhead required for both data generation and model evaluation. While generating the reference reasoning from scratch is compute-intensive, averaging approximately USD 4.53 per sample using Gemini-2.5-Pro (consuming roughly 100k input tokens, 5 search queries, and 2.5k output tokens per sample), the broader research community will not incur this expense, as we are open-sourcing benchmark. Evaluating a new candidate model, however, is highly accessible and

cost-effective. A complete evaluation requires just two API passes: one for the candidate model to generate its forecast and reasoning trace, and a second for the LLM-as-a-Judge to score it. Running this full evaluation pipeline across all reference samples costs approximately USD 45.00 in total (using Gemini-3.0-Pro, based on standard Google Vertex AI pricing[2]). This low overhead ensures that TFRBench remains a practical and easily adoptable standard for the community.

## H. *TFRBench*: Example Data Samples

**Reasoning Example (Energy Domain)**

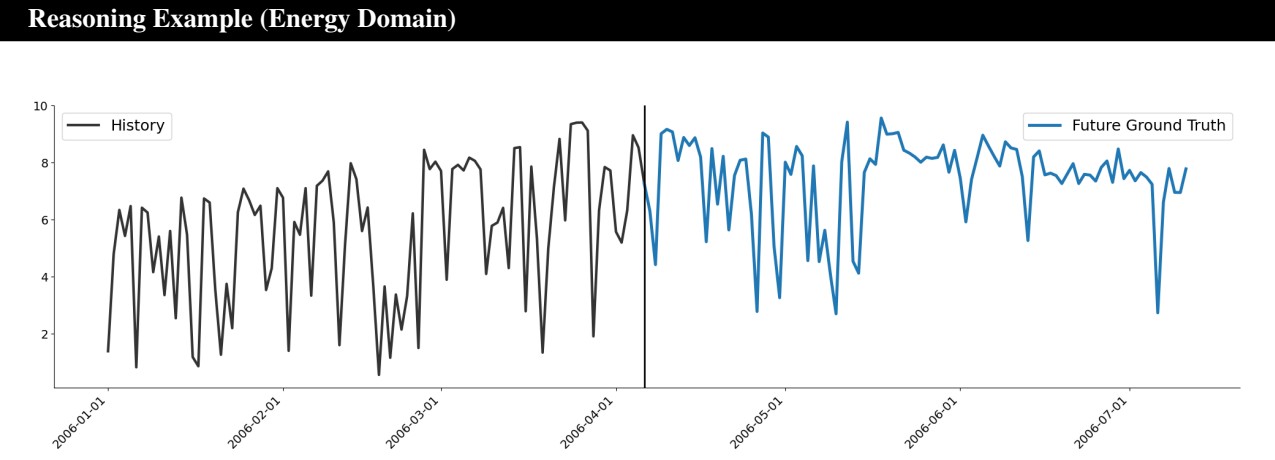

- **Cross-Channel Analysis:** The forecast for solar power production in this channel is directly driven by unobserved environmental factors. It has a strong positive relationship with solar irradiance and a strong negative relationship with cloud cover and precipitation. Therefore, any event that significantly increases cloud cover or precipitation will cause a corresponding sharp decrease in this channel's value.

- **Trend Component Plan:** The trend will continue its smooth, gradual upward trajectory throughout the forecast period. This reflects the natural seasonal increase in potential solar generation due to longer daylight hours and a higher solar angle as the season progresses from early spring towards summer. The rate of this increase should remain consistent with the pattern observed from February to March.

- **Seasonality Component Plan:** The established 7-day weekly seasonal pattern should be projected forward. This pattern should be consistently applied, maintaining the same shape and amplitude observed in the historical seasonal component data.

- **Residuals & Event Plan:** The baseline for the residuals should continue to be highly volatile to account for unpredictable daily weather changes like intermittent cloud cover. Specific adjustments must be made for the following verified event:

  - **Tornado Outbreak (April 7-8, 2006):** A severe, sharp downward shock must be applied to the forecast on these two days. The severe thunderstorms and heavy cloud cover associated with this event will drastically reduce solar production. Model a reduction in output to a level that is a small fraction of the normal baseline, likely in the range of 10% to 30% of the expected value for those days. Consider a moderate negative adjustment on April 6 and a lingering, smaller negative adjustment on April 9 to capture the storm system's arrival and departure.

  - **Non-Impactful Events:** No adjustments should be made for the "University of Alabama Founder's Day," "Confederate Memorial Day," "National Memorial Day," or "Independence Day," as these social events do not affect the physical drivers of solar power generation.

---

[2]https://cloud.google.com/vertex-ai/generative-ai/pricing

**Reasoning Example (Sales Domain)**

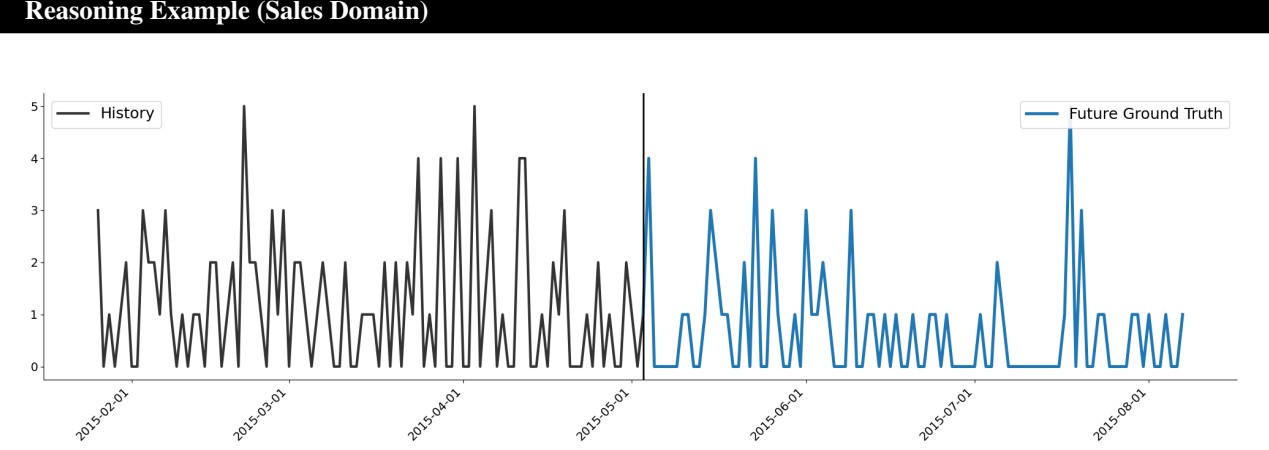

- **Cross-Channel Analysis:** As this is the only channel, analysis is based on logical product relationships. Sales of pasta are expected to be positively correlated with sales of complementary products like pasta sauce and cheese. Conversely, sales may be negatively correlated with substitute carbohydrates such as rice or potatoes, particularly if those substitutes are on promotion.

- **Trend Component Plan:** The historical trend is highly volatile and lacks a clear, sustained direction, having recently fluctuated towards zero. The plan is to stabilize the trend around this recent low baseline. This approach acknowledges the high noise level and avoids an unrealistic projection of a permanent decline. The trend should be modeled as a weak, fluctuating component.

- **Seasonality Component Plan:** The data shows a very strong and consistent 7-day seasonal pattern, which is the most reliable driver of the forecast. This weekly cycle, characterized by higher sales on weekends (peaking on Saturday) and lower sales during the early-to-mid week, must be preserved and continued into the forecast period.

- **Residuals & Event Plan:** The forecast should account for the historically large and erratic residuals, which are typical for low-volume daily sales data. This high random volatility is expected to continue. Specific event adjustments are as follows:

  - **Tornado Outbreak Sequence (May 5-10, 2015):** This event will cause a major disruption. Apply a sharp, significant positive pressure in the one to two days immediately preceding May 5th (e.g., May 3-4) to model consumer panic-buying of non-perishable staples, resulting in a temporary lift of approximately 3-5 units. During the event period (May 5-10), apply a strong negative pressure to drive sales to or near zero, reflecting potential store closures and severe shopping disruptions.
  - **Memorial Day (May 25, 2015):** Apply a moderate positive pressure on the residuals in the 2-3 days leading up to the holiday weekend (e.g., May 22-24). This reflects increased purchases for holiday gatherings where pasta salad is a common side dish, likely causing a temporary sales lift of 1-3 units.
  - **NBA Finals (starts June 4, 2015):** This event is not expected to have a discernible impact on pasta sales. No adjustments should be made.
  - **Independence Day (July 4, 2015):** Apply a moderate upward pressure on the residuals in the days immediately preceding the holiday (e.g., July 1-2). This is driven by shopping for holiday cookouts and parties and should be modeled as a temporary lift of approximately 1-3 units, similar to Memorial Day.

**Reasoning Example (Web/CloudOps Domain)**

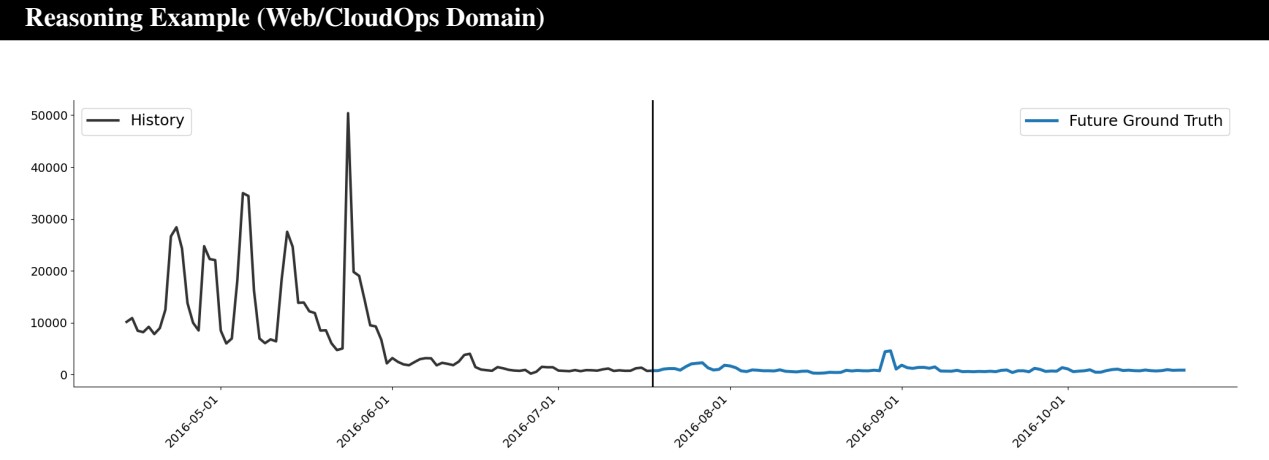

- **Cross-Channel Analysis:** As there is only one channel, cross-channel analysis is not applicable.

- **Trend Component Plan:** The time series experienced a structural break in late June 2016. The preceding period of high traffic was driven by a major external event. Following this event, the trend has sharply declined and stabilized at a much lower baseline. The forecast must project this new, low-level baseline forward. Model the trend as being predominantly flat, with a very slight negative slope to capture the final decay of public interest in the topic. Do not extrapolate from the high trend levels observed in April and May.

- **Seasonality Component Plan:** A strong weekly seasonality is present and should be maintained. This pattern consists of higher traffic on Thursdays and Fridays, with significantly lower traffic on weekends (especially Sundays) and Mondays. The amplitude of this seasonal pattern must be scaled down considerably to align with the new, lower post-event trend.

- **Residuals & Event Plan:** The historical data shows that this Wikipedia page is highly sensitive to major political news events. The extreme traffic spikes and subsequent collapse in May and June were driven by the United Kingdom's European Union membership referendum (Brexit). The forecast period begins after this event's peak influence.

  - **United Kingdom European Union membership referendum (Brexit) (June 23, 2016):** This event was the primary cause of the high traffic levels before the forecast period. Since the vote has passed, its direct impact has concluded. The forecast should reflect the post-event environment of substantially lower interest. Do not model any further positive spikes related to this event.
  - **UEFA Euro 2016 (June 10 - July 10, 2016):** This major sporting event is ongoing at the start of the forecast period. It will exert a slight, continuous downward pressure on traffic as user attention is diverted. Model a modest negative adjustment, perhaps in the range of a 5-10% reduction from the baseline, until the event concludes on July 10.
  - **Independence Day (July 4, 2016):** This US public holiday will cause a significant, one-day drop in traffic. Model a sharp negative shock on this date, comparable in magnitude to a typical low-traffic weekend day, potentially a 20-40% decrease from the expected weekday value.
  - **2016 Summer Olympics (August 5 - August 21, 2016):** This major global event will divert general user interest. Plan for a sustained, slight suppressive effect on traffic throughout this period. This should be modeled as a general downward pressure on the baseline, in the range of a 5-15% reduction, for the duration of the games.

**Reasoning Example (Transportation Domain)**

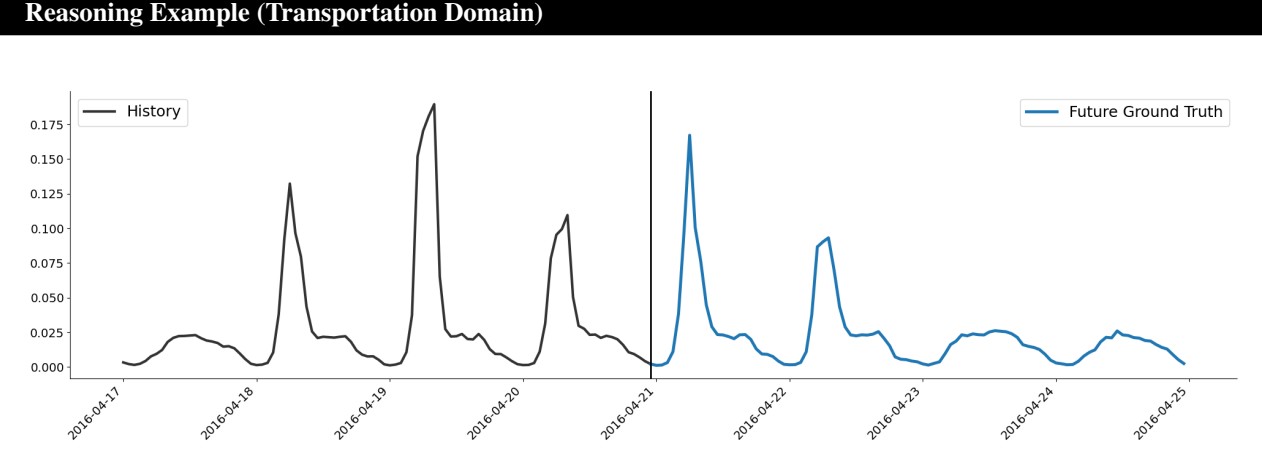

- **Cross-Channel Analysis:** This is a single-channel forecast. If data from other sensors were available, it would be expected that adjacent sensors on the same freeway would have a very high positive correlation. Sensors on alternative routes might show a slight negative correlation during periods of heavy congestion or specific event-related diversions.

- **Trend Component Plan:** The trend has shown recent volatility, and the decline to zero at the end of the historical data is an artifact of the decomposition. Do not project this sharp drop. Instead, stabilize the trend by carrying forward the approximate level observed on April 20th, which is around 0.030. The trend for the forecast period should remain relatively flat at this level.

- **Seasonality Component Plan:**

  - **For Weekdays (Thursday, April 21st and Friday, April 22nd):** The forecast must follow the established weekday seasonal pattern, which is characterized by a sharp morning commute peak (roughly 5 AM - 9 AM) and a smaller, broader evening commute peak. The magnitude of these peaks should be modeled after the typical patterns of Monday, April 18th, and Wednesday, April 20th, while disregarding the unusually high spike observed on Tuesday, April 19th.

  - **For the Weekend (Saturday, April 23rd and Sunday, April 24th):** Do not use the weekday seasonal component.

    * **For Sunday, April 24th:** The primary baseline for the forecast's shape and magnitude must be the actual historical data from the previous Sunday, April 17th. This pattern involves very low overnight traffic, a gradual rise to a broad midday/afternoon plateau (approx. 11 AM - 5 PM), and a decline into the late evening.

    * **For Saturday, April 23rd:** The baseline shape should also be modeled on the weekend pattern observed on Sunday, April 17th. However, anticipate that general Saturday traffic is typically slightly heavier and the activity may start earlier than on Sunday. Therefore, the baseline should reflect a similar shape but with a slightly higher overall volume, perhaps 5-10% greater, throughout the daytime and evening hours.

- **Residuals & Event Plan:** The baseline forecast will be constructed from the Trend and Seasonality plans. The following event-based adjustments should be applied as conservative modifications to this baseline. All major local sporting events that were initially considered have been confirmed as away games and will have no impact on local traffic.

  - **Thursday, April 21st:** The "Art Market San Francisco" begins. Apply a slight, diffuse upward adjustment to the baseline during daytime and early evening hours (e.g., 11 AM to 7 PM) to account for attendees. This should be a minor lift, in the range of 0.001 to 0.003.

  - **Friday, April 22nd:** The "Art Market San Francisco" continues. Apply a similar minor upward adjustment to the daytime baseline (0.001 to 0.003). The effect will be layered on top of the typical increase in Friday afternoon/evening traffic.

– **Saturday, April 23rd:** First, establish the weekend baseline as described in the seasonality plan. Then, apply a small positive adjustment for the "Art Market San Francisco" during its operating hours (e.g., 11 AM to 6 PM). This adjustment should be slightly more pronounced than on the weekdays, adding a lift in the range of 0.002 to 0.004 to the afternoon baseline.

– **Sunday, April 24th:** First, establish the baseline using the pattern from Sunday, April 17th. Then, apply a single, conservative adjustment for the final day of the "Art Market San Francisco". This will exert a slight upward pressure on the baseline during daytime hours, likely similar to or slightly less than Saturday's impact, in the range of 0.001 to 0.003.

**Reasoning Example (Economics/Finance Domain)**

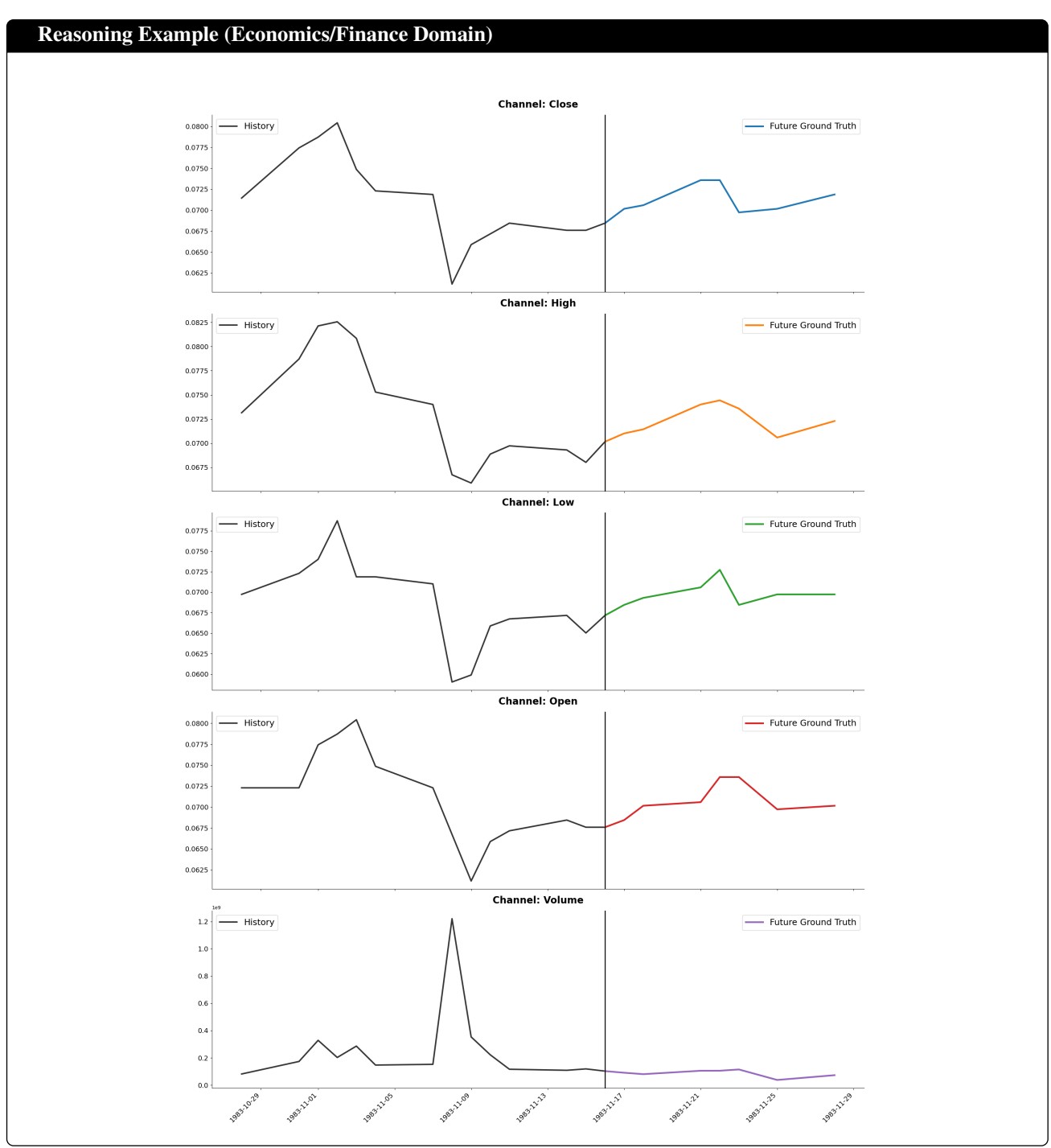

**[Close]:**
- **Cross-Channel Analysis:** The Close price forecast must be logically consistent with the other price channels, falling strictly between the High and Low forecasts for the day. It is strongly influenced by the Open price and serves as the primary anchor for the next day's Open. Significant movements in the Close price should be accompanied by a corresponding increase in the Volume forecast, signaling market conviction.

- **Trend Component Plan:** Following the recent sharp decline, model a flattening of the underlying trend. This represents a period of price stabilization. The trend should not be extrapolated downwards aggressively; instead, it should reflect the potential for the price to re-test recent low points as part of a bottoming process.

- **Seasonality Component Plan:** Project the established weekly (day-of-the-week) seasonality pattern forward. This pattern captures the regular, recurring investor behaviors observed throughout a typical trading week.

- **Residuals & Event Plan:** The "U.S. Senate Bombing (1983-11-07)" introduced significant market uncertainty.

  - **U.S. Senate Bombing (1983-11-07):** Incorporate a sharp, negative residual shock (a downward adjustment of around 2-4% of the price) on and for a few days after this date to model the resulting fear and selling pressure.
  - **Non-Impactful Events:** The "Martin Luther King, Jr. Day Bill Signed" and "Jesse Jackson Announces Presidential Candidacy" events are considered to have a negligible impact on market residuals.

**[High]:**
- **Cross-Channel Analysis:** The High forecast must be the highest value of all price channels for any given day, acting as a ceiling for the Open, Low, and Close. The spread between the High and Low is a measure of volatility and is expected to widen on days with higher forecasted Volume, reflecting greater market activity and uncertainty.

- **Trend Component Plan:** The trend for the High price will closely follow the stabilized trend modeled for the other price channels (Open, Close, Low), reflecting a pause in the recent sharp downtrend.

- **Seasonality Component Plan:** Continue to apply the established weekly seasonality pattern, which reflects recurring intraday peaks in buying pressure that vary by the day of the week.

- **Residuals & Event Plan:** The market uncertainty following the "U.S. Senate Bombing (1983-11-07)" will increase intraday volatility.

  - **U.S. Senate Bombing (1983-11-07):** Introduce moderate positive residuals (upward adjustments of 1-3% of the price) on the days following the event to model higher-than-expected price spikes as the market reacts with choppiness.

**[Low]:**
- **Cross-Channel Analysis:** The Low forecast must be the lowest value of all price channels for a given day, serving as the floor for the Open, High, and Close. A widening gap between High and Low, driven by a lower Low, indicates increased selling pressure and should be correlated with higher forecasted Volume.

- **Trend Component Plan:** The trend for the Low price will move in lockstep with the other price channels. Model a stabilization of the trend, indicating that the intense downward pressure from the recent sell-off is subsiding but that the risk of re-testing recent lows remains.

- **Seasonality Component Plan:** Project the existing weekly seasonality pattern forward, as intraday selling pressure often reaches its peak at predictable times during the trading week.

- **Residuals & Event Plan:** The "U.S. Senate Bombing (1983-11-07)" is expected to heighten investor fear.

  - **U.S. Senate Bombing (1983-11-07):** Apply a significant negative residual shock (a downward adjustment of 3-5% of the price) on and immediately after this date to capture the increased selling pressure pushing the daily low further down.

**[Open]:**

- **Cross-Channel Analysis:** The Open price is fundamentally anchored to the previous day's Close. Forecasted gaps between the prior Close and the Open signal overnight sentiment shifts and suggest a more volatile session ahead, which should be reflected in a wider High-Low range and higher Volume for that day.

- **Trend Component Plan:** The trend for the Open will be nearly identical to the Close price's trend. It should reflect the same stabilization pattern after the recent sharp price decline.

- **Seasonality Component Plan:** Maintain the established day-of-the-week seasonality pattern, as certain days may systematically open higher or lower relative to the previous close due to recurring news cycles or investor behavior.

- **Residuals & Event Plan:** The primary impact of the "U.S. Senate Bombing (1983-11-07)" would be felt at the market open on the following day.

  - **U.S. Senate Bombing (1983-11-07):** Apply a large negative residual on the morning of 1983-11-08 to model the market gapping down as it digests the overnight news.

**[Volume]:**

- **Cross-Channel Analysis:** Volume is positively correlated with volatility (the High-Low spread). A forecast for a stable or narrow trading range in the price channels should correspond with a forecast for lower Volume. Conversely, a large price movement or wide trading range implies higher Volume.

- **Trend Component Plan:** The underlying trend for Volume should show a sharp decay from the recent massive spike. Model a mean-reversion process where trading activity returns toward more normal historical levels, though potentially remaining slightly elevated compared to the pre-spike baseline.

- **Seasonality Component Plan:** Project the typical weekly seasonality forward. This often includes higher volume at the beginning and end of the week (Monday/Friday) and lower volume mid-week.

- **Residuals & Event Plan:**

  - **U.S. Senate Bombing (1983-11-07):** This event will drive a surge in trading activity. Incorporate a large positive residual shock (a spike of 1.5x to 2.5x the recent average) to Volume around this date to reflect fear-based trading.
  - **Veterans Day (1983-11-11):** Apply a moderate negative residual to model the lighter-than-expected trading volume often associated with holidays.

## I. Detailed Evaluation Results

*Table 9.* **LLM-as-a-Judge Results.** We report the mean and standard deviation (subscript) of reasoning quality scores on baseline models across three independent runs, using Gemini-3-Pro as the evaluator. Scores are measured on a scale of 1 to 5, where higher values indicate better performance. The metrics cover four dimensions: Domain Relevance (**Dom.**), Forecasting Correctness (**Fcst.**), Event Relevance (**Evt.**), and Logic-to-Number Consistency (**Logic**). Color intensity reflects performance (green: high, red: low).

| Dataset | Models | w/ Reasoning | | | | Event Forecast + Reasoning | | | |
|---|---|---|---|---|---|---|---|---|---|
| | | Dom. | Fcst. | Evt. | Logic | Dom. | Fcst. | Evt. | Logic |
| Solar Daily | Gemini-2.5-Flash | $2.472_{0.000}$ | $2.399_{0.000}$ | $1.933_{0.000}$ | $4.435_{0.000}$ | $4.378_{0.000}$ | $2.922_{0.000}$ | $2.984_{0.000}$ | $3.720_{0.000}$ |
| | Gemini-2.5-Pro | $3.026_{0.000}$ | $2.316_{0.000}$ | $2.036_{0.000}$ | $4.528_{0.000}$ | $3.803_{0.000}$ | $2.731_{0.000}$ | $2.207_{0.000}$ | $4.207_{0.000}$ |
| | Gemini-3-Pro | $4.699_{0.000}$ | $3.212_{0.000}$ | $2.508_{0.000}$ | $4.881_{0.000}$ | $4.119_{0.000}$ | $2.699_{0.000}$ | $2.368_{0.000}$ | $4.523_{0.000}$ |
| | Claude-Sonnet-4 | $4.504_{0.002}$ | $2.570_{0.000}$ | $2.503_{0.000}$ | $2.870_{0.000}$ | $4.285_{0.000}$ | $2.363_{0.000}$ | $2.228_{0.000}$ | $3.358_{0.000}$ |
| | Claude-Sonnet-4.5 | $4.922_{0.000}$ | $3.062_{0.000}$ | $2.891_{0.000}$ | $3.363_{0.000}$ | $4.554_{0.000}$ | $2.637_{0.000}$ | $2.472_{0.000}$ | $3.368_{0.000}$ |
| Electricity | Gemini-2.5-Flash | $2.856_{0.000}$ | $2.229_{0.000}$ | $1.136_{0.000}$ | $4.643_{0.000}$ | $4.616_{0.001}$ | $2.881_{0.000}$ | $3.161_{0.000}$ | $2.989_{0.000}$ |
| | Gemini-2.5-Pro | $2.318_{0.013}$ | $2.768_{0.020}$ | $1.720_{0.022}$ | $4.590_{0.015}$ | $4.688_{0.000}$ | $3.076_{0.000}$ | $3.442_{0.000}$ | $3.198_{0.000}$ |
| | Gemini-3-Pro | $4.167_{0.000}$ | $3.173_{0.000}$ | $2.178_{0.000}$ | $4.773_{0.000}$ | $4.892_{0.000}$ | $3.606_{0.000}$ | $4.147_{0.000}$ | $4.484_{0.000}$ |
| | Claude-Sonnet-4 | $3.764_{0.004}$ | $1.955_{0.002}$ | $1.432_{0.005}$ | $2.970_{0.011}$ | $4.025_{0.000}$ | $1.731_{0.000}$ | $2.451_{0.003}$ | $2.331_{0.001}$ |
| | Claude-Sonnet-4.5 | $3.810_{0.000}$ | $2.054_{0.000}$ | $2.093_{0.000}$ | $3.167_{0.000}$ | $4.609_{0.000}$ | $1.971_{0.001}$ | $2.885_{0.001}$ | $2.203_{0.001}$ |
| Car Parts | Gemini-2.5-Flash | $2.244_{0.000}$ | $2.383_{0.000}$ | $1.764_{0.000}$ | $4.845_{0.000}$ | $4.768_{0.001}$ | $2.931_{0.002}$ | $3.104_{0.001}$ | $4.619_{0.002}$ |
| | Gemini-2.5-Pro | $3.123_{0.000}$ | $2.658_{0.000}$ | $1.863_{0.000}$ | $4.181_{0.000}$ | $4.926_{0.000}$ | $3.464_{0.000}$ | $3.811_{0.000}$ | $4.923_{0.000}$ |
| | Gemini-3-Pro | $4.068_{0.000}$ | $2.900_{0.000}$ | $2.088_{0.000}$ | $4.945_{0.000}$ | $4.962_{0.000}$ | $3.012_{0.000}$ | $4.158_{0.001}$ | $4.883_{0.000}$ |
| | Claude-Sonnet-4 | $3.760_{0.000}$ | $2.605_{0.000}$ | $1.996_{0.000}$ | $3.053_{0.001}$ | $4.668_{0.000}$ | $2.305_{0.000}$ | $2.850_{0.000}$ | $3.921_{0.001}$ |
| | Claude-Sonnet-4.5 | $4.120_{0.000}$ | $2.675_{0.000}$ | $2.022_{0.000}$ | $3.384_{0.000}$ | $4.939_{0.000}$ | $3.402_{0.001}$ | $4.084_{0.000}$ | $4.402_{0.000}$ |
| Hierarchical Sales | Gemini-2.5-Flash | $2.666_{0.000}$ | $2.151_{0.000}$ | $1.035_{0.000}$ | $4.753_{0.000}$ | $2.404_{0.000}$ | $2.366_{0.000}$ | $2.123_{0.000}$ | $2.832_{0.000}$ |
| | Gemini-2.5-Pro | $2.637_{0.014}$ | $2.972_{0.007}$ | $1.635_{0.010}$ | $4.730_{0.015}$ | $2.519_{0.000}$ | $2.456_{0.000}$ | $2.230_{0.000}$ | $3.346_{0.000}$ |
| | Gemini-3-Pro | $3.701_{0.000}$ | $2.878_{0.000}$ | $1.967_{0.000}$ | $4.335_{0.000}$ | $2.476_{0.000}$ | $2.492_{0.000}$ | $2.205_{0.000}$ | $3.798_{0.000}$ |
| | Claude-Sonnet-4 | $3.091_{0.002}$ | $1.968_{0.004}$ | $1.156_{0.003}$ | $3.328_{0.010}$ | $1.896_{0.000}$ | $1.828_{0.000}$ | $1.836_{0.000}$ | $2.698_{0.000}$ |
| | Claude-Sonnet-4.5 | $3.757_{0.000}$ | $2.613_{0.000}$ | $2.046_{0.000}$ | $3.651_{0.000}$ | $2.198_{0.000}$ | $2.019_{0.000}$ | $1.947_{0.000}$ | $2.836_{0.000}$ |
| Bitbrains Fast Storage | Gemini-2.5-Flash | $3.174_{0.000}$ | $2.606_{0.000}$ | $2.699_{0.000}$ | $3.151_{0.001}$ | $3.722_{0.001}$ | $1.939_{0.001}$ | $2.353_{0.001}$ | $2.236_{0.000}$ |
| | Gemini-2.5-Pro | $3.807_{0.000}$ | $2.565_{0.000}$ | $2.944_{0.000}$ | $3.845_{0.000}$ | $3.802_{0.000}$ | $1.313_{0.003}$ | $2.652_{0.001}$ | $1.324_{0.000}$ |
| | Gemini-3-Pro | $4.531_{0.000}$ | $3.451_{0.000}$ | $3.249_{0.000}$ | $4.879_{0.000}$ | $4.746_{0.000}$ | $2.963_{0.001}$ | $2.704_{0.006}$ | $4.174_{0.002}$ |
| | Claude-Sonnet-4 | $3.942_{0.000}$ | $2.240_{0.000}$ | $3.122_{0.000}$ | $1.988_{0.000}$ | $3.557_{0.001}$ | $1.147_{0.000}$ | $2.316_{0.000}$ | $1.210_{0.001}$ |
| | Claude-Sonnet-4.5 | $4.714_{0.000}$ | $2.780_{0.000}$ | $3.427_{0.001}$ | $2.507_{0.000}$ | $3.360_{0.000}$ | $1.432_{0.000}$ | $2.083_{0.000}$ | $1.543_{0.000}$ |
| Web Traffic | Gemini-2.5-Flash | $3.158_{0.000}$ | $2.904_{0.000}$ | $1.961_{0.000}$ | $4.403_{0.000}$ | $4.544_{0.000}$ | $2.941_{0.000}$ | $3.089_{0.000}$ | $3.666_{0.000}$ |
| | Gemini-2.5-Pro | $3.585_{0.000}$ | $2.831_{0.000}$ | $2.039_{0.000}$ | $4.693_{0.000}$ | $4.576_{0.000}$ | $2.761_{0.000}$ | $3.249_{0.000}$ | $3.645_{0.000}$ |
| | Gemini-3-Pro | $4.034_{0.000}$ | $3.169_{0.000}$ | $2.179_{0.000}$ | $4.904_{0.000}$ | $4.634_{0.000}$ | $2.804_{0.000}$ | $3.435_{0.000}$ | $4.699_{0.000}$ |
| | Claude-Sonnet-4 | $3.749_{0.000}$ | $2.104_{0.000}$ | $2.080_{0.000}$ | $3.100_{0.000}$ | $4.146_{0.000}$ | $1.859_{0.000}$ | $2.520_{0.000}$ | $2.687_{0.000}$ |
| | Claude-Sonnet-4.5 | $3.928_{0.000}$ | $2.325_{0.000}$ | $2.120_{0.000}$ | $3.727_{0.000}$ | $4.276_{0.000}$ | $2.097_{0.000}$ | $2.669_{0.000}$ | $2.637_{0.001}$ |
| Traffic | Gemini-2.5-Flash | $2.708_{0.000}$ | $1.365_{0.000}$ | $1.437_{0.000}$ | $4.416_{0.000}$ | $3.216_{0.000}$ | $2.238_{0.000}$ | $1.938_{0.000}$ | $3.552_{0.000}$ |
| | Gemini-2.5-Pro | $2.762_{0.012}$ | $1.946_{0.004}$ | $1.896_{0.027}$ | $4.584_{0.034}$ | $3.643_{0.001}$ | $3.143_{0.000}$ | $2.203_{0.000}$ | $4.777_{0.000}$ |
| | Gemini-3-Pro | $3.793_{0.000}$ | $2.483_{0.000}$ | $2.249_{0.000}$ | $4.842_{0.000}$ | $2.604_{0.000}$ | $2.664_{0.000}$ | $2.022_{0.000}$ | $3.935_{0.000}$ |
| | Claude-Sonnet-4 | $3.535_{0.003}$ | $2.010_{0.004}$ | $1.468_{0.007}$ | $2.946_{0.017}$ | $2.710_{0.000}$ | $2.127_{0.000}$ | $1.606_{0.000}$ | $3.423_{0.000}$ |
| | Claude-Sonnet-4.5 | $4.294_{0.000}$ | $2.254_{0.000}$ | $2.488_{0.000}$ | $3.586_{0.000}$ | $3.381_{0.000}$ | $2.379_{0.000}$ | $1.962_{0.000}$ | $3.454_{0.000}$ |
| Nyc Taxi | Gemini-2.5-Flash | $2.808_{0.000}$ | $1.248_{0.000}$ | $1.176_{0.000}$ | $4.327_{0.000}$ | $4.670_{0.000}$ | $2.097_{0.000}$ | $2.716_{0.000}$ | $2.645_{0.000}$ |
| | Gemini-2.5-Pro | $2.814_{0.018}$ | $1.922_{0.010}$ | $1.703_{0.018}$ | $4.360_{0.040}$ | $4.972_{0.000}$ | $3.529_{0.001}$ | $3.425_{0.000}$ | $4.652_{0.000}$ |
| | Gemini-3-Pro | $3.969_{0.000}$ | $2.678_{0.000}$ | $2.113_{0.000}$ | $4.872_{0.000}$ | $4.941_{0.000}$ | $3.844_{0.000}$ | $3.941_{0.000}$ | $4.555_{0.000}$ |
| | Claude-Sonnet-4 | $3.576_{0.008}$ | $2.078_{0.004}$ | $1.327_{0.002}$ | $2.868_{0.016}$ | $4.806_{0.000}$ | $2.645_{0.000}$ | $2.831_{0.000}$ | $3.407_{0.000}$ |
| | Claude-Sonnet-4.5 | $4.532_{0.000}$ | $2.297_{0.000}$ | $2.363_{0.000}$ | $2.767_{0.000}$ | $4.890_{0.000}$ | $2.719_{0.000}$ | $3.156_{0.000}$ | $3.097_{0.000}$ |
| Amazon | Gemini-2.5-Flash | $2.082_{0.000}$ | $1.473_{0.000}$ | $1.625_{0.000}$ | $3.624_{0.001}$ | $4.652_{0.000}$ | $2.091_{0.000}$ | $2.846_{0.001}$ | $3.119_{0.000}$ |
| | Gemini-2.5-Pro | $2.262_{0.000}$ | $1.488_{0.000}$ | $1.637_{0.000}$ | $3.213_{0.000}$ | $4.966_{0.000}$ | $2.596_{0.001}$ | $3.617_{0.001}$ | $4.640_{0.002}$ |
| | Gemini-3-Pro | $2.936_{0.000}$ | $1.829_{0.000}$ | $1.784_{0.000}$ | $3.976_{0.000}$ | $4.997_{0.000}$ | $3.433_{0.000}$ | $4.125_{0.000}$ | $4.857_{0.000}$ |
| | Claude-Sonnet-4 | $2.686_{0.000}$ | $1.680_{0.000}$ | $1.677_{0.000}$ | $2.811_{0.000}$ | $4.884_{0.000}$ | $2.430_{0.000}$ | $3.195_{0.000}$ | $4.284_{0.000}$ |
| | Claude-Sonnet-4.5 | $3.588_{0.000}$ | $2.012_{0.000}$ | $1.893_{0.000}$ | $3.970_{0.000}$ | $4.988_{0.000}$ | $2.543_{0.000}$ | $3.179_{0.001}$ | $4.491_{0.000}$ |
| Apple | Gemini-2.5-Flash | $2.069_{0.000}$ | $1.523_{0.000}$ | $1.613_{0.000}$ | $3.513_{0.000}$ | $4.806_{0.000}$ | $2.081_{0.000}$ | $3.057_{0.000}$ | $3.275_{0.000}$ |
| | Gemini-2.5-Pro | $2.217_{0.001}$ | $1.557_{0.001}$ | $1.695_{0.000}$ | $3.380_{0.002}$ | $4.986_{0.000}$ | $2.556_{0.000}$ | $3.648_{0.000}$ | $4.642_{0.000}$ |
| | Gemini-3-Pro | $3.000_{0.000}$ | $1.842_{0.001}$ | $1.800_{0.000}$ | $3.821_{0.001}$ | $5.000_{0.000}$ | $3.327_{0.001}$ | $4.313_{0.001}$ | $4.826_{0.002}$ |
| | Claude-Sonnet-4 | $2.525_{0.000}$ | $1.680_{0.000}$ | $1.601_{0.000}$ | $2.608_{0.001}$ | $4.979_{0.001}$ | $2.384_{0.001}$ | $3.277_{0.000}$ | $4.462_{0.002}$ |
| | Claude-Sonnet-4.5 | $3.617_{0.000}$ | $2.075_{0.000}$ | $1.917_{0.000}$ | $3.786_{0.000}$ | $4.996_{0.000}$ | $2.515_{0.000}$ | $3.387_{0.000}$ | $4.642_{0.001}$ |

*Table 10.* **Numerical Performance Evaluation.** We report the mean and standard deviation (subscript) of Mean Absolute Error (**MAE**) and Mean Absolute Scaled Error (**MASE**) scores on baseline models. For both metrics, lower values indicate better performance. To account for varying native scales across datasets, the color gradient is normalized within each dataset block rather than across the entire table. Green indicates the best performance (lowest error) and red indicates the worst performance (highest error) per task.

| Dataset | Models | w/o Reasoning | | w/ Reasoning | | Event Forecast + Reasoning | |
|---|---|---|---|---|---|---|---|
| | | MAE | MASE | MAE | MASE | MAE | MASE |
| Solar Daily | ARIMA | $1.400_{0.000}$ | $0.694_{0.000}$ | | | | |
| | TimesFM-2.5 | $1.438_{0.000}$ | $0.732_{0.000}$ | | | | |
| | Chronos-2.0 | $1.380_{0.000}$ | $0.684_{0.000}$ | | | | |
| | Gemini-2.5-Flash | $2.445_{0.000}$ | $1.168_{0.000}$ | $1.860_{0.000}$ | $0.930_{0.000}$ | $1.810_{0.067}$ | $0.896_{0.015}$ |
| | Gemini-2.5-Pro | $2.316_{0.020}$ | $1.114_{0.006}$ | $2.802_{0.073}$ | $1.304_{0.027}$ | $2.053_{0.031}$ | $1.016_{0.015}$ |
| | Gemini-3-Pro | $2.127_{0.000}$ | $1.037_{0.000}$ | $1.705_{0.000}$ | $0.834_{0.000}$ | $2.140_{0.029}$ | $1.038_{0.010}$ |
| | Claude-Sonnet-4 | $2.065_{0.028}$ | $0.996_{0.009}$ | $2.817_{0.159}$ | $1.340_{0.092}$ | $1.925_{0.018}$ | $0.941_{0.005}$ |
| | Claude-Sonnet-4.5 | $1.870_{0.070}$ | $0.965_{0.056}$ | $1.632_{0.001}$ | $0.814_{0.002}$ | $1.887_{0.016}$ | $0.926_{0.006}$ |
| Electricity | ARIMA | $5.833_{0.000}$ | $1.427_{0.000}$ | | | | |
| | TimesFM-2.5 | $4.633_{0.000}$ | $1.243_{0.000}$ | | | | |
| | Chronos-2.0 | $4.404_{0.000}$ | $1.177_{0.000}$ | | | | |
| | Gemini-2.5-Flash | $6.137_{0.000}$ | $1.548_{0.000}$ | $6.494_{0.000}$ | $1.867_{0.000}$ | $4.785_{0.090}$ | $1.282_{0.018}$ |
| | Gemini-2.5-Pro | $5.742_{0.081}$ | $1.544_{0.007}$ | $9.426_{0.763}$ | $2.853_{0.159}$ | $5.043_{0.229}$ | $1.327_{0.045}$ |
| | Gemini-3-Pro | $4.452_{0.000}$ | $1.180_{0.000}$ | $4.063_{0.000}$ | $1.070_{0.000}$ | $3.800_{0.137}$ | $1.040_{0.025}$ |
| | Claude-Sonnet-4 | $8.580_{0.213}$ | $2.173_{0.006}$ | $8.436_{0.293}$ | $2.637_{0.420}$ | $8.037_{0.752}$ | $2.039_{0.045}$ |
| | Claude-Sonnet-4.5 | $6.996_{1.024}$ | $1.653_{0.063}$ | $5.955_{0.079}$ | $1.571_{0.045}$ | $6.360_{0.083}$ | $1.694_{0.033}$ |
| Car Parts | ARIMA | $0.531_{0.000}$ | $0.784_{0.000}$ | | | | |
| | TimesFM-2.5 | $0.328_{0.000}$ | $0.453_{0.000}$ | | | | |
| | Chronos-2.0 | $0.301_{0.000}$ | $0.425_{0.000}$ | | | | |
| | Gemini-2.5-Flash | $0.510_{0.000}$ | $0.730_{0.000}$ | $0.402_{0.000}$ | $0.588_{0.000}$ | $0.478_{0.016}$ | $0.688_{0.018}$ |
| | Gemini-2.5-Pro | $0.540_{0.003}$ | $0.777_{0.010}$ | $0.348_{0.003}$ | $0.494_{0.004}$ | $0.359_{0.006}$ | $0.502_{0.008}$ |
| | Gemini-3-Pro | $0.567_{0.000}$ | $0.821_{0.000}$ | $0.468_{0.000}$ | $0.672_{0.000}$ | $0.463_{0.000}$ | $0.646_{0.001}$ |
| | Claude-Sonnet-4 | $0.538_{0.002}$ | $0.873_{0.003}$ | $0.396_{0.004}$ | $0.571_{0.010}$ | $0.660_{0.012}$ | $0.935_{0.021}$ |
| | Claude-Sonnet-4.5 | $0.464_{0.003}$ | $0.670_{0.005}$ | $0.398_{0.000}$ | $0.579_{0.006}$ | $0.428_{0.001}$ | $0.616_{0.011}$ |
| Hierarchical Sales | ARIMA | $2.343_{0.000}$ | $0.778_{0.000}$ | | | | |
| | TimesFM-2.5 | $2.211_{0.000}$ | $0.732_{0.000}$ | | | | |
| | Chronos-2.0 | $2.258_{0.000}$ | $0.742_{0.000}$ | | | | |
| | Gemini-2.5-Flash | $2.803_{0.000}$ | $0.913_{0.000}$ | $2.694_{0.000}$ | $0.879_{0.000}$ | $2.607_{0.013}$ | $0.852_{0.003}$ |
| | Gemini-2.5-Pro | $2.842_{0.017}$ | $0.939_{0.005}$ | $2.823_{0.033}$ | $0.917_{0.009}$ | $2.955_{0.006}$ | $0.953_{0.003}$ |
| | Gemini-3-Pro | $2.736_{0.000}$ | $0.910_{0.000}$ | $2.402_{0.000}$ | $0.798_{0.000}$ | $2.632_{0.007}$ | $0.877_{0.001}$ |
| | Claude-Sonnet-4 | $2.728_{0.080}$ | $0.882_{0.010}$ | $4.053_{0.151}$ | $1.332_{0.091}$ | $3.577_{0.053}$ | $1.110_{0.011}$ |
| | Claude-Sonnet-4.5 | $2.486_{0.007}$ | $0.824_{0.002}$ | $2.506_{0.022}$ | $0.822_{0.005}$ | $3.195_{0.029}$ | $1.029_{0.005}$ |
| Bitbrains Fast Storage | ARIMA | $3.53_{0.00} \times 10^4$ | $0.804_{0.000}$ | | | | |
| | TimesFM-2.5 | $2.61_{0.00} \times 10^4$ | $0.705_{0.000}$ | | | | |
| | Chronos-2.0 | $2.66_{0.00} \times 10^4$ | $0.701_{0.000}$ | | | | |
| | Gemini-2.5-Flash | $5.07_{0.00} \times 10^4$ | $1.099_{0.000}$ | $4.58_{0.00} \times 10^4$ | $1.042_{0.000}$ | $8.64_{0.35} \times 10^4$ | $4.087_{0.168}$ |
| | Gemini-2.5-Pro | $4.83_{0.81} \times 10^4$ | $1.035_{0.008}$ | $4.58_{0.39} \times 10^4$ | $1.197_{0.032}$ | $1.22_{0.02} \times 10^5$ | $6.461_{0.048}$ |
| | Gemini-3-Pro | $4.28_{0.00} \times 10^4$ | $0.872_{0.000}$ | $4.24_{0.00} \times 10^4$ | $0.859_{0.000}$ | $3.88_{0.09} \times 10^4$ | $1.123_{0.021}$ |
| | Claude-Sonnet-4 | $5.25_{0.11} \times 10^4$ | $1.275_{0.111}$ | $5.16_{0.27} \times 10^4$ | $1.378_{0.028}$ | $1.61_{0.08} \times 10^5$ | $7.509_{0.134}$ |
| | Claude-Sonnet-4.5 | $4.63_{0.06} \times 10^4$ | $1.104_{0.002}$ | $4.08_{0.18} \times 10^4$ | $0.938_{0.010}$ | $1.18_{0.05} \times 10^5$ | $5.469_{0.066}$ |
| Web Traffic | ARIMA | $1.54_{0.00} \times 10^1$ | $0.788_{0.000}$ | | | | |
| | TimesFM-2.5 | $1.20_{0.00} \times 10^1$ | $0.676_{0.000}$ | | | | |
| | Chronos-2.0 | $1.19_{0.00} \times 10^1$ | $0.679_{0.000}$ | | | | |
| | Gemini-2.5-Flash | $1.72_{0.00} \times 10^1$ | $0.917_{0.000}$ | $2.33_{0.00} \times 10^1$ | $1.120_{0.000}$ | $1.86_{0.10} \times 10^1$ | $0.964_{0.016}$ |
| | Gemini-2.5-Pro | $1.74_{0.02} \times 10^1$ | $0.934_{0.010}$ | $2.82_{0.10} \times 10^1$ | $1.538_{0.046}$ | $1.86_{0.04} \times 10^1$ | $1.014_{0.009}$ |
| | Gemini-3-Pro | $1.54_{0.00} \times 10^1$ | $0.851_{0.000}$ | $1.38_{0.00} \times 10^1$ | $0.775_{0.000}$ | $1.63_{0.02} \times 10^1$ | $0.899_{0.005}$ |
| | Claude-Sonnet-4 | $1.98_{0.01} \times 10^1$ | $0.899_{0.008}$ | $1.99_{0.04} \times 10^1$ | $1.211_{0.040}$ | $3.03_{0.25} \times 10^1$ | $1.280_{0.047}$ |
| | Claude-Sonnet-4.5 | $1.56_{0.01} \times 10^1$ | $0.823_{0.003}$ | $1.80_{0.08} \times 10^1$ | $0.865_{0.003}$ | $2.68_{0.43} \times 10^1$ | $1.066_{0.016}$ |
| Traffic | ARIMA | $0.039_{0.000}$ | $2.628_{0.000}$ | | | | |
| | TimesFM-2.5 | $0.014_{0.000}$ | $0.847_{0.000}$ | | | | |
| | Chronos-2.0 | $0.011_{0.000}$ | $0.664_{0.000}$ | | | | |
| | Gemini-2.5-Flash | $0.020_{0.000}$ | $1.189_{0.000}$ | $0.027_{0.000}$ | $1.647_{0.000}$ | $0.020_{0.000}$ | $1.204_{0.026}$ |
| | Gemini-2.5-Pro | $0.017_{0.000}$ | $0.998_{0.010}$ | $0.030_{0.001}$ | $1.885_{0.025}$ | $0.013_{0.000}$ | $0.785_{0.007}$ |
| | Gemini-3-Pro | $0.015_{0.000}$ | $0.831_{0.000}$ | $0.018_{0.000}$ | $1.077_{0.000}$ | $0.023_{0.001}$ | $1.431_{0.036}$ |
| | Claude-Sonnet-4 | $0.032_{0.006}$ | $1.819_{0.285}$ | $0.024_{0.004}$ | $1.450_{0.168}$ | $0.018_{0.000}$ | $1.123_{0.018}$ |
| | Claude-Sonnet-4.5 | $0.024_{0.000}$ | $1.487_{0.011}$ | $0.024_{0.000}$ | $1.458_{0.001}$ | $0.021_{0.001}$ | $1.285_{0.030}$ |
| Nyc Taxi | ARIMA | $8.65_{0.00} \times 10^3$ | $3.429_{0.000}$ | | | | |
| | TimesFM-2.5 | $2.70_{0.00} \times 10^3$ | $1.051_{0.000}$ | | | | |
| | Chronos-2.0 | $1.96_{0.00} \times 10^3$ | $0.790_{0.000}$ | | | | |
| | Gemini-2.5-Flash | $2.37_{0.00} \times 10^3$ | $0.934_{0.000}$ | $5.75_{0.00} \times 10^3$ | $2.289_{0.000}$ | $4.53_{0.12} \times 10^3$ | $1.820_{0.052}$ |
| | Gemini-2.5-Pro | $2.00_{0.12} \times 10^3$ | $0.833_{0.106}$ | $7.10_{0.11} \times 10^3$ | $2.815_{0.033}$ | $2.15_{0.00} \times 10^3$ | $0.857_{0.006}$ |
| | Gemini-3-Pro | $1.70_{0.00} \times 10^3$ | $0.677_{0.000}$ | $2.91_{0.00} \times 10^3$ | $1.147_{0.000}$ | $2.56_{0.18} \times 10^3$ | $1.010_{0.064}$ |
| | Claude-Sonnet-4 | $4.76_{0.02} \times 10^3$ | $1.866_{0.019}$ | $5.31_{0.12} \times 10^3$ | $2.085_{0.055}$ | $3.52_{0.03} \times 10^3$ | $1.376_{0.013}$ |
| | Claude-Sonnet-4.5 | $5.87_{0.07} \times 10^3$ | $2.321_{0.024}$ | $4.63_{0.04} \times 10^3$ | $1.837_{0.015}$ | $4.05_{0.06} \times 10^3$ | $1.616_{0.020}$ |
| Amazon | ARIMA | $9.11_{0.00} \times 10^6$ | $0.995_{0.000}$ | | | | |
| | TimesFM-2.5 | $6.56_{0.00} \times 10^6$ | $0.747_{0.000}$ | | | | |
| | Chronos-2.0 | $7.24_{0.00} \times 10^6$ | $0.821_{0.000}$ | | | | |
| | Gemini-2.5-Flash | $7.31_{0.00} \times 10^6$ | $0.848_{0.000}$ | $1.37_{0.00} \times 10^7$ | $1.333_{0.001}$ | $1.14_{0.06} \times 10^7$ | $1.248_{0.079}$ |
| | Gemini-2.5-Pro | $6.69_{0.04} \times 10^6$ | $0.781_{0.004}$ | $1.49_{0.03} \times 10^7$ | $1.464_{0.015}$ | $6.87_{0.18} \times 10^6$ | $0.817_{0.021}$ |
| | Gemini-3-Pro | $6.36_{0.00} \times 10^6$ | $0.715_{0.000}$ | $1.05_{0.00} \times 10^7$ | $1.114_{0.000}$ | $6.72_{0.12} \times 10^6$ | $0.701_{0.002}$ |
| | Claude-Sonnet-4 | $6.82_{0.08} \times 10^6$ | $0.793_{0.005}$ | $1.34_{0.02} \times 10^7$ | $1.511_{0.050}$ | $7.40_{0.11} \times 10^6$ | $0.813_{0.011}$ |
| | Claude-Sonnet-4.5 | $7.11_{0.11} \times 10^6$ | $0.817_{0.012}$ | $7.97_{0.12} \times 10^6$ | $0.870_{0.011}$ | $6.76_{0.10} \times 10^6$ | $0.776_{0.007}$ |
| Apple | ARIMA | $2.28_{0.00} \times 10^7$ | $1.100_{0.000}$ | | | | |
| | TimesFM-2.5 | $1.86_{0.00} \times 10^7$ | $0.879_{0.000}$ | | | | |
| | Chronos-2.0 | $1.95_{0.00} \times 10^7$ | $0.936_{0.000}$ | | | | |
| | Gemini-2.5-Flash | $2.10_{0.00} \times 10^7$ | $1.011_{0.000}$ | $3.37_{0.00} \times 10^7$ | $1.482_{0.000}$ | $2.73_{0.03} \times 10^7$ | $1.273_{0.009}$ |
| | Gemini-2.5-Pro | $1.97_{0.02} \times 10^7$ | $0.939_{0.007}$ | $3.57_{0.06} \times 10^7$ | $1.589_{0.036}$ | $1.87_{0.02} \times 10^7$ | $0.875_{0.022}$ |
| | Gemini-3-Pro | $1.79_{0.00} \times 10^7$ | $0.838_{0.000}$ | $2.59_{0.00} \times 10^7$ | $1.178_{0.000}$ | $1.69_{0.03} \times 10^7$ | $0.750_{0.012}$ |
| | Claude-Sonnet-4 | $1.90_{0.01} \times 10^7$ | $0.930_{0.005}$ | $3.54_{0.09} \times 10^7$ | $1.651_{0.031}$ | $1.97_{0.02} \times 10^7$ | $0.899_{0.005}$ |
| | Claude-Sonnet-4.5 | $2.06_{0.05} \times 10^7$ | $0.947_{0.013}$ | $2.16_{0.02} \times 10^7$ | $0.989_{0.011}$ | $1.84_{0.02} \times 10^7$ | $0.858_{0.007}$ |

## J. Full Reasoning Trace (Apple Pricing)

In this section, we present one full reasoning trace for the Apple dataset.

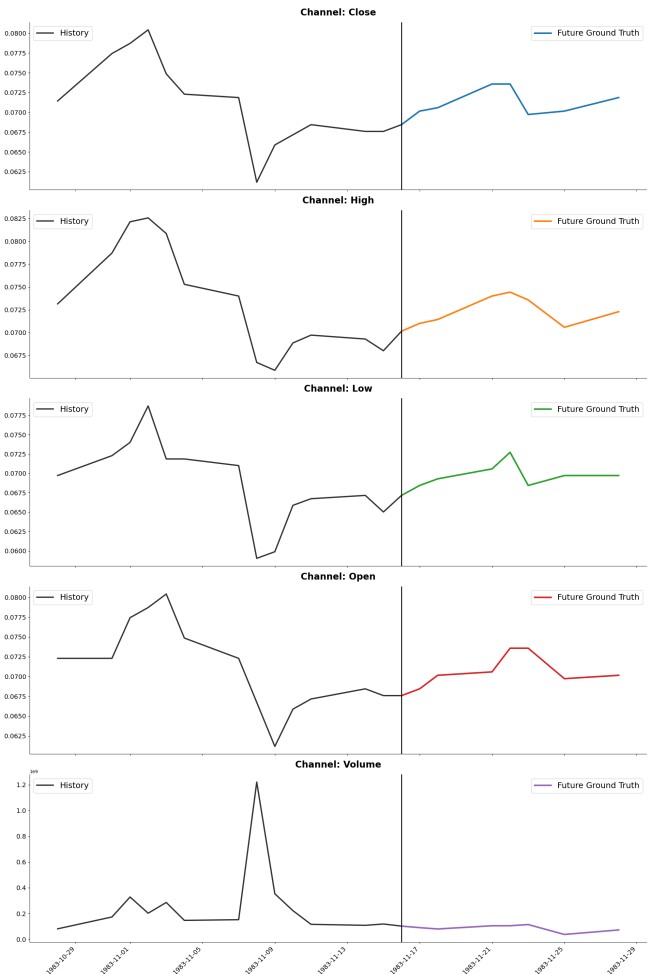

*Figure 16.* Example from Dataset Apple (Economics/Finance).

---

**Search Agent**

**Search Queries Executed:**
- "major events in USA October 1983"

- "major events in USA November 1983"

- "US public holidays November 1983"

- "major economic announcements USA November 1983"

- "significant weather events USA November 1983"
**Output:** 1. Martin Luther King, Jr. Day Bill Signed (1983-11-02) 2. Jesse Jackson Announces Presidential Candidacy (1983-11-03) 3. U.S. Senate Bombing (1983-11-07) 4. Veterans Day (1983-11-11) 5. Thanksgiving Weekend Blizzard (1983-11-26 to 1983-11-28)

---

**Initial Reasoning Generator**

**Close:**
- **Assumptions:** The Close price is driven by overall market sentiment and company-specific factors. The recent sharp price drop will lead to a period of consolidation or cautious recovery, not a continued freefall. The weekly seasonality pattern observed in

the data will persist. Events causing general market uncertainty will exert downward pressure.

- **Cross-Channel Analysis:** The Close price is highly correlated with Open, High, and Low, and its forecast must logically fit within the daily High-Low range. The previous day's Close is the strongest predictor for the current day's Open. Significant price changes in Close are expected to be accompanied by elevated Volume, indicating market conviction.

- **Trend Component Plan:** Following the recent sharp decline, the trend should not be extrapolated downwards. Instead, model a flattening of the trend, suggesting a period of price stabilization or the beginning of a slow, hesitant recovery.

- **Seasonality Component Plan:** Project the established weekly (day-of-the-week) seasonality pattern forward, as this reflects regular investor behavior throughout the trading week.

- **Residuals & Event Plan:** The 'U.S. Senate Bombing (1983-11-07)' (Source: Google Search) likely introduced fear into the market, contributing to negative residuals (downward shocks) around that date. Lingering uncertainty may cause continued, albeit smaller, negative residuals. The 'Martin Luther King, Jr. Day Bill Signed' (Source: Google Search) and 'Jesse Jackson Announces Presidential Candidacy' (Source: Google Search) are assumed to have a negligible impact.

**High:**
- **Assumptions:** The High price is a function of the daily trend and intraday volatility. The recent period of high volatility, characterized by a wide price range, will likely moderate but remain elevated in the short term. Events that increase market uncertainty will tend to widen the daily trading range, pushing the High price further from the Open/Close.

- **Cross-Channel Analysis:** The High forecast must be strictly greater than or equal to the forecasts for Open, Low, and Close. The spread between High and Low is a measure of volatility and is expected to be positively correlated with Volume. A strong upward trend in other price channels will necessitate a corresponding or stronger upward move in the High.

- **Trend Component Plan:** The trend for High will closely track the trend of the other price channels (Open, Close, Low). It should show a similar pattern of stabilization after the recent sharp drop.

- **Seasonality Component Plan:** Continue to apply the established weekly seasonality pattern. The peak intraday price likely follows a recurring rhythm based on the day of the week.

- **Residuals & Event Plan:** Residuals represent unexpected intraday price spikes. The 'U.S. Senate Bombing (1983-11-07)' (Source: Google Search) could lead to increased market choppiness, creating larger positive residuals (higher-than-expected peaks) on subsequent days as the market reacts with uncertainty. Other listed events are assumed to have no impact.

**Low:**
- **Assumptions:** The Low price reflects the maximum intraday selling pressure. Given the recent sharp sell-off, selling pressure remains a key risk, and the Low price may re-test recent bottoms. Market fear induced by external events will put downward pressure on the daily Low.

- **Cross-Channel Analysis:** The Low forecast must be strictly less than or equal to the forecasts for Open, High, and Close. The High-Low spread is positively correlated with Volume. A significant drop in the Close price will be preceded or accompanied by an even lower Low for that day.

- **Trend Component Plan:** The trend for the Low price will move in lockstep with the other price channels. The plan is to model a stabilization of the trend, indicating that the intense downward pressure is subsiding.

- **Seasonality Component Plan:** Project the existing weekly seasonality pattern forward, as intraday selling pressure often varies predictably across the days of the trading week.

- **Residuals & Event Plan:** The 'U.S. Senate Bombing (1983-11-07)' (Source: Google Search) is expected to cause negative shocks (lower-than-expected lows) to the residuals due to heightened investor anxiety. This effect may persist for a few trading sessions as the market digests the news.

**Open:**
- **Assumptions:** The Open price is predominantly determined by the previous day's Close and any overnight news flow. Following the recent major price move, opening gaps (differences from the previous close) may be more frequent. Geopolitical events can impact overnight sentiment and thus the Open.

- **Cross-Channel Analysis:** The Open price is the starting point for the day's trading and is strongly anchored to the previous day's Close. A large gap between the prior Close and the Open often signals a volatile session with higher Volume and a wider High-Low range.

- **Trend Component Plan:** The trend for the Open will be nearly identical to that of the Close. It should reflect the same stabilization pattern after the recent sharp decline.

- **Seasonality Component Plan:** Maintain the day-of-the-week seasonality pattern. Certain days of the week may systematically open higher or lower relative to the previous close.

- **Residuals & Event Plan:** The primary impact of the 'U.S. Senate Bombing (1983-11-07)' (Source: Google Search) on the Open would be seen on the following morning (11-08), likely causing a significant negative residual as the market opened down on the news. Any aftershocks could cause minor residual volatility on subsequent opens. Other events are assumed to have no impact.

**Volume:**

- **Assumptions:** Trading Volume reflects market interest and conviction. The recent volume spike was an anomalous event and will not persist; volume will revert towards the mean but may stay elevated above the pre-spike baseline. Days around holidays may experience lighter trading volume.

- **Cross-Channel Analysis:** Volume is positively correlated with volatility (the High-Low spread). A forecast for a large price movement in any of the OHLC channels should correspond with a forecast for higher Volume. Conversely, a stable price forecast implies lower Volume.

- **Trend Component Plan:** The trend for Volume should show a sharp decay from the recent massive peak, modeling a return to more normal levels of trading activity. It should not be projected to stay at its peak.

- **Seasonality Component Plan:** Project the weekly seasonality forward. Volume often follows a U-shaped pattern during the week, being higher on Mondays and Fridays and lower in the middle of the week.

- **Residuals & Event Plan:** The 'U.S. Senate Bombing (1983-11-07)' (Source: Google Search) would cause a positive shock to Volume residuals, as uncertainty and fear drive trading. 'Veterans Day (1983-11-11)' (Source: Google Search), while a market day, could exert slight downward pressure on the residuals, resulting in lighter-than-expected trading volume.

---

## Search Verifier

**Verification Queries:** "market behavior after sharp price drop", "effect of market uncertainty on stock prices", "trading volume behavior after a spike", "mean reversion in trading volume", "trading volume around holidays".
**Output:**
1. The Close price is driven by overall market sentiment and company-specific factors. - **VERDICT: RIGHT** (Reason: Stock prices are influenced by a combination of market sentiment, which reflects the collective mood of investors, and fundamental factors such as company performance and earnings.)

2. The recent sharp price drop will lead to a period of consolidation or cautious recovery, not a continued freefall. - **VERDICT: UNVERIFIABLE** (Reason: While markets can experience periods of consolidation or recovery after a sharp drop, predicting the exact future movement is not possible. Some sources suggest that after a significant drop, a retest of the lows is possible.)

3. The weekly seasonality pattern observed in the data will persist. - **VERDICT: UNVERIFIABLE** (Reason: While historical seasonal patterns in the stock market are documented, their persistence is not guaranteed as market dynamics can change.)

4. Events causing general market uncertainty will exert downward pressure. - **VERDICT: RIGHT** (Reason: Market uncertainty generally leads to risk aversion among investors, which can result in selling pressure and downward movement in stock prices.)

5. The High price is a function of the daily trend and intraday volatility. - **VERDICT: RIGHT** (Reason: The daily high price is inherently linked to the overall trend of the day's trading and is pushed higher by increased intraday volatility.)

6. The recent period of high volatility, characterized by a wide price range, will likely moderate but remain elevated in the short term. - **VERDICT: UNVERIFIABLE** (Reason: While volatility can sometimes remain elevated after a significant market event, predicting its exact behavior is not possible.)

7. Events that increase market uncertainty will tend to widen the daily trading range, pushing the High price further from the Open/Close. - **VERDICT: RIGHT** (Reason: Increased uncertainty often leads to higher market volatility, which manifests as a wider daily trading range (the difference between the high and low prices).)

8. The Low price reflects the maximum intraday selling pressure. - **VERDICT: RIGHT** (Reason: The low of the day represents the lowest price at which a stock trades and is a direct reflection of the peak of selling pressure during that trading session.)

9. Given the recent sharp sell-off, selling pressure remains a key risk, and the Low price may re-test recent bottoms. - **VERDICT: RIGHT** (Reason: After a significant sell-off, it is a recognized pattern in technical analysis that markets will often retest previous lows as part of the bottoming process.)

10. Market fear induced by external events will put downward pressure on the daily Low. - **VERDICT: RIGHT** (Reason: Fear and negative sentiment in the market lead to increased selling pressure, which can drive stock prices down, including the daily low.)

11. The Open price is predominantly determined by the previous day's Close and any overnight news flow. - **VERDICT: RIGHT** (Reason: The opening price is heavily influenced by the previous day's closing price, as well as overnight news, pre-market trading, and changes in supply and demand that occur after the market has closed.)

12. Following the recent major price move, opening gaps (differences from the previous close) may be more frequent. - **VERDICT: RIGHT** (Reason: Significant news and market-moving events often occur overnight, leading to a higher likelihood of the opening price gapping up or down from the previous day's close.)

13. Geopolitical events can impact overnight sentiment and thus the Open. - **VERDICT: RIGHT** (Reason: Geopolitical events can significantly influence investor sentiment, and if they occur after market hours, this change in sentiment will be reflected in the opening price the next day.)

14. Trading Volume reflects market interest and conviction. - **VERDICT: RIGHT** (Reason: High trading volume accompanying a price move is generally interpreted as a sign of strong market conviction behind that move.)

15. The recent volume spike was an anomalous event and will not persist; volume will revert towards the mean but may stay elevated above the pre-spike baseline. - **VERDICT: RIGHT** (Reason: Financial theory suggests that trading volume, like other market metrics, tends to exhibit mean reversion. After a spike, it is likely to return to its average level over time.)

16. Days around holidays may experience lighter trading volume. - **VERDICT: RIGHT** (Reason: Trading volumes are typically lower around major holidays as many market participants are on vacation, leading to reduced market liquidity.)

## Candidate 1 (Selected Best) — MASE: 0.106 — Score: 5/5

**Close:**
- **Cross-Channel Analysis:** The Close price forecast must be logically consistent with the other price channels, falling strictly between the High and Low forecasts for the day. It is strongly influenced by the Open price and serves as the primary anchor for the next day's Open. Significant movements in the Close price should be accompanied by a corresponding increase in the Volume forecast, signaling market conviction.

- **Trend Component Plan:** Following the recent sharp decline, model a flattening of the underlying trend. This represents a period of price stabilization. The trend should not be extrapolated downwards aggressively; instead, it should reflect the potential for the price to re-test recent low points as part of a bottoming process.

- **Seasonality Component Plan:** Project the established weekly (day-of-the-week) seasonality pattern forward. This pattern captures the regular, recurring investor behaviors observed throughout a typical trading week.

- **Residuals & Event Plan:** The "U.S. Senate Bombing (1983-11-07)" introduced significant market uncertainty.

  - **U.S. Senate Bombing (1983-11-07):** Incorporate a sharp, negative residual shock (a downward adjustment of around 2-4% of the price) on and for a few days after this date to model the resulting fear and selling pressure.
  - **Non-Impactful Events:** The "Martin Luther King, Jr. Day Bill Signed" and "Jesse Jackson Announces Presidential Candidacy" events are considered to have a negligible impact on market residuals.

**High:**
- **Cross-Channel Analysis:** The High forecast must be the highest value of all price channels for any given day, acting as a ceiling for the Open, Low, and Close. The spread between the High and Low is a measure of volatility and is expected to widen on days with higher forecasted Volume, reflecting greater market activity and uncertainty.

- **Trend Component Plan:** The trend for the High price will closely follow the stabilized trend modeled for the other price channels (Open, Close, Low), reflecting a pause in the recent sharp downtrend.

- **Seasonality Component Plan:** Continue to apply the established weekly seasonality pattern, which reflects recurring intraday peaks in buying pressure that vary by the day of the week.

- **Residuals & Event Plan:** The market uncertainty following the "U.S. Senate Bombing (1983-11-07)" will increase intraday volatility.

  - **U.S. Senate Bombing (1983-11-07):** Introduce moderate positive residuals (upward adjustments of 1-3% of the price) on the days following the event to model higher-than-expected price spikes as the market reacts with choppiness.

**Low:**
- **Cross-Channel Analysis:** The Low forecast must be the lowest value of all price channels for a given day, serving as the floor for the Open, High, and Close. A widening gap between High and Low, driven by a lower Low, indicates increased selling pressure and should be correlated with higher forecasted Volume.

- **Trend Component Plan:** The trend for the Low price will move in lockstep with the other price channels. Model a stabilization of the trend, indicating that the intense downward pressure from the recent sell-off is subsiding but that the risk of re-testing recent lows remains.

- **Seasonality Component Plan:** Project the existing weekly seasonality pattern forward, as intraday selling pressure often reaches its peak at predictable times during the trading week.

- **Residuals & Event Plan:** The "U.S. Senate Bombing (1983-11-07)" is expected to heighten investor fear.

    – **U.S. Senate Bombing (1983-11-07):** Apply a significant negative residual shock (a downward adjustment of 3-5% of the price) on and immediately after this date to capture the increased selling pressure pushing the daily low further down.

**Open:**
- **Cross-Channel Analysis:** The Open price is fundamentally anchored to the previous day's Close. Forecasted gaps between the prior Close and the Open signal overnight sentiment shifts and suggest a more volatile session ahead, which should be reflected in a wider High-Low range and higher Volume for that day.

- **Trend Component Plan:** The trend for the Open will be nearly identical to the Close price's trend. It should reflect the same stabilization pattern after the recent sharp price decline.

- **Seasonality Component Plan:** Maintain the established day-of-the-week seasonality pattern, as certain days may systematically open higher or lower relative to the previous close due to recurring news cycles or investor behavior.

- **Residuals & Event Plan:** The primary impact of the "U.S. Senate Bombing (1983-11-07)" would be felt at the market open on the following day.

  - **U.S. Senate Bombing (1983-11-07):** Apply a large negative residual on the morning of 1983-11-08 to model the market gapping down as it digests the overnight news.

**Volume:**
- **Cross-Channel Analysis:** Volume is positively correlated with volatility (the High-Low spread). A forecast for a stable or narrow trading range in the price channels should correspond with a forecast for lower Volume. Conversely, a large price movement or wide trading range implies higher Volume.

- **Trend Component Plan:** The underlying trend for Volume should show a sharp decay from the recent massive spike. Model a mean-reversion process where trading activity returns toward more normal historical levels, though potentially remaining slightly elevated compared to the pre-spike baseline.

- **Seasonality Component Plan:** Project the typical weekly seasonality forward. This often includes higher volume at the beginning and end of the week (Monday/Friday) and lower volume mid-week.

- **Residuals & Event Plan:**

  - **U.S. Senate Bombing (1983-11-07):** This event will drive a surge in trading activity. Incorporate a large positive residual shock (a spike of 1.5x to 2.5x the recent average) to Volume around this date to reflect fear-based trading.
  - **Veterans Day (1983-11-11):** Apply a moderate negative residual to model the lighter-than-expected trading volume often associated with holidays.

**Verifier Feedback:** "The reasoning is exceptionally detailed and coherent, providing specific, logical plans for trend, seasonality, and event impacts that are well-grounded in the historical data. The resulting forecast exhibits an excellent MASE score and perfectly adheres to the cross-channel constraints and dynamics outlined in the reasoning."

## Candidate 2 (Rejected) — MASE: 1.859 — Score: 2/5

**Close:**
- Cross-Channel Analysis: The Close price is intrinsically linked to the Open, High, and Low, and its forecast must logically fall within the daily High-Low range. The previous day's Close is the primary anchor for the current day's Open. Significant price changes in Close are expected to be accompanied by elevated Volume, signifying strong market conviction behind the move.

- Trend Component Plan: Following the recent sharp decline, the intense downward trend momentum is expected to slow. Model a flattening of the trend, which could represent a period of price consolidation or a potential re-test of recent lows as the market seeks a new equilibrium.

- Seasonality Component Plan: Continue to model the historical weekly (day-of-the-week) seasonality. This pattern reflects a baseline of regular investor behavior, though its magnitude may be dampened or altered by the recent increase in market volatility.

- Residuals & Event Plan: The 'U.S. Senate Bombing (1983-11-07)' introduced significant fear into the market. Incorporate a sharp negative residual (downward shock) on November 7th and a subsequent negative shock on November 8th as the market fully digests the news. Lingering uncertainty may cause smaller, negative residuals for a few days afterward. The 'Martin Luther King, Jr. Day Bill Signed (1983-11-02)' and 'Jesse Jackson Announces Presidential Candidacy (1983-11-03)' are not expected to have a material impact on the residuals.

**High:**
- Cross-Channel Analysis: The High forecast must be greater than or equal to the forecasts for Open, Low, and Close. The spread between the High and Low, a key measure of volatility, is positively correlated with Volume. A strong upward move in the

Close or Open should be reflected in a correspondingly strong move in the High.

- Trend Component Plan: The trend for High will closely track the stabilizing trend of the other price channels (Open, Close, Low). Model a flattening of the trend, moving in parallel with the Close price trend.

- Seasonality Component Plan: Apply the established weekly seasonality pattern, as intraday peaks often follow a recurring rhythm based on the day of the week.

- Residuals & Event Plan: Residuals represent unexpected intraday price spikes. The market uncertainty following the 'U.S. Senate Bombing (1983-11-07)' will likely increase market choppiness. Model larger positive residuals (higher-than-expected peaks) for several days starting November 7th, reflecting an increase in intraday volatility and a wider daily trading range.

**Low:**
- Cross-Channel Analysis: The Low forecast must be less than or equal to the forecasts for Open, High, and Close. The High-Low spread is positively correlated with Volume. After the recent sell-off, the Low price is a critical indicator of remaining selling pressure.

- Trend Component Plan: The trend for the Low price will move in lockstep with the other price channels. Model a stabilization of the trend, indicating that the most intense downward pressure is subsiding but the risk of re-testing recent lows remains.

- Seasonality Component Plan: Project the existing weekly seasonality pattern forward, as intraday selling pressure often varies predictably across the days of the trading week.

- Residuals & Event Plan: The 'U.S. Senate Bombing (1983-11-07)' is expected to cause significant negative shocks (lower-than-expected lows) to the residuals due to heightened investor fear. This effect should be most pronounced on November 7th and 8th and may persist with smaller negative residuals for a few subsequent sessions.

**Open:**
- Cross-Channel Analysis: The Open price is strongly anchored to the previous day's Close, with deviations (gaps) often signaling a volatile session with higher Volume and a wider High-Low range. It sets the initial tone for the day's trading.

- Trend Component Plan: The trend for the Open will be nearly identical to that of the Close. It should reflect the same flattening pattern after the recent sharp decline, indicating a potential stabilization.

- Seasonality Component Plan: Maintain the day-of-the-week seasonality pattern, as certain days of the week may systematically open higher or lower relative to the previous day's close based on historical patterns.

- Residuals & Event Plan: The primary impact of the 'U.S. Senate Bombing (1983-11-07)' on the Open would be a significant negative residual on the morning of November 8th, reflecting overnight absorption of the news. This could manifest as a large downward gap from the previous close, potentially in the range of a 3-6% drop from the expected value. Any aftershocks could cause minor residual volatility on subsequent opens.

**Volume:**
- Cross-Channel Analysis: Volume is positively correlated with volatility (the High-Low spread). A forecast for a wide daily price range should correspond with a forecast for higher Volume. Conversely, a period of price consolidation implies a return to lower Volume.

- Trend Component Plan: The trend for Volume should show a sharp decay from the recent massive peak. Model a swift reversion toward the historical mean, as such extreme trading activity is not sustainable. The trend should settle at a level that may be slightly elevated compared to the pre-spike baseline due to lingering uncertainty.

- Seasonality Component Plan: Project the weekly seasonality forward. Volume often follows a U-shaped pattern during the week, being higher on Mondays and Fridays.

- Residuals & Event Plan: The 'U.S. Senate Bombing (1983-11-07)' will cause a large positive shock to Volume residuals on that day and potentially the next, as fear and uncertainty drive a surge in trading activity. 'Veterans Day (1983-11-11)', while a market holiday, often sees lighter trading in the surrounding days. Model a moderate negative residual for November 11th, reflecting lighter-than-expected trading volume as market participation thins.

**Verifier Feedback:** "The reasoning provides a detailed and logical breakdown of trends, seasonality, and event impacts, correctly identifying the need for stabilization after the recent shock. However, the resulting forecast contradicts this plan by showing continued high volatility and a new record volume spike, leading to a very poor MASE score."

---

**Candidate 3 — MASE: 1.807 — Score: 2/5**

**Close:**
- Cross-Channel Analysis: The Close price forecast is tightly linked with the Open, High, and Low forecasts and must always be bounded by the High-Low range for the day. The previous day's Close is the primary anchor for the current day's Open. Significant price changes forecasted for Close must be accompanied by a corresponding increase in the Volume forecast, signaling market conviction.

- Trend Component Plan: After the recent sharp decline, the downward trend momentum is expected to cease. Model a flattening

of the trend, indicating a period of price stabilization. This phase does not imply a strong recovery but rather a consolidation, where the price may move sideways.

- Seasonality Component Plan: Continue to project the established weekly (day-of-the-week) seasonality pattern forward, as this captures the regular, recurring investor behavior observed in the historical data.

- Residuals & Event Plan: The 'U.S. Senate Bombing (1983-11-07)' introduced significant market uncertainty. Model a sharp, negative residual on this date, representing a downward shock to the price. Lingering fear may cause smaller, negative residuals in the following one to two trading sessions. The 'Martin Luther King, Jr. Day Bill Signed (1983-11-02)' and 'Jesse Jackson Announces Presidential Candidacy (1983-11-03)' are not expected to have a material impact on residuals.

**High:**
- Cross-Channel Analysis: The High forecast must be greater than or equal to the forecasts for Open, Low, and Close. The spread between the High and Low forecasts is a direct measure of volatility and should be positively correlated with the Volume forecast. A strong upward move in the Close forecast necessitates a corresponding or stronger upward move in the High.

- Trend Component Plan: The trend for High will closely follow the stabilizing pattern of the other price channels (Open, Close, Low). Model a flattening of the trend, moving in unison with the Close price trend.

- Seasonality Component Plan: Continue to apply the established weekly seasonality pattern, which reflects recurring intraday peaks in buying pressure based on the day of the week.

- Residuals & Event Plan: Residuals represent unexpected intraday price spikes. The uncertainty following the 'U.S. Senate Bombing (1983-11-07)' will increase market choppiness. Model larger positive residuals (higher-than-expected peaks) for a few days after the event, reflecting this heightened intraday volatility. This effect could widen the daily range by an additional 1-2% compared to typical days.

**Low:**
- Cross-Channel Analysis: The Low forecast must be less than or equal to the forecasts for Open, High, and Close. The High-Low spread is positively correlated with Volume. A significant drop in the Close price will be accompanied by an even lower Low for that day, reflecting peak selling pressure.

- Trend Component Plan: The trend for the Low price will move in lockstep with the other price channels. Model a stabilization of the trend, indicating that the recent intense downward pressure is subsiding. The model should allow for the possibility of re-testing the recent bottom established during the sell-off.

- Seasonality Component Plan: Project the existing weekly seasonality pattern forward, as intraday selling pressure often varies predictably across the days of the trading week.

- Residuals & Event Plan: The 'U.S. Senate Bombing (1983-11-07)' is expected to cause a significant negative residual (a lower-than-expected low) on the day of the event due to heightened investor fear. This downward pressure on the daily low may persist with smaller negative residuals for the next one to two sessions.

**Open:**
- Cross-Channel Analysis: The Open price is the starting point for the day's trading and is strongly anchored to the previous day's Close. A large gap between the prior Close and the Open often signals a volatile session, which should be reflected with higher Volume and a wider High-Low range in their respective forecasts.

- Trend Component Plan: The trend for the Open will be nearly identical to that of the Close. Model the same stabilization and flattening pattern after the recent sharp decline.

- Seasonality Component Plan: Maintain the day-of-the-week seasonality pattern, as certain days of the week may systematically open higher or lower relative to the previous day's close based on historical patterns.

- Residuals & Event Plan: The primary impact from the 'U.S. Senate Bombing (1983-11-07)' will be on the following morning's open (1983-11-08). Model a significant negative residual for this date, representing the market gapping down as it digests the overnight news. The increased frequency of opening gaps after major price moves should be reflected in slightly higher overall residual volatility.

**Volume:**
- Cross-Channel Analysis: Volume is positively correlated with volatility (the High-Low spread). Forecasts for a wide daily price range should correspond with forecasts for higher Volume. Conversely, a stable or narrowing price range implies lower Volume.

- Trend Component Plan: The recent massive volume spike was an anomaly. The trend component must model a sharp decay from this peak, representing a reversion toward the historical mean trading volume. The trend should stabilize at a level that may be slightly elevated compared to the pre-spike baseline but significantly lower than the peak.

- Seasonality Component Plan: Project the weekly seasonality forward. This pattern often reflects higher trading activity at the beginning and end of the trading week (Monday/Friday) and lower activity mid-week.

- Residuals & Event Plan: The uncertainty and fear from the 'U.S. Senate Bombing (1983-11-07)' will drive trading activity. Model a large positive residual on this date, representing a volume surge (e.g., 30-50% above the decaying trend). For 'Veterans Day (1983-11-11)', which is a trading day, model a small negative residual to account for lighter-than-expected trading volume, a common occurrence around holidays.

**Verifier Feedback:** "The reasoning is detailed and logical, particularly the plan for volume to revert to the mean after a spike and for prices to stabilize. However, the resulting forecast directly contradicts this reasoning by predicting new, massive volume spikes, which is likely the cause of the very poor MASE score."

## Candidate 4 — MASE: 0.250 — Score: 5/5

**Close:**
- Cross-Channel Analysis: The Close forecast must be logically consistent with the day's other price forecasts, falling strictly between the Low and High values. It is the primary anchor for the next day's Open price. Significant movements in the Close price must be supported by a corresponding increase in the Volume forecast, indicating market conviction.

- Trend Component Plan: Model a flattening of the trend component, indicating a slowdown in the recent sharp downward momentum. This represents a potential stabilization phase. However, the forecast should allow for the possibility of the price re-testing recent lows before establishing a clear recovery. Do not extrapolate the recent steep decline.

- Seasonality Component Plan: Project the established weekly (day-of-the-week) seasonality pattern forward, as this reflects recurring investor behavior and liquidity patterns throughout the trading week.

- Residuals & Event Plan: The 'U.S. Senate Bombing (1983-11-07)' introduced significant market uncertainty. Incorporate a sharp negative residual on this date. Lingering fear may cause smaller, decaying negative residuals for the next one to two trading sessions. The 'Martin Luther King, Jr. Day Bill Signed (1983-11-02)' and 'Jesse Jackson Announces Presidential Candidacy (1983-11-03)' are considered non-market moving events and should have no residual adjustments.

**High:**
- Cross-Channel Analysis: The High forecast must be the maximum value for the day, strictly greater than or equal to the Open, Low, and Close forecasts. The spread between the High and Low forecasts, a proxy for volatility, should be modeled to moderate from its recent peak but remain elevated compared to the pre-event baseline. This elevated spread should correlate with the Volume forecast remaining above its historical average.

- Trend Component Plan: The trend for High will closely track the stabilizing trend of the other price channels (Open, Close, Low). Model a flattening of the trend to reflect the end of the immediate sharp price decline.

- Seasonality Component Plan: Continue to apply the established weekly seasonality pattern, which captures the typical intraday price peaks expected on different days of the week.

- Residuals & Event Plan: The uncertainty following the 'U.S. Senate Bombing (1983-11-07)' will increase market choppiness. Model this by introducing larger positive residuals (higher-than-expected price spikes) on November 7th and the subsequent one to two days, reflecting investor panic and reactive buying.

**Low:**
- Cross-Channel Analysis: The Low forecast must be the minimum value for the day, strictly less than or equal to the Open, High, and Close forecasts. The High-Low spread is a key indicator of volatility; a wider spread, driven by a lower Low, should be accompanied by a higher Volume forecast.

- Trend Component Plan: The trend for the Low price will move in lockstep with the other price channels, showing a similar stabilization. However, the forecast must accommodate the potential for the Low price to re-test the recent bottom established during the sell-off, reflecting persistent selling pressure as per technical analysis patterns.

- Seasonality Component Plan: Project the existing weekly seasonality pattern forward, as intraday selling pressure often varies predictably across the days of the trading week.

- Residuals & Event Plan: The fear and anxiety from the 'U.S. Senate Bombing (1983-11-07)' will exert downward pressure. Incorporate a significant negative residual (a lower-than-expected price floor) on November 7th. This effect may persist with smaller negative residuals for a few subsequent trading sessions as the market digests the news.

**Open:**
- Cross-Channel Analysis: The Open price is strongly anchored to the previous day's Close. Following recent volatility, model a higher probability of opening gaps (a noticeable difference between the prior Close and the current Open). A significant gap should signal a potentially volatile session with a wider High-Low range and higher Volume for that day.

- Trend Component Plan: The trend for the Open will be nearly identical to that of the Close. It should reflect the same stabilization pattern, moving away from the recent sharp decline.

- Seasonality Component Plan: Maintain the established day-of-the-week seasonality pattern, as certain days may systematically open higher or lower relative to the previous close due to recurring news cycles or settlement periods.

- Residuals & Event Plan: The primary impact of the 'U.S. Senate Bombing (1983-11-07)' would be reflected in the next day's open. Introduce a large negative residual on the morning of November 8th to model the market opening down on the overnight news. Any aftershocks could cause minor residual volatility on subsequent opens.

**Volume:**

- Cross-Channel Analysis: Volume is positively correlated with volatility (the High-Low spread). A forecast for a wide daily price range should correspond with a forecast for higher Volume. Conversely, a forecast for price stabilization and consolidation implies a return to more moderate Volume levels.

- Trend Component Plan: The recent volume spike was anomalous. Model a sharp decay in the Volume trend, causing it to revert towards its long-term mean. The trend should not remain at its peak but should settle at a level slightly elevated above the pre-spike baseline, reflecting heightened market awareness.

- Seasonality Component Plan: Project the weekly seasonality forward. This typically reflects higher activity on Mondays and Fridays and lower activity mid-week.

- Residuals & Event Plan: The 'U.S. Senate Bombing (1983-11-07)' would cause a positive shock to Volume. Incorporate a significant positive residual (a spike of 50-100% above the trend) on November 7th and a smaller, decaying positive residual on the following one to two days, reflecting the surge in trading due to uncertainty. For 'Veterans Day (1983-11-11)', introduce a negative residual to account for lighter-than-average holiday trading, reducing the expected volume by a notable fraction, such as in the range of 15-25%.
**Verifier Feedback:** "The reasoning is excellent, providing a detailed and coherent multi-channel analysis. The plan correctly identifies the impact of the key event, models a logical stabilization phase, and the resulting forecast aligns perfectly with this strategy, leading to a very strong MASE score."

---

### Candidate 5 — MASE: 0.776 — Score: 5/5

**Close:**
- Cross-Channel Analysis: The Close price forecast must be logically constrained within the forecasted High-Low range and will serve as the primary anchor for the next day's Open. Periods of significant price change in the Close forecast should be accompanied by a corresponding increase in the Volume forecast, reflecting higher market conviction.

- Trend Component Plan: Following the recent sharp decline, the downward trend should not be extrapolated. Instead, flatten the trend component to model a period of stabilization. This captures the potential for either a slow, cautious recovery or a re-testing of recent price lows.

- Seasonality Component Plan: Project the established weekly (day-of-the-week) seasonality pattern forward. This pattern represents the best estimate of recurring investor behavior throughout the trading week.

- Residuals & Event Plan: The 'U.S. Senate Bombing (1983-11-07)' introduced market uncertainty. Model a moderate negative residual (a downward shock) on the days immediately following the event, reflecting lingering investor fear. The 'Martin Luther King, Jr. Day Bill Signed (1983-11-02)' and 'Jesse Jackson Announces Presidential Candidacy (1983-11-03)' are not expected to have a material impact on residuals.

**High:**
- Cross-Channel Analysis: The High forecast must be greater than or equal to the forecasts for Open, Low, and Close. The spread between the High and Low forecasts, a proxy for volatility, should be positively correlated with the Volume forecast. A strong upward move in the Close or Open should be reflected in the High.

- Trend Component Plan: The trend for High will closely follow the stabilized trend of the other price channels (Open, Close, Low), reflecting a pause in the recent sharp downward movement.

- Seasonality Component Plan: Continue to apply the established weekly seasonality pattern, as intraday price peaks often follow a recurring rhythm based on the day of the week.

- Residuals & Event Plan: The market uncertainty following the 'U.S. Senate Bombing (1983-11-07)' will likely increase market choppiness. Model this by introducing larger positive residuals (higher-than-expected price peaks) for several days after the event, widening the daily trading range.

**Low:**
- Cross-Channel Analysis: The Low forecast must be less than or equal to the forecasts for Open, High, and Close. The High-Low spread is a measure of volatility and should move in tandem with Volume. After the recent sell-off, the Low price remains vulnerable, and any significant drop in the Close should be preceded by a test of a new daily Low.

- Trend Component Plan: The trend for the Low price will move in lockstep with the other price channels, modeling a stabilization that indicates the most intense downward pressure is subsiding for now.

- Seasonality Component Plan: Project the existing weekly seasonality pattern forward, as intraday selling pressure often varies predictably across the days of the trading week.

- Residuals & Event Plan: In response to the heightened investor anxiety from the 'U.S. Senate Bombing (1983-11-07)', apply a significant negative residual (a lower-than-expected price floor) on the day of and for one to two sessions after the event. This reflects the market's fear-driven selling pressure.

**Open:**

- Cross-Channel Analysis: The Open price is strongly anchored to the previous day's Close. A large forecasted gap between the prior Close and the Open signals a volatile session and should correlate with higher Volume and a wider High-Low range.

- Trend Component Plan: The trend for the Open will be nearly identical to that of the Close, reflecting the same stabilization pattern after the recent sharp decline.

- Seasonality Component Plan: Maintain the day-of-the-week seasonality pattern, as certain days of the week may systematically open higher or lower relative to the previous close due to recurring weekend or overnight effects.

- Residuals & Event Plan: The primary impact of the 'U.S. Senate Bombing (1983-11-07)' would manifest as a significant negative residual on the Open of the next trading day (1983-11-08). Model a gap down, potentially in the range of 3-6% below the previous close, as the market digests the overnight news.

**Volume:**
- Cross-Channel Analysis: Volume is positively correlated with volatility (the High-Low spread). Forecasts for stable prices should correspond with lower Volume, while forecasts for large price movements or wide High-Low ranges should be accompanied by higher Volume.

- Trend Component Plan: The trend for Volume should show a sharp decay from the recent massive peak, modeling a mean-reversion process back toward more normal levels. The trend should settle at a level slightly elevated above the pre-spike baseline, reflecting heightened market attention.

- Seasonality Component Plan: Project the weekly seasonality forward, which often shows a U-shaped pattern with higher activity on Mondays and Fridays and lower activity mid-week.

- Residuals & Event Plan: The 'U.S. Senate Bombing (1983-11-07)' will drive uncertainty and fear-based trading. Model a significant positive residual (a spike in trading activity, perhaps 50-100% above the recent trend) on the day of and the day after the event. For 'Veterans Day (1983-11-11)', introduce a moderate negative residual, reflecting lighter-than-expected trading as some market participants may be absent.

**Verifier Feedback:** "The reasoning is exceptionally detailed and demonstrates a sophisticated understanding of time series decomposition and market dynamics, correctly identifying the need for trend stabilization after a sharp drop. The resulting forecast is logically consistent with the reasoning and achieved an excellent MASE score."

## Summary Agent

**Overall Forecast Strategy:**
- **Trend:** For all price channels (Open, High, Low, Close), model a flattening of the underlying trend after the recent sharp decline to represent price stabilization and a bottoming process. For Volume, model a sharp decay from its recent spike, representing a mean-reversion back towards more normal historical levels.

- **Seasonality:** For all channels, project the established weekly (day-of-the-week) seasonality pattern forward.

**Cross-Channel Relationships (Constraints):**
- The High forecast must be the highest value of all price channels for any given day.

- The Low forecast must be the lowest value of all price channels for any given day.

- The Close forecast must fall strictly between the High and Low forecasts.

- The Open price is anchored to the previous day's Close. Forecasted gaps between the prior Close and the Open signal overnight sentiment shifts.

- Volume is positively correlated with volatility (the High-Low spread). Large price movements or a wide High-Low range must correspond with higher forecasted Volume.

**Residuals & Event Adjustments:**
- **U.S. Senate Bombing (1983-11-07):**

  - **Close:** Apply a sharp negative residual shock (a downward adjustment of 2-4% of the price) on and for a few days after this date.
  - **High:** Apply a moderate positive residual (an upward adjustment of 1-3% of the price) on the days following the event to model increased intraday volatility and price spikes.
  - **Low:** Apply a significant negative residual shock (a downward adjustment of 3-5% of the price) on and immediately after this date to capture heightened selling pressure.
  - **Open:** Apply a large negative residual on the morning of 1983-11-08 to model the market gapping down.
  - **Volume:** Incorporate a large positive residual shock (a spike of 1.5x to 2.5x the recent average) around this date to reflect fear-based trading.

- **Veterans Day (1983-11-11):**

  - **Volume:** Apply a moderate negative residual to model lighter-than-expected trading volume.

- **Other Events:**
  - The 'Martin Luther King, Jr. Day Bill Signed' and 'Jesse Jackson Announces Presidential Candidacy' events are considered to have a negligible impact. Do not apply any residual adjustments for them.

## K. LLM-as-Judge Example

We present a detailed example of LLM-as-Judge for the Amazon dataset from "Event Forecast + Reasoning" method. The Judge compares the Candidate's logic against the Reference Sample with detailed report.

The LLM-as-a-Judge protocol prioritizes a candidate model's logical and factual integrity over a verbatim match with our reference reasoning traces. By scoring across four independent dimensions (Domain Relevance, Forecasting Correctness, Event Relevance, and Logic Consistency), the rubric rewards any trajectory that is factually grounded and causally aligned with the numerical output. For instance, a model can receive high scores for Event Relevance by identifying different yet valid ex-ante events that successfully explain the forecasted dynamics. This multidimensional assessment ensures that the benchmark rewards diverse but correct explanations while strictly penalizing narrative bias or logical contradictions between text and numbers.

---

**Reference Reasoning (Target)**

**Close:**
- **Cross-Channel Analysis:** The Close price is fundamentally linked to the other price channels. It will typically be near the session's High on positive days and near the Low on negative days, reflecting the final market sentiment. A significant change in the Close price, especially on high Volume, suggests a strong market conviction that will influence the next day's Open. A large gap between the Open and Close indicates a strong intra-day trend.

- **Trend Component Plan:** The recent upward trend should be projected to continue, but its rate of growth should moderate, reflecting a natural consolidation phase after a period of gains. The forecast should not assume a simple straight-line continuation but rather a gentle flattening of the upward curve.

- **Seasonality Component Plan:** The model must adhere to the established weekly seasonality pattern observed in the historical data, which reflects typical intra-week trading behaviors.

- **Residuals & Event Plan:** The 'China announces major economic reforms' event on March 19th is expected to introduce a modest positive shock. Apply upward pressure on the residuals, resulting in a temporary upward shift in the forecast, potentially in the range of 1-3% above the baseline. The 'President Clinton invokes executive privilege' event on March 20th could introduce minor negative pressure or increased noise due to political uncertainty; model this as a potential downward adjustment or increased random fluctuation, possibly within a +/- 1% range of the baseline.

**High:**
- **Cross-Channel Analysis:** The High channel is strongly and positively correlated with Close, Open, and Low. The spread between High and Low is a key measure of volatility; an increase in this spread often correlates with a spike in Volume. On days with positive news, the High will likely be significantly above the Open and Low, with the Close finishing nearer to the High.

- **Trend Component Plan:** The trend for the High price will closely follow the primary trend of the Close price, but will consistently maintain a premium over it. Project a continued, but slightly decelerating, upward movement for this channel's trend.

- **Seasonality Component Plan:** The established weekly seasonal pattern for the High channel should be maintained, mirroring the general shape of the other price channels' seasonality.

- **Residuals & Event Plan:** The 'China announces major economic reforms' will exert upward pressure, causing a more pronounced positive deviation in the High price than in the Close price as intra-day optimism peaks. The 'President Clinton invokes executive privilege' event may increase volatility, pushing the High upwards even if the Close trends down, thus widening the daily price range.

**Low:**
- **Cross-Channel Analysis:** The Low channel is positively correlated with the other price channels. A widening gap between High and Low indicates higher volatility and will be reflected in this channel. On negative days, the Close price will settle near the Low, marking the session's point of maximum pessimism.

- **Trend Component Plan:** The trend for the Low price will move in lockstep with the other price channels, forming the lower boundary of the price trend. This upward trend should also be projected with a moderating slope.

- **Seasonality Component Plan:** Follow the established weekly seasonality for the Low channel, which will be structurally similar to the patterns in High, Open, and Close.

- **Residuals & Event Plan:** The 'China announces major economic reforms' should pull the daily Low upward, in line with general positive sentiment, establishing a higher floor for the day's trading. The 'President Clinton invokes executive privilege' is the most likely event to exert downward pressure on the residuals for the Low channel, as market uncertainty could lead to a temporary sell-off and test a new support level.

**Open:**
- **Cross-Channel Analysis:** The Open is highly correlated with the previous day's Close. A significant gap between the previous Close and the current Open indicates a strong reaction to overnight events, which often predicts higher Volume and a wider High-Low range for the day.

- **Trend Component Plan:** The trend for the Open will track the overall price trend established by the Close, High, and Low channels. Project a continued, but gently flattening, upward trend.

- **Seasonality Component Plan:** Maintain the characteristic weekly seasonal pattern for the Open price, reflecting consistent intra-week opening dynamics.

- **Residuals & Event Plan:** The Open on the day following the 'China announces major economic reforms' is likely to experience a positive shock, potentially gapping up slightly from the prior day's close. The Open on the Monday following the 'President Clinton invokes executive privilege' news from Friday may open flat or with slight downward pressure as the market digests the political uncertainty over the weekend.

---

**Volume:**
- **Cross-Channel Analysis:** Volume is positively correlated with volatility (the High-Low spread). Large absolute changes in the Close price are often accompanied by a surge in Volume, signifying strong conviction behind the price move. A price move on low volume may be less sustainable than one on high volume.

- **Trend Component Plan:** The underlying trend of increasing market participation should be projected to continue at a steady pace. The absolute level is less important than the relative spikes caused by events.

- **Seasonality Component Plan:** The weekly seasonality, which often shows patterns like higher activity on certain days of the week, should be applied as a baseline for trading activity.

- **Residuals & Event Plan:** Expect a moderate, short-lived spike in Volume residuals following the 'China announces major economic reforms' as investors react to the news; model this as an increase of approximately 20-40% above the recent average trading volume. A second, likely smaller, spike in Volume should be modeled around the 'President Clinton invokes executive privilege' event, reflecting trading activity driven by uncertainty, potentially in the 15-30% range above the recent average.

## Candidate Reasoning (Predicted)

**Step 1: Historical Search Results (1998-02-25 to 1998-03-16)**
I'll search for significant events related to Amazon during the specified timeframe in early 1998. Based on my search results, I found several significant events related to Amazon during the timeframe February 25 - March 16, 1998:
**Historical Context for Amazon Dataset (February 25 - March 16, 1998) Key Events During This Period:**
1. **Amazon.com Advantage Program Launch (February 1998):** Amazon launched Amazon.com Advantage, an innovative new program designed to increase the visibility and sales of titles from independent publishers and authors.

2. **Amazon.com Kids Launch (March 1998):** Amazon launched Amazon.com Kids, a comprehensive resource for children's and young adult books, featuring a catalog of more than 100,000 books for children, teens, and parents.

3. **Business Context:** The first quarter ending March 31, 1998 showed net sales of $87.4 million, a 32 percent increase over the fourth quarter of 1997 and a 446 percent increase over the first quarter of 1997.

4. **Customer Growth:** Cumulative customer accounts grew to over 2,260,000 at March 31, 1998, an increase of 50 percent from 1,510,000 customer accounts at December 31, 1997.

5. **Strategic Expansion Period:** This timeframe fell during Amazon's critical expansion phase as the company was preparing to move beyond books into music and other product categories later in 1998.

These events occurred during a period of rapid growth for Amazon as it solidified its position as a leading online bookseller before its major diversification into other product categories.

**Step 2: Analysis of History**
The historical data from February 25 - March 16, 1998 shows significant volatility and growth patterns that align with the external events. The stock price increased from $0.274 to $0.329 (20% gain) during this period. Key correlations include: (1) The massive volume spike on March 10 (495M shares) coincides with the Amazon.com Kids launch announcement in March 1998, representing peak investor enthusiasm. (2) The sustained high volumes from Feb 26-27 (307M and 494M shares) align with the Amazon.com Advantage Program launch in February, which generated significant market interest. (3) The upward trend component shows consistent growth from $0.304 to $0.333, reflecting the 446% year-over-year sales growth and 50% customer account growth mentioned in Q1 1998 results. (4) The price consolidation after March 10 (ranging $0.327-$0.329) suggests profit-taking after the news-driven rally, typical of event-driven stock movements. The decomposition shows a clear positive trend with cyclical seasonality patterns repeating every 7 days, and residuals near zero indicating the model captures the dynamics well.

**Step 3: Future Event Prediction**
For the next 7 trading days (March 17-25, 1998), the following events and market dynamics are likely: (1) Continued consolidation phase as the market digests the recent Amazon.com Kids launch, with reduced volatility compared to the March 9-10 spike. (2) Gradual volume normalization toward the 150-200M range as speculative interest subsides. (3) Potential minor pullback or sideways movement as early investors take profits from the 20% February-March rally. (4) Underlying positive sentiment maintained due to strong Q1 1998 fundamentals (2.26M customer accounts, $87.4M quarterly sales). (5) Anticipation building for Amazon's planned expansion into music and other categories later in 1998, providing support for the stock price. (6) Normal market microstructure with the established 7-day seasonality pattern continuing. (7) No major product launches expected immediately, leading to news-vacuum consolidation typical of post-announcement periods.

## Judge's Evaluation Report (Score: 3.5/5.0)

**Final Critique:** The model excels in domain terminology and internal consistency; its numerical predictions perfectly align with its generated narrative. However, the forecast itself was inaccurate because the model failed to retrieve or account for the specific macro-economic and political events (China reforms, Clinton executive privilege) that drove market volatility during the forecast window. Consequently, it predicted a quiet consolidation period while the actual market experienced a volatility-driven rally.
**Detailed Metric Breakdown:**
- **Domain Relevance (Score: 5):** The candidate demonstrates expert-level domain relevance, utilizing appropriate financial terminology such as 'consolidation,' 'profit-taking,' 'market microstructure,' and 'volume normalization.' The analysis of fundamentals (Q1 results, customer accounts) fits the equity forecasting domain perfectly.

- **Forecasting Correctness (Score: 2):** The forecast is weak compared to the Ground Truth. The candidate predicted a 'minor pullback' and 'consolidation' (prices dipping to 0.32), whereas the actual values and Ground Truth showed a continued upward trend to 0.35. Additionally, the candidate predicted normalized volume (150-200M), missing the significant volatility and spikes (up to 273M) seen in the actual data.

- **Event Relevance (Score: 2):** The candidate failed to identify the specific external events occurring during the forecast window (March 19th China reforms, March 20th Clinton news) which were central to the Ground Truth analysis. Instead, it assumed a 'news-vacuum' based only on the lack of company-specific product launches. While the historical context found (Amazon

Kids) was factually true, the failure to detect the relevant future macro events led to an incorrect narrative.

- **Logic Consistency (Score: 5):** The logic-to-number consistency is excellent. The candidate reasoned for 'consolidation,' 'minor pullback,' and 'volume normalization to 150-200M,' and the generated numbers reflected exactly that (Close prices drifting sideways/down and Volume staying strictly within the 145M-195M range).

### K.1. Event Temporal Integrity Audit

To ensure the validity of TFRBench, we conducted a rigorous temporal integrity audit on the 5,379 identified "Reference Reasoning" samples. Using a robust natural language date parser, we extracted 25,701 event references from the generated reasoning traces and compared them against each sample's prediction window end date to verify adherence to temporal boundaries. The audit revealed a **99.68% compliance rate**, with only 83 instances (0.32%) where an event's start date post-dated the window. Qualitative analysis indicates these flagged instances primarily consist of either predictable recurrent events (e.g., "Christmas Day") or significant, scheduled near-future events that exert a direct, anticipatable impact on the current forecasting window. Additionally, A post-hoc temporal audit of all baseline evaluations confirmed a 98.15% strict compliance rate. Qualitative analysis of the remaining 1.85% reveals they are mostly predictable calendar events with anticipatory impacts. For example, models reason: "...due to the upcoming Christmas, sales will increase..." or "...anticipating Black Friday, demand will peak." This confirms that the multi-agent framework successfully restricts context retrieval to the valid timeline, ensuring agents leverage valid anticipatory intelligence regarding imminent external drivers.

## L. Case Study: Narrative Bias and Causal Reasoning

To empirically demonstrate the phenomenon of "narrative bias" in baseline large language models, we conduct a comparative case study using the Amazon dataset. We evaluate the reasoning capabilities of **Gemini-3-Pro** (Baseline) against the reasoning generated by *TFRBench*. The forecasting task involved predicting Amazon's stock movement.

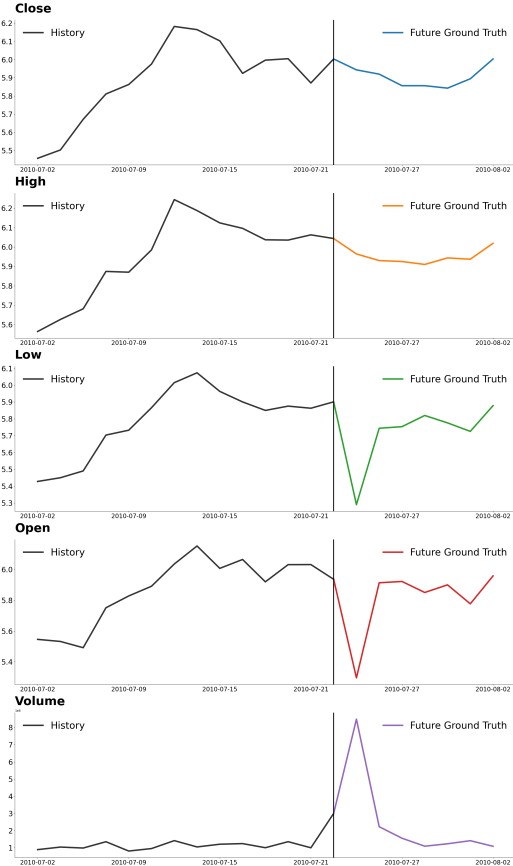

*Figure 17.* **Visualizing a sample from Amazon dataset.** The Ground Truth (Blue/Orange/Green) shows a significant price drop and volume spike around July 23, driven by earnings news.

As illustrated in Figure 17, the difference in forecasting performance is driven entirely by the quality of the underlying reasoning. The Baseline model, lacking a rigorous verification loop, hallucinates a sophisticated technical narrative to justify a stable forecast. It interprets data artifacts as "7-step seasonal cycles" and "zeroed-out trends," completely **missing the causal driver**. In contrast, TFRBench correctly identifies the specific external event, the Q2 2010 Earnings Report and accurately predicts the resulting negative market reaction and volume surge.

Below, we present the side-by-side reasonings. We annotate the Baseline's hallucinations in **red** and TFRBench's correct attributions in **green**.

---

**Baseline (Gemini-3-Pro): Narrative Bias**

The forecasting process begins by analyzing the pre-computed decomposition features. The 'Seasonality' component exhibits a **clear 7-step repeating cycle**, evidenced by the matching values between indices 12-21 and 13-22 (e.g., Close seasonality repeats -0.028, 0.142). This 7-day cycle is extrapolated for the next 7 steps. The 'Trend' component for the last three observed days is zeroed out (likely an artifact), so the **underlying trend is reconstructed** by subtracting the seasonality from the raw data for these days. The average of these reconstructed values (Close approx 5.94, High approx 6.04, Low approx 5.88, Open approx 6.00) serves as the baseline trend for the forecast horizon, **assuming stability after the recent volatility**. The 'Volume' channel shows a massive outlier spike on the last day; therefore, a **mean-reversion approach is applied**, setting the baseline volume to approximately 120M before adding the seasonality components. Finally, the forecast is generated using the additive model (Forecast = Baseline Trend + Extrapolated Seasonality), with post-processing to ensure structural consistency where High must be greater than or equal to both Open and Close, and Low must be less than or equal to both.

---

*TFRBench*

**Close:**
- **Cross-Channel Analysis:** The Close price is highly correlated with the Open, High, and Low channels and will move in the same direction, representing the final consensus price of the day. A significant drop in the Close price will be accompanied by a similar drop in the other price channels. **This price drop is expected to be correlated with a significant spike in the Volume channel, as major news events spur higher trading activity**.

- **Trend Component Plan:** The recent upward trend has shown signs of flattening. Project the trend to continue this flattening, with a slight downward bias in anticipation of and reaction to the upcoming earnings event.

- **Seasonality Component Plan:** The established weekly seasonality should be maintained, representing the typical ebb and flow of trading patterns within the week.

- **Residuals & Event Plan:** The primary event is the market's negative reaction to the **Amazon (AMZN) Q2 2010 Earnings Report** on July 22. Although the report showed strong revenue growth, the stock is expected to react negatively as results or guidance may miss elevated expectations. This will cause a **significant negative shock** to the residuals on the following trading day (July 23). This shock should be modeled to cause a moderate price drop, resulting in a Close price significantly below the prior day's close of $6.00. The 'Dodd-Frank Wall Street Reform and Consumer Protection Act Signed into Law' on July 21 is expected to add minor, non-directional noise but **not a significant directional shock**.

**High:**
- **Cross-Channel Analysis:** The High channel is fundamentally linked to the other price channels, but the gap between High and Low will widen, defining an increase in daily volatility. Following the negative market reaction to the earnings report, the entire trading range will shift down. Therefore, the High price will also drop significantly, with **the new high on July 23 being lower than the previous day's high.**

- **Trend Component Plan:** The trend for the High channel will mirror the Close channel's trend, continuing its recent flattening trajectory before shifting downward due to the event.

- **Seasonality Component Plan:** The established seasonal pattern for daily price peaks should be applied, following the general shape of the other price channels' seasonality.

- **Residuals & Event Plan:** The market reaction to the 'Amazon (AMZN) Q2 2010 Earnings Report' will apply strong downward pressure on the residuals for the High channel on July 23. The new daily High will be substantially lower, reflecting the new, lower valuation of the stock. The magnitude of this downward shock should be consistent with the overall downward shift in the price range.

**Low:**
- **Cross-Channel Analysis:** The Low channel sets the bottom of the daily trading range and is intrinsically tied to the other price channels. **A sharp downward move in the Open and Close will push the Low even further down. The Low will define the bottom of the new, lower trading range post-earnings. The increased volatility means the Low is expected to drop sharply, widening the spread between the High and Low prices.**

- **Trend Component Plan:** The trend for the Low channel will follow the overall price trend, which is projected to be flat to slightly down, reflecting the recent price action and anticipating the event impact.

- **Seasonality Component Plan:** Apply the established weekly seasonal pattern, which dictates the typical rhythm of intraday lows throughout the week.

- **Residuals & Event Plan:** **A significant negative shock must be applied to the residuals for the Low channel on July 23, following the market's reaction to the 'Amazon (AMZN) Q2 2010 Earnings Report'.** This will establish a new, much lower floor for the stock price, dropping from around $5.90 to the low $5.30s, capturing the day's most pessimistic valuation.

**Open:**
- **Cross-Channel Analysis:** The Open price sets the initial tone for the trading day and is a leading indicator for the High, Low, and Close of that same day. A large negative gap in the Open will pull the entire day's trading range downwards. It is the primary mechanism through which the shock from the after-hours earnings announcement is transmitted to the market.

- **Trend Component Plan:** The trend for the Open price will follow the same flat-to-downward trajectory as the other price channels, reflecting a pause in the prior uptrend.

- **Seasonality Component Plan:** The model should incorporate the established seasonal component for the Open price, which may differ slightly from the Close (e.g., Monday morning effects).

- **Residuals & Event Plan:** **The most immediate and sharpest impact of the 'Amazon (AMZN) Q2 2010 Earnings Report' will be on the Open price of July 23. A large, negative shock to the residuals is required to model the price gapping down significantly at the start of trading, from a close of $6.00 to an open in the vicinity of $5.30.**

**Volume:**
- **Cross-Channel Analysis:** **Volume has a strong inverse relationship with price during this specific event. As the Open, High, Low, and Close prices drop in reaction to the market's interpretation of the earnings news, Volume is expected to surge.** This surge represents intense activity, including selling by existing shareholders and new short-selling positions.

- **Trend Component Plan:** The underlying trend in volume is volatile. Project the recent trend cautiously, understanding that it will be completely overshadowed by the event-driven spike.

- **Seasonality Component Plan:** **The established weekly seasonal pattern for volume should be applied as a baseline, but its effect will be minimal compared to the event's impact.**

- **Residuals & Event Plan:** The 'Amazon (AMZN) Q2 2010 Earnings Report' will cause a very large positive shock to the Volume residuals on July 23. This represents a massive surge in trading activity as the market digests the news. Model a volume spike of **approximately 2.5x to 3x the previous day's volume**, reflecting the intense investor reaction.

This case study confirms that performance in high-entropy domains is not merely a function of model scale, but of *contextual grounding*. The Baseline model's reliance on internal "world knowledge" failed to retrieve the specific earnings date, forcing it to fabricate a technical justification for a stable forecast. TFRBench's multi-agent framework successfully retrieved, verified, and applied the correct event logic, resulting in a forecast that captured the structural break in the time series.

## M. Mitigating "Result-First" Bias: Analysis of Verifier Feedback

To prevent "result-first" confirmation bias, the verifier scores logical coherence and factuality alongside accuracy, ensuring forecasts with flawed reasoning are penalized even if they achieve a low MASE. Below are two examples where candidates achieved a $MASE < 1.0$ but achieved low score via verifier.

**Energy Domain (Electricity Dataset)**

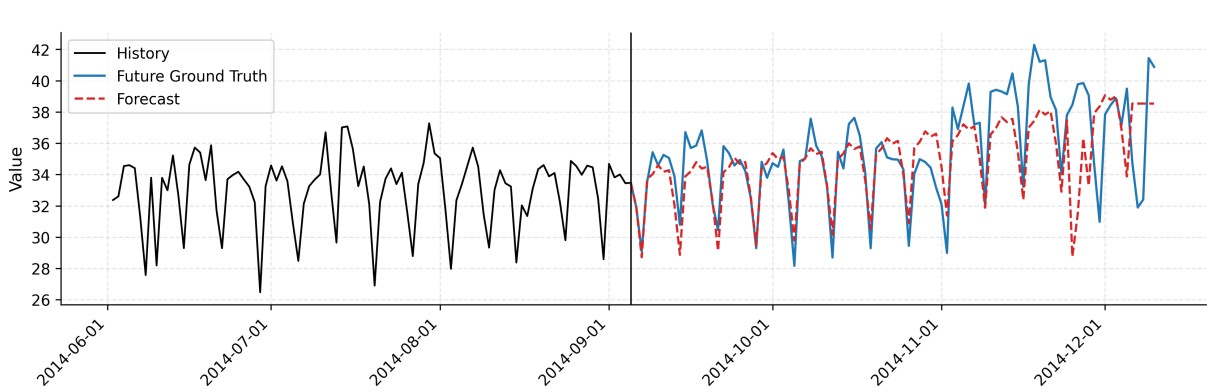

**MASE:** 0.8088 — **Verifier Score:** 2

**Verifier Feedback:** 'The reasoning is incoherent with the resulting forecast, and the forecast itself is flawed. The plan for a stable trend was ignored in favor of a strong upward trend, and the holiday adjustment for Thanksgiving was not applied as described, leading to a major contradiction. Furthermore, the forecast breaks the established seasonal pattern and becomes nonsensical towards the end of the horizon.'

- Cross-Channel Analysis: This is a single-channel forecast for MT_012. The forecast will be generated independently without considering dependencies on other channels.

- Trend Component Plan: The underlying trend exhibits a slow, wave-like pattern and is expected to continue this behavior. Project the trend forward with gentle, non-linear fluctuations, maintaining its recent stability within a general range of 32.5 to 33.5. Avoid projecting a simple straight-line trend; instead, continue the meandering pattern observed at the end of the historical data.

- Seasonality Component Plan: The forecast must be driven by the strong, established 7-day weekly pattern. This pattern consists of consumption rising from Monday to a peak on Wednesday (seasonal factor around +1.5), then declining through the rest of the week to a low point on Saturday (around -1.1) and a significant trough on Sunday (around -4.3). This weekly cycle is the baseline for the forecast and should be applied consistently, unless overridden by a specific event.

- Residuals & Event Plan: Model the residuals to be near zero on non-event days. Apply targeted adjustments for the following verified public holidays, which will cause weekday consumption to resemble weekend levels.

  - **Labor Day (September 1, 2014):** This holiday occurs on a Monday. Apply a significant negative adjustment to override the typical Monday seasonal value. The magnitude should be strong enough to push the consumption level down to a value typical of a Sunday, implying a downward shock in the range of -4.5 to -5.5.

  - **Thanksgiving Day (November 27, 2014):** This is a major holiday on a Thursday. Implement a very strong negative adjustment to depress consumption far below the normal Thursday level. The magnitude should be comparable to the Labor Day shock, aiming to align the day's consumption with a typical Sunday (a downward shock in the range of -5.0 to -6.0).

  - **Day after Thanksgiving (November 28, 2014):** On this Friday, apply a moderate negative adjustment. Consumption should be noticeably lower than a typical Friday but remain higher than on Thanksgiving Day itself. A downward shock in the range of -2.0 to -3.5 is appropriate.

  - **Christmas Day (December 25, 2014):** This holiday falls on a Thursday. Apply a very strong negative adjustment, similar in effect and magnitude to Thanksgiving, to bring consumption down to Sunday levels (a downward shock in the range of -5.0 to -6.0).

  - **Christmas Eve (December 24, 2014):** On this Wednesday, apply a moderate negative adjustment to reflect reduced activity ahead of the holiday. The shock should be noticeable but less severe than the main holiday, in

the range of -1.5 to -2.5.

- **Day after Christmas (December 26, 2014):** On this Friday, apply a moderate negative adjustment, similar to the day after Thanksgiving, to account for continued reduced commercial activity. A downward shock in the range of -2.0 to -3.5 is appropriate.

## Sales Domain (Hierarchical Sales)

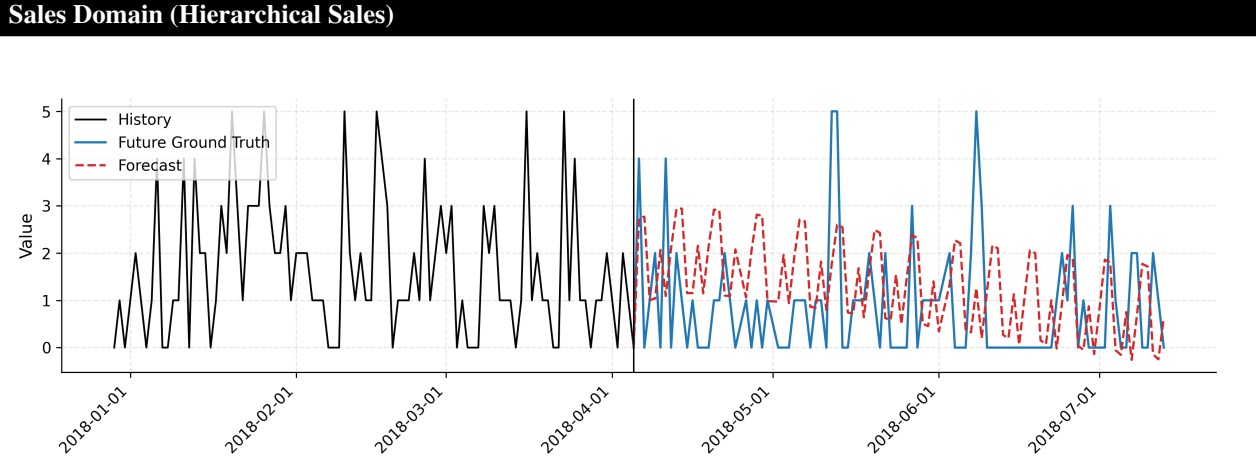

**MASE:** 0.8844 — **Verifier Score:** 2

**Verifier Feedback:** 'While the MASE is good, the reasoning is incoherent because it specifies an upward trend, whereas the forecast clearly trends downward. The forecast is also flawed, predicting impossible negative sales values, which makes the model untrustworthy despite the score.'

QTY_B1_26:

- Cross-Channel Analysis: Pasta sales are expected to be positively correlated with sales of complementary products like pasta sauce and cheese. Conversely, sales are likely negatively correlated with substitute meal bases like rice, potatoes, or ready-to-eat meals as consumers choose one type of meal over another.

- Trend Component Plan: The underlying trend showed volatility in January and February before stabilizing in March. The forecast should begin from this more recent, stable base level, which is around 1.5 to 2.0 units. Project a modest, gradual upward trend, reflecting slow market growth. This trend should be modeled as a gentle, wave-like fluctuation rather than a straight line. Do not extrapolate the sharp drop to zero at the very end of the historical data; treat it as an anomaly.

- Seasonality Component Plan: The dominant 7-day weekly pattern must be consistently applied throughout the forecast horizon. Replicate the pattern of sales peaking on Fridays and Saturdays (an increase of roughly 1 unit over the daily average) and hitting low points on Sundays and Wednesdays (a decrease of roughly 0.8 units below the daily average). The magnitude of these weekly peaks and troughs should remain consistent with historical observations.

- Residuals & Event Plan: Adjust the forecast baseline to account for the impact of specific external events.

  - New Year's Day: For future occurrences, apply a moderate downward adjustment to account for reduced store hours and shopping activity.
  - Super Bowl LII (February 4): This event has a neutral to slightly negative impact on pasta sales. Apply a minor downward adjustment, potentially reducing sales by around 0.5 to 1 unit, as consumers are known to prefer other food categories like pizza and wings.
  - NCAA Men's Basketball Tournament (March 13 - April 2): This event is expected to cause a slight but persistent downward pressure on sales, particularly on weekends. Consumers are more likely to eat at

restaurants or order takeout (e.g., pizza, wings) while watching games. Model this by applying a small negative adjustment, reducing the forecast by approximately 1 unit on weekend days during the tournament.
- Easter Sunday (April 1): This holiday causes a significant negative shock to sales. Implement a strong downward adjustment on Easter Sunday and the following Monday (April 2), reducing the forecast to near-zero or by at least 2-4 units. This reflects grocery store closures and the consumer shift to traditional holiday meals like ham or lamb.

