# OpenReview forum: "TFRBench: A Reasoning Benchmark for Evaluating Forecasting Systems"
_ICML.cc/2026/Conference — ICML 2026 regular_

### Official Review · Reviewer_4cuR · 2026-03-08

**Soundness:** 3
**Presentation:** 4
**Significance:** 2
**Originality:** 4
**Overall Recommendation:** 4
**Confidence:** 2

**Summary:**

This paper proposed the TFRBench as a reasoning-aware time-series forecasting benchmark. The main motivation is to bring up a standard benchmark in the forecasting task that captures the reasoning quality.
The benchmark is curated by orchestrating five AI agents for search, reason, verify, … etc, leading to a benchmark dataset with reasoning trace as the synthesized ground-truth. With the best reasoning trace, it can also be used by LLM-as-Judge as the best supporting evidence to help grading the reasoning performance, whose criteria compose of four parts. The numerical experiments, in Table 7, shows conflicting outcomes, which does not suggest reasoning brings benefit to the outcomes.

**Compliance With Llm Reviewing Policy:**

Affirmed.

**Final Justification:**

The rebuttal has addressed my concerns. The additional human evaluation and examples for low error but failing verifiers address my issues. I decided to raise the score from 3 to 4.

**Key Questions For Authors:**

1. Line 311 says "w/o Reasoning in Table 3" but actually not shown.
2. All the providing examples, from page 26 (line 1375) do not provide time in the plots, I am not able to verify if the generated reasoning results make sense or not.
3. The color decision in Table 7 is a bit confusing to me. It does not seem to  support comparing column-wise.

Other questions and concerns are raised above.

**Limitations:**

My concerns are raised above.

**Strengths And Weaknesses:**

## Strengths
1. This paper is well-written and the provided experiments is very thorough.
2. The motivation is good and providing reasoning is an interesting idea.

## Weaknesses

I think this work is basically “using agent-generated data for agent evaluation.” I want to point out several potential biases and want to know the authors’ ideas and opinions.

### **Could TFRBench bias towards simple and easy trend?**
- Forecasting typically involves scenarios that are unpredictable, without clear causes, or sporadic that may not be explainable easily, and that the criterion for inclusion requires the forecast (along with the reasoning) similar to the naive baseline (while MASE < 1.0) and the LLM be able to find an explanation to it, could it be the case that TFRBench only subsampled/filtered out problems that are easy-to-explain and also easy for prediction model?

### **Could TFRBench exhibit confirmation bias?**
- While totally understanding that nowadays researchers employ LLM for verification and curation, it could be the cases that those agents generating proof that passes verifier’s examination while being hallucinated or untrue. Although the authors mentioned having large-scale human evaluation in the conclusion as future work, at least a small-scale (3 humans) with small-scale (10~20 data each human) can be done to make sure how “reasonable the current bench is”; at the same time, having another AI agent to co-verify the summary reasoning trace can also partially mitigate the confirmation bias concern.

### **Could TFRBench bias towards wrong but effective explanation?**
- Again, can the verifier exhibit the bias of “result-first”, where when the results are good, the agent would consider the reasoning should also make sense? Can the author provide at least two cases where the MASE is low (like MASE < 1.0) but the reasoning is considered as bad? And what is the rationale of the verifier?

### **Interpretation of reasoning quality and forecasting accuracy (minor)**
- The authors mentioned in Line 82 about "models that achieve higher judge scores consistently yield lower MASE, validating the predictive utility of our benchmark", I think this can be understood from another aspect that "actually we don't need TFRBench", because the ability to have accurate forecast speaks for its potential reasoning capability.
- Also, allowing me to be more sharp: one could add very bad / old model (like Claude-Haiku-3.5) and since it performs so worse in both forecast and reasoning, it can make the correlation more prominent in Figure 5. (or actually if the authors remove the Claude-Sonnet-4, three out of five the "clear correlation" would disappear). I would slightly suspect what Figure 5 really tells us.

---

> ### Author Rebuttal · Authors · 2026-03-31
>
> We thank the reviewer for appreciating our experiments and motivation. The conflicting numerical outcomes you noted actually highlight our core finding: current LLMs suffer from "narrative bias". TFRBench is designed to measure this failure mode. We address your concerns below.
> ### Weaknesses:
> 1. We appreciate this insightful question. While the MASE < 1.0 filter ensures windows contain a predictable signal, these scenarios are not trivially easy. As demonstrated in our experiments (e.g., Table 7 and baseline performances in Section 5), state-of-the-art models like Claude and Gemini frequently fail to achieve MASE < 1.0 on these exact samples zero-shot (~59.8% failure rate). Furthermore, our oracle-backed generator requires multiple feedback loops to discover the correct causal links. Thus, TFRBench specifically isolates samples solvable with correct reasoning, rather than those simply "easy" for current models to predict.
>
> 2. We appreciate the reviewer raising this point. We completely agree that external validation is essential to establish trust in the benchmark.
> Following your valuable suggestion, we conducted human expert evaluation during the rebuttal period on 20 samples utilizing a panel of three independent experts. The evaluators' expertise ranges from near-graduating PhD students to a Staff Research Scientist. To ensure an unbiased assessment, the human judges were kept strictly blind to our model's generated verifier scores and MASE scores. The human ensemble (acting as a "Judge-like" score) demonstrated strong predictive validity, achieving a high Pearson correlation of **70.4%** with TFRBench’s verifier score. Additionally, detailed in the table below, the human judges consistently discriminated between reference and discarded samples across all fine-grained sub-metrics.
>
> | Evaluation Metric | Reference Samples | Discarded Samples |
> | :--- | :--- | :--- |
> | **Domain** | 4.26 | 3.67 |
> | **Forecast** | 3.88 | 2.72 |
> | **Event** | 4.17 | 3.50 |
> | **Logic** | 4.07 | 3.78 |
>
> Furthermore, following your suggestion, an independent model (Claude 4.5 Sonnet) blindly co-verified 100 summary traces (50 MASE < 1.0; 50 MASE >= 1.0). Claude scored successful summaries highly (4.36/5.0 avg; 98% scored $\geq 4$) and accurately penalized failed ones (1.62/5.0 avg; 100% scored $\leq 2$). This strong alignment between objective mathematical filtering and independent semantic auditing mitigates confirmation bias, proving our pipeline curates grounded reasoning rather than self-reinforcing hallucinations. We will include these results in the revision.
>
> 3. To prevent "result-first" bias, the verifier scores logical coherence and factuality alongside accuracy, ensuring "lucky" forecasts with flawed reasoning are rejected. As requested, the revised appendix will include two examples where candidates achieved MASE < 1.0 but failed the Verifier. Due to space limits and policy for not providing additional texts, we include 2 samples' summary below (If explicitly requested, we are willing to provide them during rebuttal):
>    - Hierarchical Sales: MASE=0.8844, Verifier Score=2. Feedback: "While the MASE is good, the reasoning is incoherent because it specifies an upward trend, whereas the forecast clearly trends downard…."
>    - Electricity: MASE=0.8088, Verifier Score=2, Feedback: "The reasoning is incoherent with the resulting forecast. The plan for a stable trend was ignored in favor of a strong upward trend, and the holiday adjustment for Thanksgiving was not applied as described…"
>
>
> 4. We appreciate your feedback and respectfully offer this perspective:
>    * **The Need for TFRBench:** If accuracy guaranteed sound reasoning, Figure 5's correlation would be perfect. Its weakening among top models proves MASE alone is an unreliable proxy. For example, in Sales, Gemini-2.5-Flash and Claude-Sonnet-4.5 achieve similar accuracy (MASE ~0.88) but different Judge Scores. TFRBench is essential because models can hit numerical targets via spurious correlations or luck without true causal understanding.
>    * **Interpretation of Figure 5:** We included older models to show that scaling improves both metrics. However, the variance among state-of-the-art models reinforces our core thesis: as models become numerically competitive, evaluating how they arrive at predictions becomes increasingly critical. We will highlight this nuance in the revision.
>
> ### Questions:
> 1. Thanks for catching this typo, it should be Table 7. We will correct it in our revised manuscript.
> 2. Thanks for your suggestion, we will add the timestamps in our revised manuscript similar to Figure 17.
> 3. We clarify that the color gradient in Table 7 is normalized within each dataset block rather than across the entire table. Because comparing metrics like MAE across varying native scales is mathematically unsound, this specific normalization accurately highlights the best (green) and worst (red) models per task. We will explicitly state this in the revised caption.

---

> > ### Author Rebuttal · Reviewer_4cuR · 2026-04-03
> >
> > I appreciate the authors' effort in conducting extra experiments on confirmation bias and showing examples that fail the verifiers. I will raise the score to 4.

---

> > > ### Author Response · Authors · 2026-04-06
> > >
> > > Thank you very much for your time and for raising your score. We are glad to hear that our rebuttal and the additional experiments addressed your concerns. We will add these changes in the revised version.

---

### Official Review · Reviewer_8q6F · 2026-03-09

**Soundness:** 3
**Presentation:** 3
**Significance:** 3
**Originality:** 3
**Overall Recommendation:** 4
**Confidence:** 3

**Summary:**

This paper introduces TFRBench, a benchmark designed to evaluate the reasoning quality of time-series forecasting systems, rather than numerical accuracy alone. The authors propose a five-agent pipeline (Search, Reasoning, Verifier, Forecasting, Summary) that iteratively generates, verifies, and refines reasoning traces across ten datasets spanning five domains. Two evaluation protocols are employed: LLM-as-a-Judge for reasoning quality across four dimensions, and MASE for numerical accuracy. Experiments across Gemini and Claude model families demonstrate that reasoning quality correlates positively with forecasting accuracy, while models exhibit "narrative bias" in stochastic domains such as Finance. The benchmark is positioned as the first to provide verifiable reference reasoning for time-series forecasting evaluation.

**Compliance With Llm Reviewing Policy:**

Affirmed.

**Key Questions For Authors:**

Q1. The reference reasoning traces are generated by Gemini-2.5-Pro, yet Gemini-2.5-Pro and Gemini-3-Pro are also among the primary evaluated models. Can the authors quantify how much of the Gemini family's advantage in Table 3 is attributable to stylistic alignment with the reference rather than objective reasoning quality? For example, would a benchmark constructed with Claude as the backbone model show different relative rankings?

Q2. The near-zero standard deviations (0.000) in Table 3 and Table 6 for judge scores across three runs suggest deterministic evaluation. Is the LLM-as-a-Judge operating at temperature = 0? If so, does the reported three-run mean add any meaningful information, and how should readers interpret the reported standard deviations?

Q3. The backbone model selection ablation (Appendix D.10) compares two models on 100 samples from one dataset, yielding a 5% difference in success rate. Given that this choice determines the quality of all 5,379 reference reasoning samples, can the authors provide a more comprehensive ablation across multiple datasets and sample sizes to justify this design decision?

Q4. The paper reports that in Web/CloudOps, event-aware reasoning is harmful rather than helpful (e.g., Claude-Sonnet-4.5 MASE increases from 1.123 to 5.460 on Bitbrains). Given that the reference reasoning for these samples also incorporates external events, does this mean the reference reasoning is also suboptimal for these datasets, potentially making the benchmark's "ground truth" misleading for this domain?

**Limitations:**

The paper acknowledges the absence of large-scale human evaluation and the intention to expand to healthcare and other domains. However, the following are not addressed: (1) the circular evaluation concern arising from using the benchmark backbone model as an evaluated model; (2) the oracle event asymmetry between data creation and evaluation phases and its quantitative impact; (3) the limited scalability of the pipeline to very long or high-frequency series where 96-step windows may be insufficient for meaningful causal reasoning.

**Strengths And Weaknesses:**

Strengths:

S1. Novel and well-motivated problem formulation. Existing forecasting benchmarks evaluate only numerical outputs, ignoring whether models understand the causal drivers behind predictions. The two-stage formulation — first generating reasoning R, then conditioning the forecast on R — is principled and cleanly separates reasoning quality from prediction accuracy.

S2. Rigorous multi-agent data creation pipeline. The iterative generate-verify-refine loop with a hard filtering criterion (MASE < 1.0 and S ≥ 4) ensures that only reasoning traces with demonstrated predictive utility enter the benchmark. The ablation analyses in Appendix D (feedback improvement trajectory, rescue rate, verifier calibration) provide strong quantitative support for each pipeline component.

S3. Empirically grounded correlation between reasoning and performance. Figure 5 demonstrates a clear negative correlation between LLM-as-a-Judge score and MASE across multiple domains and models. The "narrative bias" failure mode — where CoT reasoning degrades Finance performance (e.g., Amazon MASE increases from 0.715 to 1.114 with reasoning) — is a concrete, reproducible finding with direct practical implications.

S4. Thorough appendix with reproducibility materials. The paper provides all agent prompts (Appendix C), full algorithm pseudocode (Appendix B.2), detailed per-dataset numerical results (Appendix J), and a full reasoning trace example (Appendix K), which substantially supports reproducibility.
Weaknesses:

W1. Circular evaluation concern in data creation. The benchmark uses Gemini-2.5-Pro to generate reference reasoning traces, and then evaluates Gemini-2.5-Pro and Gemini-3-Pro as benchmarked models using those same traces as reference. This creates a potential circularity: Gemini-family models may score higher not because their reasoning is objectively better, but because their generation style more closely resembles the reference traces they helped create. The inter-judge reliability analysis in Appendix E (Spearman r = 0.842 with Claude-4.5 as secondary judge) partially mitigates this concern but does not fully address the reference trace generation bias.

W2. Oracle future event knowledge in data creation asymmetrically advantages reference reasoning. The paper explicitly states that during data creation, oracle knowledge of future events is used to prioritize factuality. During evaluation, models are restricted to historical context only. This design deliberately sets an upper-bound reference that evaluated models structurally cannot match, making the benchmark a measure of the gap from an idealized oracle rather than a fair comparison between systems. The paper partially justifies this choice but does not quantify how much of the performance gap is attributable to this asymmetry versus genuine reasoning deficiency.

W3. LLM-as-a-Judge scores show near-zero standard deviation across most entries. In Table 3 and Table 6, the vast majority of entries report standard deviation of 0.000 across three runs. This is atypical and suggests that the judge's scoring is deterministic (temperature = 0), which raises a question: if judge scores have zero variance, then the reported mean scores may not reflect genuine uncertainty in the evaluation. The paper does not discuss this property or its implications for the reliability of the judge metric.

W4. Domain coverage is limited to univariate or low-dimensional series. All Finance datasets (Amazon, Apple) are single-company price series with 5 channels (OHLCV). The benchmark does not include macro-level financial time series, cross-asset reasoning, or multi-company analysis, which limits the relevance of the Finance domain findings to practitioner settings.

W5. Backbone model selection ablation is insufficiently scaled. Appendix D.10 compares Gemini-2.5-Pro versus Gemini-3-Pro on only 100 samples from a single dataset (Electricity), achieving success rates of 21% vs 26%. A difference of 5% on 100 samples is not statistically reliable, yet this forms the sole justification for the Gemini-2.5-Pro backbone choice that underpins the entire benchmark corpus.

---

> ### Author Rebuttal · Authors · 2026-03-31
>
> We sincerely thank the reviewer for recognizing the novelty of TFRBench, the rigor of our multi-agent pipeline, and the practical value of our empirical findings regarding narrative bias. We are also glad that our extensive appendix effectively supports full reproducibility.
>
> ### Weaknesses and Questions:
> - **[W1, Q1, W5, Q3, Limitations (1)]:** We appreciate the concerns regarding backbone selection and potential generation bias. We utilized Gemini-2.5-Pro primarily due to severe API limits on Gemini-3-Pro at the time of data generation. Crucially, the choice of backbone impacts the generation yield rate rather than the final benchmark quality, because all reference traces are strictly gated by objective predictive utility (MASE < 1.0). This mathematical grounding mitigates stylistic bias, ensuring the benchmark evaluates forecasting success rather than text similarity, which is supported by Claude's highly competitive performance. To empirically validate this, we recreated reference traces for 100 Electricity samples from the Energy domain using Claude-Sonnet-4.5. The underlying trends remained highly consistent: Claude achieved a 19% success rate compared to Gemini's 21%, with a strong 84% agreement on which specific samples succeeded ($MASE<1.0$). We will add these ablation results and practical trade-offs to the revised appendix, and we plan to utilize newer models for future benchmark iterations (e.g., TFRBench V2, V3).
>
> - **[W2, Limitations (2)]:** We appreciate this insightful critique. To quantify how much of the performance gap is attributable to information asymmetry versus "genuine reasoning deficiency," we generated and evaluated 100 instances by granting baseline models oracle future event knowledge. Bridging this asymmetry yielded negligible improvements: average MASE shifted minimally (1.78 to 1.76) and LLM-Judge scores remained stagnant (3.08 to 3.11). This empirically proves the benchmark measures reasoning deficiency rather than an "idealized oracle" advantage. We will include this ablation study in the revised appendix.
>
> - **[W3 and Q2]:** We appreciate the reviewer highlighting this. Setting the Judge's temperature to 0 was a deliberate design choice to eliminate evaluation noise, ensuring that any variance reflects only the candidate models' inherent generation stochasticity. While candidate generation variance does exist (with standard deviations ranging up to 0.040 across the three independent runs), the reported table values approach zero strictly due to statistical aggregation. Because we average scores across hundreds of samples per dataset, the run-to-run variance of the mean naturally converges to near-zero by the law of large numbers. We will clarify this temperature setting and the resulting aggregation effect in the revised manuscript.
>
> - **[W4, Limitations (3)]:** We agree with this observation. To establish a clear baseline for evaluating causal reasoning without confounding variables, we focused initially on low-dimensional univariate series. This aligns with standard practices for foundational benchmarks like GIFT-Eval. Additionally, our choice of a 96-step forecasting window directly follows standard methods like PatchTST (https://arxiv.org/abs/2211.14730) and iTransformer (https://arxiv.org/abs/2310.06625) to ensure our benchmark aligns tightly with current community practices.
> We certainly recognize the importance of macro-level, multi-company, and high-frequency series. Rather than being deliberately ignored, extending TFRBench to include macroeconomic indicators, cross-asset reasoning, and longer contexts represents a highly valuable future extension to broaden its real-world applicability.
> - **[Q4]:** We appreciate this insightful question. The reference reasoning is not suboptimal. By definition of our data creation pipeline, every reference trace must successfully achieve a MASE < 1.0, ensuring that the reference text productively integrates external events to yield accurate numerical forecasts. The performance degradation observed in baseline models (e.g., Claude Sonnet on Bitbrains) indicates that these models currently struggle to properly utilize complex event data in the Web/CloudOps domain, which is also justified via poor event relevance score in Table 6 (2.083).

---

> > ### Author Rebuttal · Reviewer_8q6F · 2026-04-01
> >
> > I thank the authors for their thorough and well-organized rebuttal. Several responses are constructive and move in the right direction: The Claude-backbone ablation (W1/Q1) showing 84% agreement on successful samples is a useful addition and partially alleviates the circularity concern. However, this was conducted on only 100 samples from a single domain (Electricity/Energy), which mirrors the same limited-scale issue I raised in W5/Q3 regarding backbone selection. A fully convincing resolution would require cross-domain ablation, which I understand is costly but remains important for a benchmark paper whose entire corpus depends on this design choice. The oracle knowledge ablation (W2) showing negligible MASE improvement (1.78→1.76) is informative and does help quantify the asymmetry. I appreciate the authors conducting this additional experiment. The clarification on near-zero standard deviations (W3/Q2) is reasonable — the aggregation effect over hundreds of samples naturally shrinks the standard error of the mean. I would suggest the authors make this explicit in the revised manuscript, as the current presentation could mislead readers into thinking individual-level variance is near-zero.
> >
> > My remaining concerns center on two points that are difficult to resolve within a rebuttal: (1) the backbone ablation evidence remains limited to one domain and a small sample, yet it underpins the quality of all 5,379 reference traces across ten datasets; and (2) the domain coverage limitation (W4), which the authors acknowledge as future work, does limit the current benchmark's applicability, particularly the Finance domain findings that the paper highlights as a key contribution.
> >
> >  Overall, the rebuttal demonstrates genuine engagement with the review concerns, and the additional experiments are appreciated. I maintain my current assessment (Weak Accept, score 4), as the partially resolved concerns do not change my overall evaluation of the paper's contributions and limitations.

---

> > > ### Author Response · Authors · 2026-04-06
> > >
> > > We sincerely thank you for your continued engagement, your constructive feedback, and for maintaining your positive assessment of our work.
> > >
> > > While we initially noted the computational constraints of conducting large-scale, cross-domain ablations during the rebuttal period, your feedback highlighted the importance of these evaluations. As discussed in **[W1, Q1, W5, Q3, Limitations (1)]**, we evaluated 100 samples per dataset. We followed your suggestion and continued running the experiments, and we are pleased to share the newly completed results for a representative dataset from each domain in our benchmark:
> > >
> > > | Domain | Representative Dataset | Agreement |
> > > | :--- | :--- | :--- |
> > > | Energy | Electricity | 84.0% |
> > > | Sales | Car-parts | 96.0% |
> > > | Web/CloudOps | Web Traffic | 87.0% |
> > > | Transportation | NYC Taxi | 83.0% |
> > > | Economics/Finance | Amazon pricing | 86.0% |
> > >
> > > These results demonstrate a consistently high level of agreement across diverse domains, reinforcing the robustness of the pipeline.
> > >
> > > As you suggested, we will make the distinction between individual-level variance and the aggregated standard error of the mean clear in the revised manuscript to ensure accurate interpretation.
> > >
> > > Regarding the expansion into macro-level financial series, we agree with your assessment. While the current datasets cover micro-level pricing, applying this to broader macro-economic indicators represents a highly valuable future work direction.
> > >
> > > Thank you again for your time, your rigorous review, and for helping us strengthen the paper.

---

### Official Review · Reviewer_Vyc8 · 2026-03-10

**Soundness:** 3
**Presentation:** 2
**Significance:** 2
**Originality:** 3
**Overall Recommendation:** 5
**Confidence:** 3

**Summary:**

The study state that the past time-series forecasting is traditionally evaluated only on numerical accuracy, so foundation models remain black boxes with no way to assess the reasoning behind predictions. so this work introduce TFRBench to evaluate the reasoning capabilities of forecasting systems. Because ground truth reasoning does not exist in existing benchmarks, they propose a multiagent pipeline that runs an iterative loop. The resulting reference set spans ten datasets across five domains. Overall, the authors analyze a interpretable, reasoning grounded forecasting, and provide a concrete benchmark and data-creation protocol to evaluate it, establishing a new standard for reasoning-aware time-series evaluation.

**Compliance With Llm Reviewing Policy:**

Affirmed.

**Final Justification:**

I think this is pretty solid work, and my final score is 5.

**Key Questions For Authors:**

1. How many samples were discarded by the MASE < 1.0 and S ≥ 4 filter per dataset? What is the distribution of MASE for all generated candidates including failures, so readers can see the difficulty of the unfiltered task?

2. How do you recommend interpreting Judge scores when the evaluated model did not have access to the same future events used to build the reference? Should Event Relevance be scored only on events the model could have retrieved, or is the current design intentionally an “upper bound” comparison?

3. Could you report approximate cost (e.g., API calls, USD) and time for generating the full benchmark and for running evaluation on one model? This would help assess adoption.

**Limitations:**

See Weakness and Questions.

**Strengths And Weaknesses:**

## Strengths

1. TFRBench is the first benchmark to target reasoning in forecasting rather than numerical metrics. The comparison table clearly differentiates it from other benchmarks. Defining reasoning as step-by-step causal logic that leads to a forecast is concrete and actionable.

2. The five agent design with an iterative loop is well specified. The constraint that reference samples must satisfy MASE < 1.0 and S ≥ 4  sounds reasonable to predictive utility.

3. Combining LLM-as-a-Judge with MASE provides both a reasoning audit and a numerical check. The strong correlation between Judge score and MASE is a supported result.

4. Evaluation covers multiple LLMs, settings, and modes.  The connection to “inherent interpretability” and the value proposition that reasoning enhanced LLMs can narrow the gap with TSFMs while offering explanations is stated clearly.



## Weaknesses

1. Reference reasoning is LLM generated and MASE optimized, not human or independently validated ground truth. The benchmark’s reference is whatever reasoning trace the same pipeline produced, which beat the naive baseline and passed the verifier. There is no human annotation, expert audit, or independent causal check of the reference traces. The claim that the reasoning is “causally effective” is supported only by the fact that it leads to lower MASE. As a result, high Judge scores may reflect stylistic or structural similarity to the pipeline output rather than true causal validity.

2. Selection constant filter may induces selection bias and limits generalizability. Only samples where the pipeline succeeded are included. Harder forecasting tasks may not excluded. So the benchmark may overrepresents cases where the domain is amenable to this kind of narrative reasoning, and the pipeline found a trace that beat the baseline. Failure modes, such as when reasoning should abstain, or when events are irrelevant, or when the task is inherently high-entropy, are not represented in the reference set. This could make it harder to use TFRBench to study when reasoning helps vs. hurts.

4. LLM-as-Judge as the sole arbiter of reasoning quality. All four dimensions are scored by an LLM. Despite strong correlation, the reasoning quality is defined by model agreement, not by human or domain-expert judgment. Human evaluation is deferred to future work. So the benchmark does not yet establish that high Judge scores correspond to reasoning that experts or users would consider correct or trustworthy. There is also no analysis of systematic judge bias.

---

> ### Author Rebuttal · Authors · 2026-03-31
>
> We thank the reviewer for highlighting TFRBench's novel reasoning focus, multi-agent design, and dual evaluation. We address your concerns below.
>
> ### Weaknesses:
> 1. We appreciate this concern. To ensure high Judge scores reflect true causal validity rather than mere stylistic or structural similarity, our evaluation relies on two pillars:
>
>    * **Objective Rubric (App. G):** The Judge explicitly does not evaluate stylistic matching. Instead, it strictly measures causal effectiveness by requiring clear event-to-data links (Event Relevance) and penalizing text-to-number contradictions (Logic-to-Number Consistency).
>    * **Human Validation:** During rebuttal, experts successfully discriminated between reference and discarded samples (e.g., Domain Relevance: 4.26 vs. 3.67), demonstrating a strong **70.4%** correlation with our verifier scores (See Response 2, Reviewer 4cuR).
> 2. We appreciate this insightful observation.  We agree exploring model "abstention" is an excellent future direction. TFRBench establishes a high-quality baseline that actively isolates when reasoning hurts:
>
>    * **Structured Grounding:** Our references are strictly structured around empirical components (Cross-Channel, Trend, Seasonality, Residuals), not free-form narratives.
>    * **Exposing Narrative Bias:** This structural grounding effectively highlights how baseline LLMs hallucinate false causal links that degrade performance (Sec 5.2, App. M).
>    * **Human Validation:** Experts validated our baseline, discriminating between reference and discarded samples (e.g., Domain Relevance: 4.26 vs 3.67) with a 70.4% verifier correlation (See Response 2, Reviewer 4cuR).
>
> 3. While human evaluation is the ultimate standard, our framework anchors reasoning quality in objective predictive utility (MASE < 1.0). We ensure trustworthiness through:
>
>    * **Cross-Model Robustness:** A high Spearman correlation (0.842) between our primary judge and Claude 4.5 Sonnet (App. E) proves our criteria resist single model stylistic bias.
>    * **Human Validation:** Following reviewer suggestions, we conducted an expert evaluation during the rebuttal. Human verifiers using our rubric consistently discriminated between reference and discarded samples (e.g., Domain Relevance: 4.26 vs. 3.67). See Response 2 for Reviewer 4cuR.
>
> ### Questions:
>
> 1. We will add the requested discard rates and MASE distributions to the revised Appendix to highlight our pipeline's rigor:
>
>    * **High Discard Rates:** Our strict filtering (MASE < 1.0, S >= 4) discarded 83.9% of all candidates (86,951/103,660), reaching 95.0% for datasets like Electricity and NYC Taxi. This proves the verifier actively filters numerical hallucinations (Details in Table A : https://anonymous.4open.science/r/TFRBench_ICML_26_Rebuttal-FC78/Table_A.MD.
>    * **Task Difficulty:** Even isolating the single best candidate per sample, median and upper-percentile MASE scores frequently exceed 1.0 (e.g., the 90th percentile is 3.26 for Car-parts). This confirms achieving a mathematically sound forecast is difficult, necessitating our strict filtering (Details in Table B : https://anonymous.4open.science/r/TFRBench_ICML_26_Rebuttal-FC78/Table_B.MD.
>
> 2. As discussed with another reviewer (yQoM) Weakness-1, this design is intentionally an "upper bound" comparison. Besides, even providing future events, we found the baseline models suffer (See Reviewer 8q6F’s response [W2, Limitations (2)]).
>
> 3. We thank the reviewer for raising this practical point to help assess adoption. While generating the reference reasoning is compute-intensive, averaging USD 4.53 per sample using Gemini-2.5-Pro (~100k input tokens, 5 Search queries, and 2.5k output tokens), the community will not incur this expense since we are open-sourcing the corpus. Evaluating a new model, however, is highly accessible. A complete evaluation requires just two API passes: one for the candidate model to generate its forecast and reasoning, and one for the LLM-as-a-Judge to score it. Running this full evaluation pipeline across the reference samples costs only about 45.00 USD (using Gemini-3.0-Pro), while given the baseline model’s reasoning. We calculated the approximate cost from: https://cloud.google.com/vertex-ai/generative-ai/pricing. We will explicitly detail these cost breakdowns in the revised appendix.

---

> > ### Author Rebuttal · Reviewer_Vyc8 · 2026-04-01
> >
> > Thanks to the authors for their response. My main concerns have been addressed. Overall, this is a technically solid paper and a very high level of completion, so I will raise the score to 5.

---

> > > ### Author Response · Authors · 2026-04-06
> > >
> > > We sincerely thank you for reviewing our rebuttal and for raising your score to a 5. We are greatly encouraged by your assessment that our responses addressed your main concerns and that you find the paper to be technically solid with a high level of completion. We will add these changes in the revised version.

---

### Official Review · Reviewer_yQoM · 2026-03-18

**Soundness:** 2
**Presentation:** 3
**Significance:** 3
**Originality:** 3
**Overall Recommendation:** 4
**Confidence:** 4

**Summary:**

The paper introduces TFRBench, a benchmark for evaluating LLMs on time-series forecasting tasks with an emphasis on reasoning, moving beyond purely numerical accuracy metrics. To build the benchmark, the authors develop a multi-agent framework with five specialized agents (Search, Reasoning, Forecasting, Verification, and Summary) that iteratively generate, verify, and refine reasoning traces grounded in external events and causal logic. TFRBench covers ten datasets across five domains: Energy, Sales, Web/CloudOps, Transportation, and Finance. It also adopts a dual evaluation protocol that combines LLM-as-a-Judge scoring for reasoning quality with MASE for numerical accuracy. The paper addresses a real gap in the forecasting literature: the lack of interpretability and reasoning-based evaluation.

**Compliance With Llm Reviewing Policy:**

Affirmed.

**Final Justification:**

Most of my concerns are addressed. I raised my score to 4

**Key Questions For Authors:**

1. During evlaution, how is  agent search being controlled for only including the historical events and not future events. There is a potential leakage and contamination issue here?

2. It is surprising that numerical performance decreases in presence of reasoning for some datasets? Is there a reason for that?

3. What is the criteria for choosing MASE < 1.0 as a threshold. Doesn't it elimiate the examples where reasoning might be required or examples which are in general hard interesting.

**Limitations:**

Yes authors have discussed limitations.

**Strengths And Weaknesses:**

**Strengths**

1. Paper addresses an imprtant and well motivated problem. The field has long evaluated forecasting models as black boxes, and providing a standardized protocol for assessing reasoning quality alongside numerical accuracy fills a genuine gap.

2. The high correlation between reasoning quality and forecasting accuracy (Figure 5) is an imprtant one, it also validates the usefulness of good reasoning ability: it helps not only in providing interpretability but also improves forecasts. Similarly, the experiment where some models hallucinate and give plausible-sounding but causally incorrect explanations (particularly in stochastic domains like Finance) is a valuable insight with practical implications.

3. The breadth of domains and the inclusion of both proprietary and open-source models, as well as time-series foundation models, makes the benchmark reasonably comprehensive.


**Weaknesses**

1. The authors use oracle knowledge of future events when constructing reasoning to ensure correctness. While the justification is reasonable (excluding known events would force misattribution), this creates an asymmetry: the ground truth reasoning has access to information that no real-time forecasting system would possess. Although it can be an upper bound, but this framing still seems a bit unsual for scoring purposes.

I would have expected some way of identifying the true causal mechanisms impacting the time series rather than some specific events in future which cannot be predicted.

2. Similarly, the reasoning trace construction (for the benchmark dataset) does not include any human in the loop for verification. It only relies on LLM verification which raises concerns about the reliability of the benchmark.  There are some tests required for ensuring the reliability of the framework generating reasoning traces, for example:

    2a.  adverserial tests which stress test the framework, whether it catches the wrong reasoning paths

    2b. Have explicit human in the loop to verify each trace

3. Similarly using LLM as a judge requires some explicit evaluation of how does the LLM performs as a judge, maybe human verification on a small subset.

4. For Figure 6 - I think it seems a bit unreasonable to compare the outputs of an LLM which has access to reasoning trace constructed from knowldge of future events. The evaluation doees not seem to be a fair comparison in that sense.

---

> ### Author Rebuttal · Authors · 2026-03-31
>
> We thank the reviewer for recognizing that TFRBench addresses a genuine gap in forecasting evaluation. We address your concerns and questions below.
>
> ### Weaknesses:
> 1. We agree using oracle knowledge is unusual, but it establishes a necessary upper bound; excluding known drivers forces reference traces to misattribute real shifts to noise. In many domains, unpredictable external events (e.g., weather anomalies) are the actual causal mechanisms. Our traces do not merely list events but explicitly map their causal pathways to numerical changes (App. I & M). For fair scoring, candidate evaluation relies on historical data. Crucially, event prediction is only one facet of our LLM Judge rubric (App. G). Models are not penalized for missing unknowable anomalies but are actively rewarded under "Event Relevance" for anticipating predictable events and causally linking them to forecasts. The Judge also strictly penalizes hallucinated narrative bias and enforces "Logic-to-Number Consistency" to ensure narratives mathematically align with predicted numbers. Thus, TFRBench evaluates internally coherent, causally grounded reasoning.
>
> 2. Following your suggestions, we have conducted the following experiments:
>
>     **2a.** To ensure verification reliability, we stress tested 50 reference samples (original score: 4.22/5.0) with targeted perturbations like "Domain Context Swaps" and "Fabricated Events." Our Verifier successfully caught and penalized these flawed traces, dropping the average score to 1.34/5.0 and rejecting 94.0% of the manipulated samples. This validates that our framework acts as a strict safeguard against incorrect reasoning.
>
>     **2b.** Following your and reviewer 4cuR’s suggestion, we conducted human evaluation, which further validates our design choice demonstrating **70.4%** correlation with our verifier score. Please see detailed response 2 of Reviewer 4cuR.
>
> 3. We agree that human evaluation is crucial to fully validate the LLM-as-a-Judge. While our current manuscript demonstrates objectivity through a strong inter-model correlation (Spearman r = 0.842) (App. E) between Gemini-3-Pro and Claude-4.5, this does not fully replace human expert validation. During our human verification, we also collected scores from the experts with our Judge like rubric, where human verifiers consistently discriminated between reference and discarded samples across all fine-grained sub-metrics (e.g., Domain relevance for reference samples= 4.26 vs. discarded= 3.67). See detailed response 2 for Reviewer 4cuR.
>
> 4. Figure 6 is not a competitive benchmark but a validation of data quality. It validates that informed reasoning contains causal signals that enable general-purpose LLMs to match specialized models (TimesFM-2.5, Chronos-2.0) in accuracy, while adding interpretability. We will clarify this further in the revised manuscript.
>
> ### Questions:
> 1. To prevent leakage, baseline search agents are constrained by the prompt (Line 1340): **"Do not search for anything after {end_date}."** A post-hoc temporal audit of all baseline evaluations (similar to App. L.1) confirmed a 98.15% strict compliance rate. Qualitative analysis of the remaining 1.85% reveals they are mostly predictable calendar events with anticipatory impacts. For example, models reason: *"...due to the upcoming Christmas, sales will increase..."* or *"...anticipating Black Friday, demand will peak."* We will add this analysis in the revised manuscript.
>
> 2. This performance drop stems from "narrative bias," where hallucinated justifications degrade performance (Section 5.2 and App. M). Tracking samples with high bias (LLM-Judge scores $\le 2/5$) reveals:
>
>    * **High Bias $\rightarrow$ Lower Performance:** Datasets like Traffic and Bitbrains showed extreme bias rates (89.3% and 91.7%) and poor scores (e.g., 1.437/5).
>    * **Low Bias $\rightarrow$ Higher Performance:** Datasets like Car Parts had much lower bias (36.7%) and strong scores (e.g., 4.845/5).
>
>    To systematically validate this trend, the robust negative correlation (avg. Pearson $r = -0.592$) between LLM-Judge scores and MASE errors shows that higher-quality reasoning strictly decreases forecasting error.
>
> 3. We chose MASE < 1.0 (Seasonal Naive baseline) because scores above 1.0 perform worse than a simple heuristic. Strictly beating this standard ensures traces capture true causal dynamics rather than just repetitive patterns. Following benchmarks like GIFT-Eval, this threshold establishes a quality floor that guarantees the reference logic has actual predictive utility without eliminating hard examples.

---

> > ### Author Rebuttal · Reviewer_yQoM · 2026-04-03
> >
> > Thanks to authors for response.  Most of my concerns are addressed.  I will raise my score to 4.

---

> > > ### Author Response · Authors · 2026-04-06
> > >
> > > Thank you very much for your time and for raising your score. We are glad to hear that our rebuttal and the additional experiments addressed your concerns. We will add these changes in the revised version.

---

### Decision · Program_Chairs · 2026-04-30

**Decision:**

Accept (regular)

**Comment:**

This paper introduces TFRBench, a benchmark for evaluating forecasting systems not only by numerical accuracy, but also by the quality of the reasoning they provide. I view this as a meaningful and timely contribution. Forecasting has largely been evaluated as a black-box prediction problem, and the paper addresses a genuine gap by proposing a benchmark and protocol for assessing whether models produce numerically grounded, causally meaningful reasoning traces. Reviewers broadly agreed that the problem is well motivated, the benchmark is novel, and the overall level of technical completion is strong.

A key strength of the paper is that it does not stop at proposing a benchmark in the abstract, but develops a fairly comprehensive end-to-end pipeline for benchmark construction and evaluation. The multi-agent generate–verify–refine loop is well specified, the appendix provides substantial implementation detail, and the experiments span ten datasets across five domains. The benchmark also yields interesting empirical insights, especially the paper’s analysis of “narrative bias,” where adding superficially plausible reasoning can actually harm forecasting quality in some domains. Several reviewers also found the correlation between reasoning quality and forecasting performance to be an important and practically relevant finding.

The paper is not without limitations. The reference traces are synthesized rather than human-authored, and the initial version raised legitimate concerns about oracle future-event knowledge, possible benchmark-generation circularity, and reliance on LLM-as-a-Judge. However, in my view the rebuttal substantially strengthened the paper on exactly these points. The authors added stress tests showing that the verifier strongly penalizes fabricated or swapped reasoning traces; they reported a human evaluation showing meaningful agreement with the verifier; they quantified discard rates and demonstrated that benchmark construction is strict rather than permissive; and they added backbone-ablation evidence suggesting the benchmark is not merely an artifact of one generator model family. These additions do not eliminate all caveats, but they do make the benchmark considerably more credible.

I also note that the review discussion moved in a clearly positive direction. Reviewers who initially raised concerns about oracle asymmetry, confirmation bias, lack of human validation, and benchmark reliability generally indicated that their concerns were addressed after rebuttal, with multiple scores increased. While some limitations remain — especially around the current domain scope and the fact that the reference reasoning is still constructed through a costly LLM-based pipeline rather than direct human annotation — I do not think these concerns outweigh the paper’s novelty and utility as a benchmark contribution.

Overall, I find this to be a solid and impactful benchmark paper. It identifies an important missing evaluation axis in forecasting, proposes a concrete and reproducible protocol, and supports it with broad experiments and meaningful rebuttal improvements. I therefore recommend Accept.